

# A climatology of cold pools distinct from background turbulence at the Eastern North Atlantic observations site

Mark A. Smalley[1,2], Mikael K. Witte[1,2,3], Jong-Hoon Jeong[1], Maria J. Chinita[1,2]

[1]Joint Institute for Regional Earth System Science and Engineering, University of California, Los
Angeles, Los Angeles, California
[2]Jet Propulsion Laboratory, California Institute of Technology, Pasadena, California
[3]Naval Postgraduate School, Monterey, California

*Correspondence to*: Mark A. Smalley (mark.a.smalley@jpl.nasa.gov)

**Abstract**

We develop an algorithm to identify cold pools at the DOE's Eastern North Atlantic (ENA) site on Graciosa Island and examine the statistics of retrieved cold pools for the entire observational record from late 2016 to 2023. The retrieval strategy
relies on leveraging above-background bivariate deviations in near-surface temperature and water vapor mixing ratio from the ENA station time series. Cold pools at ENA tend to be weak with a prominent annual cycle peaking in the cooler months and caused by reductions in the background turbulence during those months. Often, surface rain events are not associated with cold pools due to a combination of factors including but not limited to high background turbulence, high relative humidity, and low rain rate. The retrieval correctly identifies cold pools that are not associated with observed surface rain at
the met station. Understanding the factors that lead to the formation of weak cold pools will lead to a greater understanding of the dynamics of the marine boundary layer at ENA and how those dynamics feed back to the cloud morphological structures.

**Short Summary**

Evaporation of rain leads to cooler and sometimes moister surface conditions (cold pools), which can lead to further convection that alters convective, cloud, precipitation, and radiation properties. We introduce a new method of measuring cold pools, which accounts for the seasonal and daily changes in dry air turbulence in which the cold pool signatures are embedded. We then apply it to 8 years of observations in the north midlatitude Atlantic Ocean.



## 1 Introduction

State of the art earth system models continue to struggle to simulate the geometrically thin but highly reflective stratocumulus clouds that frequently cover the eastern portion of subtropical oceans (Sc; Jiang et al., 2021), with much of their uncertainty in global cloud feedbacks being traced to the representation of Sc clouds (Klein et al., 2017; Scott et al., 2020; Myers et al., 2021; Ceppi and Nowack, 2021). The transitions of boundary layer clouds from stratocumulus to shallow cumulus (Cu) following the trade winds results in a strong increase in the amount of radiation absorbed by the ocean surface (Goren and Rosenfeld, 2014). The breakup of these expansive stratocumulus clouds is largely determined by meteorology and sea surface temperature (Bretherton and Wyant, 1997; Wyant et al., 1997), but recent works (Eastman and Wood, 2016; Yamaguchi et al., 2017; Goren et al., 2019; Blossey et al., 2021; Smalley et al., 2021) have emphasized the role of precipitation in determining the timing of the cloud regime transition. It is therefore imperative that we understand the physical connections between precipitation and marine boundary layer cloud regime change.

Falling precipitation begins to evaporate as it encounters subsaturated air below the cloud layer. This evaporation cools the air, reducing its buoyancy and leading to downdrafts. Once these downdrafts reach the surface, they spread horizontally and form what is defined as a cold pool. These cold pools are characterized by decreases in temperature and increases in moisture, air density, and wind speed, compared to the surrounding near-surface air. Propagating cold pools may induce upward motion by displacing more buoyant air or by colliding with other outward-propagating cold pools, forcing air upwards and potentially leading to further convection that can destabilize the cloud layer and leading to a fundamental change in the character of the boundary layer clouds.

While much of the study of cold pools has been related to spatial organization and development of deep convection (Feng et al., 2015; de Szoeke et al., 2017), here we focus our attention to the much smaller temperature and moisture signals from boundary layer precipitation including warm rain from Sc. Previous studies have examined the cold pools resulting from boundary layer rain. Terai and Wood (2013) leveraged time series of potential temperature from sub-cloud aircraft measurements over the subtropical southeastern Pacific Ocean to retrieve cold pool signatures. They found that small decreases in potential temperature (stronger than -0.36 K) were associated with small gust fronts and increases in both coarse mode aerosols and dimethyl sulfide which is relevant for secondary particle formation in the atmosphere. Wilbanks et al. (2015) used measurements of air density instead of potential temperature and similarly found that temperature decreases were strongly associated with their retrieved density currents. Their threshold roughly corresponds to a change of -0.24 K. In manually selected cases of precipitating post-cold frontal marine stratocumulus clouds, Ghate et al. (2020) used vertically pointing Doppler radar and lidars to analyze 76 drizzle shafts. They found that downdraft strength correlated with cloud-base drizzle intensity but not with drizzle shaft width. Vogel et al. (2021) also studied near-surface temperature deviations above an eastern Caribbean island with (among other requirements) a threshold of -0.05 K/min. They demonstrated associations between precipitation duration, retrieved cold pool strength, and cloud regime categorizations.



Though Terai and Wood (2013), Wilbanks et al. (2015), and Vogel et al. (2021) attempted to remove the influence of small-scale turbulence that is not associated with precipitation-driven cold pools by time-averaging surface meteorological station observations and requiring perturbations to exceed fixed thresholds, boundary layer turbulence exists over a spectrum of spatial scales and intensities and can therefore be expected to contaminate their datasets to varying degrees. Cold pool signatures also exist with a spectrum of intensities, which leaves methods that employ fixed thresholds both unable to capture the smallest cold pools and at the same time unable to remove all the signatures of the background turbulence. In addition, the background turbulence can be expected to change throughout the diurnal and annual cycles, which are themselves variable with geographic location. We therefore seek a method to retrieve cold pools in the presence of, but separable from, the background turbulence in a way that is flexible to diurnal and annual cycles as well as location.

Here, we present a method to measure cold pools from surface-based observation sites with an emphasis on removing the signals of the background turbulence that are *not* associated with precipitation-driven cold pools (Sect. 2). We validate the algorithm using scanning radar and surface rain observations and then apply the algorithm to eight years of surface station observations representing a short climatology in Sect. 3. Section 4 provides an analysis of which confirmed contiguous rainy objects lead to cold pools, and Sect. 5 provides a discussion of our findings.

## 2 Cold Pool Detection Algorithm

### 2.1 Observations from the ARM ENA site

The goal of this work is to confidently diagnose cold pools via one-dimensional temperature and moisture measurements from surface-based observation. We therefore select the Department of Energy (DOE) Eastern North Atlantic (ENA) observatory on Graciosa Island in the Azores, which is characterized by its location at the northern edge of the subtropical ocean weather regime but can also experience mid-latitude synoptic systems (Remillard and Tselioudis, 2015; Mechem et al., 2018; Giangrande et al., 2019). The ENA site contains a comprehensive suite of surface-based instrumentation but this work will mainly utilize observations of near-surface temperature and moisture from the surface meteorological station (MET; *enametC1.b1* datastream; Kyrouac and Shi 2011) and the Meteorological Automated Weather Station (MAWS; *enamawsC1.b1* datastream; Keeler et al., 2017) to retrieve cold pools. The temperature and moisture time series are derived from a serialized combination of the MET and MAWS systems due to elevated noise in the MET temperature deviations during extended time periods. Preceding about 2016 Oct 01 and following about 2017 May 05, the minute-to-minute noise in the 1-minute MET temperatures (shown in Supplemental Fig. 1) is too large to be useful in diagnosing small changes due to cold pools from light rain and virga. Therefore, in this work we utilize the 1-minute temperature and moisture changes from MET between 2016 Jan 01 to 2017 May 05 and then MAWS from 2017 May 05 forward. The main conclusions and results are not significantly affected by the addition of these MET observations to the MAWS observations following the start of 2017 May 05. We note that the MAWS data are provided at 0.1 K resolution and that such coarse resolution is large compared to some of the cold pool signatures examined here. Thus, we perform temporal



averaging (explained later) to reduce noise, but this temporal averaging also smooths the MAWS data such that they take values more finely spaced than 0.1 K.

Observations of surface precipitation at ENA are taken from a combination of three sources: the (1) optical rain gauge (ORG) and (2) present weather detector (PWD), and (3) laser disdrometer measurements (LDIS; *enaldC1.b1* datastream; Wang et al. 2023). The ORG and PWD rain measurements are provided in the MET data product. These surface

rain data are used to exclude times when true cold pools might be affecting temperature and moisture deviations (Sect. 2c), so we take a conservative approach to the existence of surface precipitation by labeling a given minute as raining if any of the three sensors report a positive rain rate. This also permits the inclusion of more overall data, as the three surface rain sensors have extended non-intersecting data gaps.

The time series begins 2016 Jan 01 and continues to provide observations at the time of this manuscript's

preparation, resulting in about 8 years of observations with which to diagnose both background turbulence and precipitation-induced cold pools. After removing times with missing data in any of the temperature, moisture, or merged precipitation time series, we are left with 3,998,870 individual one-minute observations, within which we search for cold pools. While we will not require the presence of surface-reaching precipitation to detect a cold pool, we aid interpretation of cold pool results by constructing "rain objects" from contiguous raining minutes, as determined by the any of the ORG, PWD, or LDIS

sensors reporting any rain rate greater than 0 mm/hr. After filling gaps of less than 5 minutes, we identify 26,241 distinct rain events during the ~8-year period of record.

Throughout this work, we will often refer to variables with an apostrophe to indicate that the quantity is the 11-minute moving mean minus the 121-minute moving mean of that variable. The moving mean is intended to remove spurious noise resulting from very small-scale turbulence, while the subtraction of the 121-minute mean is to remove short-term

manifestations of the diurnal cycle and provide a common ground for temperature and moisture deviations that occur during different weather types. For example, the relative temperature $T'=mean(T_{11})-mean(T_{121})$ and the relative water vapor mixing ratio $q_v'=mean(q_{v11})-mean(q_{v121})$. Changes in these quantities over the course of one minute are denoted as $\Delta T'/\Delta t$ and $\Delta q_v'\Delta t$ and carry the units K/min and g/kg/min, respectively.

For validation, we also utilize the vertically-pointing Doppler lidar (Newsom et al. DLFPT), which we average to 1-

minute resolution to match the meteorological station observations, and Ka-band scanning ARM cloud radar (KaSACR; Kollias et al., 2016), which was implemented for a short period during the ACE-ENA campaign (Wang et al., 2019) and provided plan-position indicator (PPI) and range-height indicator (RHI) scans. This study used the PPI scans at the lowest elevation (0.5°) below the cloud base to show the location and intensity of falling hydrometeors.

## 2.2 Characteristics of ENA rain events

Inherent to this work is the assumption that cold pools form due to negatively buoyant air resulting from evaporation of falling rain. This must also produce an increase in $q_v'$ where evaporation occurs, though that increase in $q_v'$ aloft might not be enough to overcome higher $q_v'$ values near the surface. So cold pools must be associated with a decrease





in T' ($\Delta T'/\Delta t<0$) and a *possible* increase in $q_v$' ($\Delta q_v'/\Delta t>0$), as measured by the surface meteorological station. It is important to note that the existence of cold pools does not require the existence of co-located rain at the surface (cold pool area is greater than rain area), nor does the existence of a cold pool require any rain to reach the surface (Jeong et al., 2023). Conversely, surface-reaching rain may not produce a cold pool at the time the airmass intersects the met station. We therefore avoid requiring surface-detected rain for the retrieval of cold pool properties. We now examine how changes in temperature and moisture are associated with rain and wind at ENA.

Figure 1a reveals the strong association between $\Delta T'/\Delta t<0$ and $\Delta q_v'/\Delta t>0$ with the onset of contiguous rain events. We will target this quadrant when searching for cold pools. However, many of the rain starts are associated with slightly negative $\Delta q_v'/\Delta t$ (especially when $\Delta T'/\Delta t$ is strongly negative), possibly resulting from a more humid surface layer below drier layers in which evaporation occurs. We will account for this possibility in the cold pool detection algorithm. Figure 1b and Figure 1c further reveal the environment when rain is first detected by the surface instrumentation at ENA. When rain is first observed, wind speed and rain rate are most strongly associated with $\Delta T'/\Delta t<0$. While there appears to be an association between wind speed and rain starts (potentially related to cold pool gust fronts), we leave the inclusion of wind speed information for future algorithm refinements.

Figure 1d reveals bivariate histograms of $\Delta T'/\Delta t$ and $\Delta q_v'/\Delta t$ for time periods that are associated with rain (NearRain; surface rain observed within 1 hour before or after) and not associated with rain (FarFromRain; no surface rain observed within 6 hours before or after). We assume that the FarFromRain category does not contain cold pool signatures and is therefore representative of the "background turbulence" experienced at the site, though we note the possibility of an isolated rain shower passing near the met station, thus contaminating the FarFromRain category with larger deviations. In total, there are 703,041 individual FarFromRain minutes, about 17.6% of the total observational record. Temperature and moisture deviations during FarFromRain periods are generally symmetric about the origin and are usually very small at the 1-minute time scale used in this work. In contrast, the temperature deviation extremes for NearRain periods are skewed towards $\Delta T'/\Delta t<0$, indicating a tail in the distribution resulting from evaporation of rain. Note that the NearRain category is expected to contain most of the true cold pools but there is no distinct distribution of $\Delta T'/\Delta t$ and $\Delta q_v'/\Delta t$ that can be clearly delineated from the non-cold pool distribution. Therefore, defining a strict threshold in $\Delta T'/\Delta t$ and/or $\Delta q_v'/\Delta t$ would result in both missed detections of small cold pools and false alarms due to anomalously large background turbulence.

In addition to these metrics that are observable with the surface meteorological instrumentation, we expect that rain events whose first surface-reaching drops fall upon the surface met station have not yet had enough time to develop a cold pool, leaving them unobservable by the surface instrumentation. On the other hand, we expect cold pools to survive longer than the rain falling from an individual cloud. Figure 1 illustrates some of the challenges in detecting cold pools formed by boundary layer precipitation.





**Figure 1:** Displayed are the (a) probability of rain starts, (b) average maximum rain rate just after rain start, and (c) average wind speed preceding rain start. Rain starts are defined as the first index in contiguous rainy objects after rain gaps of less than 5 minutes have been filled. There are a total of 26,241 rainy objects in the ENA time series during this period. Also shown in (d) are bivariate distributions of $\Delta T'/\Delta t$ and $\Delta q_v'/\Delta t$ for NearRain (within 2 hours of observed surface rain) and FarFromRain (more than 6 hours from observed surface rain) periods.

### 2.3 Characterization of bivariate background T' and qv' deviations

At the heart of our cold pool (CP) retrieval algorithm is the requirement for the bivariate deviation of the temperature and moisture to be greater than that of the background turbulence, which occurs throughout the day and year at varying strengths. At ENA, the strength of the background turbulence is often of a similar strength to the cold pools. If a simple threshold based on temperature change is used, as it has been for cold pools from surface measurements at the Barbados Cloud Observatory (Vogel et al., 2021) and airborne data from the VOCALS field campaign (Terai and Wood, 2013), many of the retrieved cold pools are simply signatures of the background turbulence instead of precipitation-driven





cold pools. Figure 2 demonstrates this by translating the Vogel et al. (2021) detection algorithm for the full record and also
only the FarFromRain and NearRain time periods. The fixed $\Delta T/\Delta t$ detections in FarFromRain periods peak during the sunlit
hours (dotted lines in Fig. 2b), when solar radiation leads to increased turbulence in the boundary layer. Throughout the day
(Fig. 2b) and year (Fig. 2a) and regardless of the specific temperature change threshold used, the frequency of retrieved cold
pool objects during FarFromRain periods (therefore likely false alarms) is similar to the values in NearRain periods, which is
when we expect cold pools to be observed most frequently. Therefore, we can confidently say that using only a simple
temperature change threshold results in a diurnal cycle of cold pools that is too similar to that of known false detections
occurring during times when we are confident there are no cold pools. For the -0.08 K/min threshold (as shown in the solid
black lines in Fig. 2), cold pools are estimated to be only about 144 % more frequent in NearRain periods than FarFromRain
periods (3.78 CPs/24hrs and 1.55 CPs/24hrs, respectively). Because this characteristic holds for different 1-minute $\Delta T/\Delta t$
thresholds, we conclude that a simple temperature threshold is unable to adequately distinguish between cold pools and
background turbulence, at least at ENA, where the cold pool signature is relatively weak and often similar to the background
turbulence.

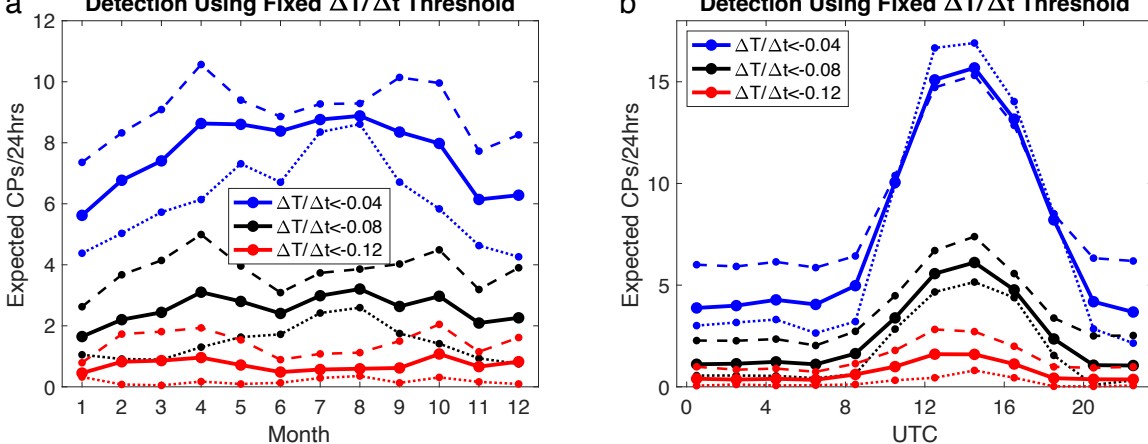

**Figure 2: (a) Annual cycle and (b) diurnal cycle of retrieved cold pools using different choices of fixed temperature-change**
**thresholds (K/min). Solid lines are retrievals from the entire dataset, dotted lines are for the FarFromRain periods and dashed**
**lines are for the NearRain periods. Each value is normalized by the number of corresponding observations to facilitate comparison**
**between the diurnal cycles of the full dataset and FarFromRain background periods.**

Combining the desire for cold pools to be separable from the background turbulence with the requirement that they
be characterized by decreasing temperature at the same time as a preference for increasing moisture, we must first
characterize the background bivariate changes in temperature and moisture for times we are confident *do not* contain cold
pools. Then, when analyzing all data, instances of combined $\Delta T'/\Delta t$ and $\Delta q_v'/\Delta t$ that are more extreme than the values
observed during background periods can be more confidently classified as cold pools.



Here we define the "background" time periods as those that are more than 6 hours from the nearest observed surface rain (Sect. 2a; labeled as FarFromRain in Fig. 1d). We again recognize the undesirable possibility that an isolated rain event may pass near enough to the instrumentation that its cold pool contaminates the background bivariate distribution without its rain being detected by the sensors. This would result in an overly broad background turbulence distribution and, consequently, missed detections of weaker cold pools when searching the full dataset. Our results will therefore represent a

conservative estimate of cold pools passing over ENA. We will see that the background temperature and moisture deviations exhibit strong diurnal and seasonal cycles. With this in mind, we gather the 1-minute $\Delta T'/\Delta t$ and $\Delta q_v'/\Delta t$ FarFromRain values and group them into one-month/two-hour groups. Figure 3a and Figure 3b show the 1st percentile of the $\Delta T'/\Delta t$ and 99th percentile $\Delta q_v'/\Delta t$, respectively. The changes in the diurnal cycle throughout the year are clear, with longer periods of heightened background deviations occurring the local daytime in the boreal summer and early fall seasons. If care were not

taken to remove the seasonal/diurnal dependent background turbulence, many nighttime winter-season cold pools would be missed while false detections would exist during the daytime summer periods.

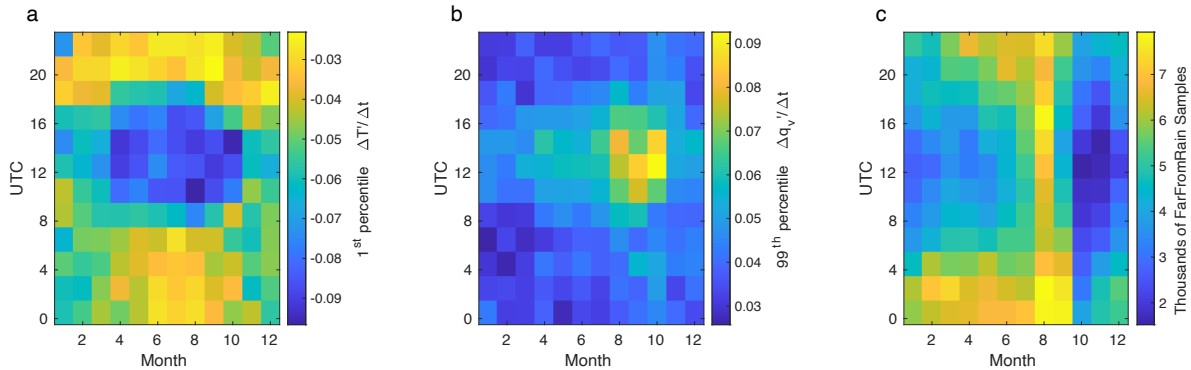

**Figure 3: (a) 1st percentile of 1-minute changes in T'. (b) 99th percentile of 1-minute changes in $q_v$'. (c) Number of samples more**
**than 6 hours from the nearest observed surface rain (FarFromRain).**

        The temporal changes in temperature and moisture we seek to exploit are of course a proxy for the intensity of turbulent eddies in the FarFromRain periods, but we measure temperature and moisture changes *in time* at the met station. One concern is that higher mean wind speeds for the same true eddy size and depth could potentially lead to an

overestimation of eddy strength due to the nature of our deviations being taken as a function of time instead of distance. In such a scenario, it would be necessary to adjust our deviations to be a function of distance instead of time, which would involve division by the wind speed, which would create numerical issues when the wind speed is very low. Fortunately, Supplemental Fig. 2 establishes that the FarFromRain percentiles of temperature and moisture change weakly with wind speed, permitting us to proceed with $\Delta T'/\Delta t$ and $\Delta q_v'/\Delta t$. The Terai and Wood (2013) and Vogel et al. (2021) techniques also

exploit temperature deviations as a function of time rather than distance.



To characterize the background bivariate deviations separately in each month/hour group, we apply singular value decomposition (SVD) to the $\Delta T'/\Delta t$ and $\Delta q_v'/\Delta t$ belonging to that group and the eight neighboring groups. This produces the eigenvalues and eigenvectors that represent the relative magnitude of variability in the two rotated $\Delta T'/\Delta t$ and $\Delta q_v'/\Delta t$ dimensions and rotation of the semimajor axis from the x-axis, respectively. We then define an ellipse that has been rotated

by this value and scaled such that its major and minor axes are proportional to the two eigenvalues. These two properties (ellipse eccentricity and rotation angle) thus describe the background bivariate $\Delta T'/\Delta t$ and $\Delta q_v'/\Delta t$ distribution that is *not* associated with cold pools for a given month/hour group. Figure 3c shows the number of FarFromRain samples (rain not within 6 hours of surface rain) in each month/hour group, revealing that there are few background samples during the daytime in late autumn, which would lead to uncertainty in the characterization of the extremes in bivariate distributions. We

therefore include $\Delta T'/\Delta t$ and $\Delta q_v'/\Delta t$ values from the 8 neighboring month/hour groups, relying on smooth transitions between diurnal/annual regimes that are resolved by the 1-month, 2-hour groups.

Ellipse rotation angles and eccentricities are shown in Fig. 4a and Fig. 4b, respectively. The rotation and correlation patterns present consistent changes in boundary layer turbulence with both diurnal and seasonal cycles. Rotation angles inhabit a wide range between about 0° (elongated along the abscissa) and 70° and tend to be maximized during summer

nights with minima at night and early morning during the winter. Eccentricity is greatest during spring and summer days. Smaller eccentricities represent weaker correlations between temperature and moisture deviations. The diurnal and seasonal changes in rotation angle and eccentricity indicate fundamental changes in background turbulence characteristics as a function of time at ENA.

Figures 4c, 4d, and 4e display normalized bivariate histograms of three month/hour groups, illustrating the overall

positive correlation between $\Delta T'/\Delta t$ and $\Delta q_v'/\Delta t$ in the ENA background turbulence. Note that the distributions of $\Delta T'/\Delta t$ and $\Delta q_v'/\Delta t$ vary not only in magnitude, but also in rotation angle from the x-axis between these examples. Ellipses in these figures are defined for display purposes such that 95% of values fall within the ellipse. The ellipses follow the rotation angle of the background deviations and scale with their magnitude. Figure 4f reveals all the ellipses, illustrating that the background turbulence at ENA is highly variable throughout the day and year.






**Figure 4: (a) Background ellipse rotation angle. (b) Background ellipse eccentricity. (c-e) Background ΔT'/Δt and Δqᵥ'/Δt are shown in grey scale and corresponding ellipse that surrounds 95% of those deviations is overlayed with the color that corresponds to their respective month/hour groups. (f) All background ellipses.**



## 2.4 Retrieval of cold pools as distinct from the background turbulence

With these tools in hand, we now turn to the detection and characterization of cold pools at ENA. Any time point for which $\Delta T'/\Delta t < 0$ and $\Delta q_v'/\Delta t$ is greater than the line extending along the ellipse's major axis is nominated as a cold pool *candidate*, although most candidates will be later rejected due to being indistinguishable from the background turbulence. For each candidate time point, the algorithm seeks the most recent local maximum in T' within the previous hour and the next local minimum in T' in the following hour. The ±1 hour limit imposes a loose time limit on the duration and size of

cold pool candidates and eliminates longer-term anomalous changes that are unlikely to be associated with precipitation. If either the local maximum or local minimum is not found in those respective hours, the cold pool candidate is rejected. Of course, if there are multiple time points that qualify as a cold pool between an individual local maximum in T' and the following local minimum in T', they contribute to the same retrieved cold pool candidate.

        Considering a candidate's corresponding month/hour background ellipse rotation angle and eccentricity, a single

bivariate deviation then corresponds to a unique semimajor axis with that same eccentricity and rotation. Thus, the projection of the bivariate deviations onto the rotated coordinate system via SVD reduces the bivariate $\Delta T'/\Delta t$ and $\Delta q_v'/\Delta t$ problem to a single dimension: the semimajor axis of its ellipse, which can be easily compared for the analogous semimajor axis and eccentricity that we are confident occur due to background turbulence (FarFromRain periods). The semimajor axes have units of either K/min or g/kg/min but we label our figures with K/min for simplicity.

Figure 5a provides a histogram of the largest semimajor axis in each cold pool candidate (hereafter referred to as $a_{max}$). The distribution peaks at a semimajor axis of $a_{max} \approx 0.035$ K/min and sharply decreases as the semimajor axis increases above that range. However, many of the small deviations are likely due to background turbulence and are therefore not associated with cold pools. Note again that a strict threshold in semimajor axis would result in both missed and false detections because the cold pools at ENA tend to be weak and are often indistinguishable from the background turbulence.

Since we assume that the background turbulence for a given month/hour group exists regardless of the existence of cold pools, we may extrapolate the turbulence statistics that belong to the FarFromRain portion of month/hour group into the full time period for that month/hour group. We estimate the expected number of total false alarms ($N_{False}$) as a function of semimajor axis (index *s*) in the entire month/hour group (index *g*) by counting the number of known false alarm candidates occurring during FarFromRain periods, ($N_{FalseFarFromRain}$) and extrapolating to the full time period by dividing by the

fraction of time that group is in the FarFromRain periods. This provides the number of expected false detections as a function of semimajor axis for the entirety of that month/hour group. The distribution of true cold pools is then estimated by examining the differences between this estimated false alarm distribution and the observed distribution of CP candidate strengths with the following procedure.



$$N_{False}(g,s) = \frac{T_{Total}(g)}{T_{FarFromRain}(g)} N_{FalseFarFromRain}(g,s) \tag{1}$$

Iterating through $a_{max}$ bins starting with the strongest candidates, we estimate the number of true cold pools in the strongest bin by subtracting the expected number of false alarms from the total number of candidates, with the result being the deduced number of true cold pools. We then do the same for the 2nd strongest bin but we also remove the number of estimated true cold pools from all larger strength bins (Eq. 2) and so on for successively weaker candidate bins, as depicted in Eq. 2. Here, $N_{True}$ is the estimated number of true CPs, $N_{Cand}$ is the number of CP candidates, $N_{False}$ is the expected number of false alarms with strength greater than the current strength bin $s$. The removal of stronger CPs from the current candidates results in a focus on the stronger candidates that we can be confident are distinct from the background turbulence and a removal of the influence of noise in the weakest candidates without the need for fixed thresholds on CP candidate strength.

$$N_{True}(g,s) = N_{Cand}(g, a > s) - N_{False}(g, a > s) - N_{True}(g, a > s) \tag{2}$$

$$W(g,s) = {N_{true}(g,s)} \Big/ {(N_{cand}(g, a > s) - N_{True}(g, a > s))} \tag{3}$$

The number of CP candidates must be greater or equal to the number of estimated true cold pools in each month/hour/strength bin but we are unable to know *exactly* which individual candidates are true cold pools and which are false alarms. We therefore assign a weight (*W*) to each CP candidate belonging to a month/hour group/strength bin (Eq. 3), determined by the number of candidates, the estimated number of true cold pools, and the number of stronger true cold pools. In effect, this weight is a measure of confidence that a given cold pool candidate is an actual cold pool and not a false alarm. We are more confident that candidate objects with greater semimajor axis are true cold pools because they represent greater deviations from the background turbulence. Conversely, cold pool candidates with a small semimajor axis (e.g. $a < 0.05$ K/min) receive a very small weight, meaning most of them are considered to be false alarms (Fig. 5b). We compute the weighted influence of these very small cold pool candidates to retain the total number and strength distribution of the estimated true cold pools shown in Fig. 5a. Figure 5c confirms the results of Fig. 5a and Fig. 5b by demonstrating that the weak CP candidates do not have an association with the frequency of coincident surface-reaching precipitation, in agreement with the small weight given to these candidates. This weighting technique conserves the total number of estimated true cold pools and their strength distribution separately for each month/hour group. These weights are used in computing the climatology of cold pools presented in the next section.



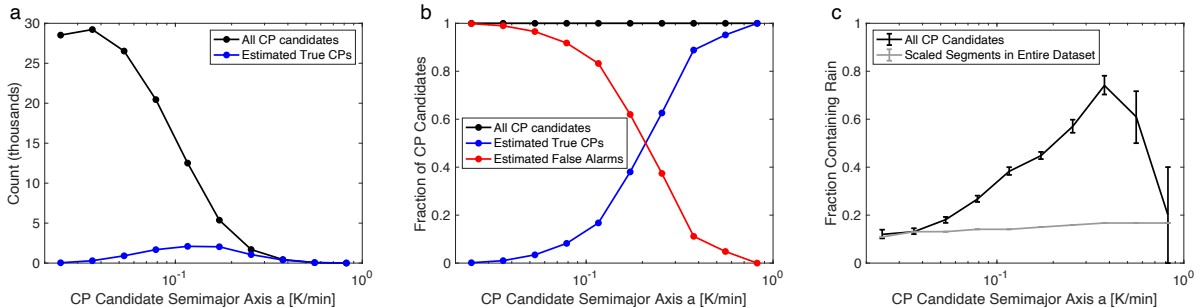

**Figure 5: (a) histogram of the strength of candidate cold pools and true cold pools. (b) estimated fraction of candidates that are false alarms and true cold pools. (c) Fraction of candidates containing surface precipitation (black) compared to the chance of finding precipitation in time series chunks that are of similar sizes to the cold pool objects (gray).**


## 2.5 Algorithm justification via scanning radar and observed surface rain

Figure 6 provides a proof of concept for cold pool identification through a series of relatively heavy rain events occurring within a few hours of each other. In these cases, the T' decreases and $q_v$' increases are timed closely to the observations of surface rain. The strongest rain during this time occurs during 13:00 UTC and the algorithm assigns that

object a weight of 1.0, indicating that there are zero instances of background turbulence stronger than the ellipse created by the $\Delta T'/\Delta t$ and $\Delta q_v'/\Delta t$ during its month/hour group. The KaSACR during another CP candidate (time point A) reveals the horizontal distribution of rain during this cold pool to accompany the observed surface rain. This cold pool, while accompanied by observed rain from both at the surface and KaSACR, is not associated with as strong of a decrease in T' and increase in $q_v$' and the background turbulence is strong enough that this cold pool candidate's weight is only 0.36.

Conversely, there is no observed surface rain accompanying the final cold pool at about 15:15 UTC but it receives a weight of 1.0 due to its steep T' descent and corresponding increase in $q_v$'. In this case, the time points B and C reveal that the rain passed just to the north of the ENA site, while the cooled and humid air in the cold pool had spread far enough to the south to intersect the ENA instrumentation and be identified as a cold pool by the algorithm described above. Wind gusts also tend to accompany the observations of rain, but these gusts are muted when the rain does not pass directly over the

instrumentation, as seen in the final cold pool in Fig. 6. We therefore leave inclusion of wind speed and direction changes to potential future augmentations to this algorithm. At ENA (and presumably in other locations), the cold pool-induced changes in temperature and moisture are embedded within the background turbulence, which can sometimes mask and at other times enhance cold pool signatures.





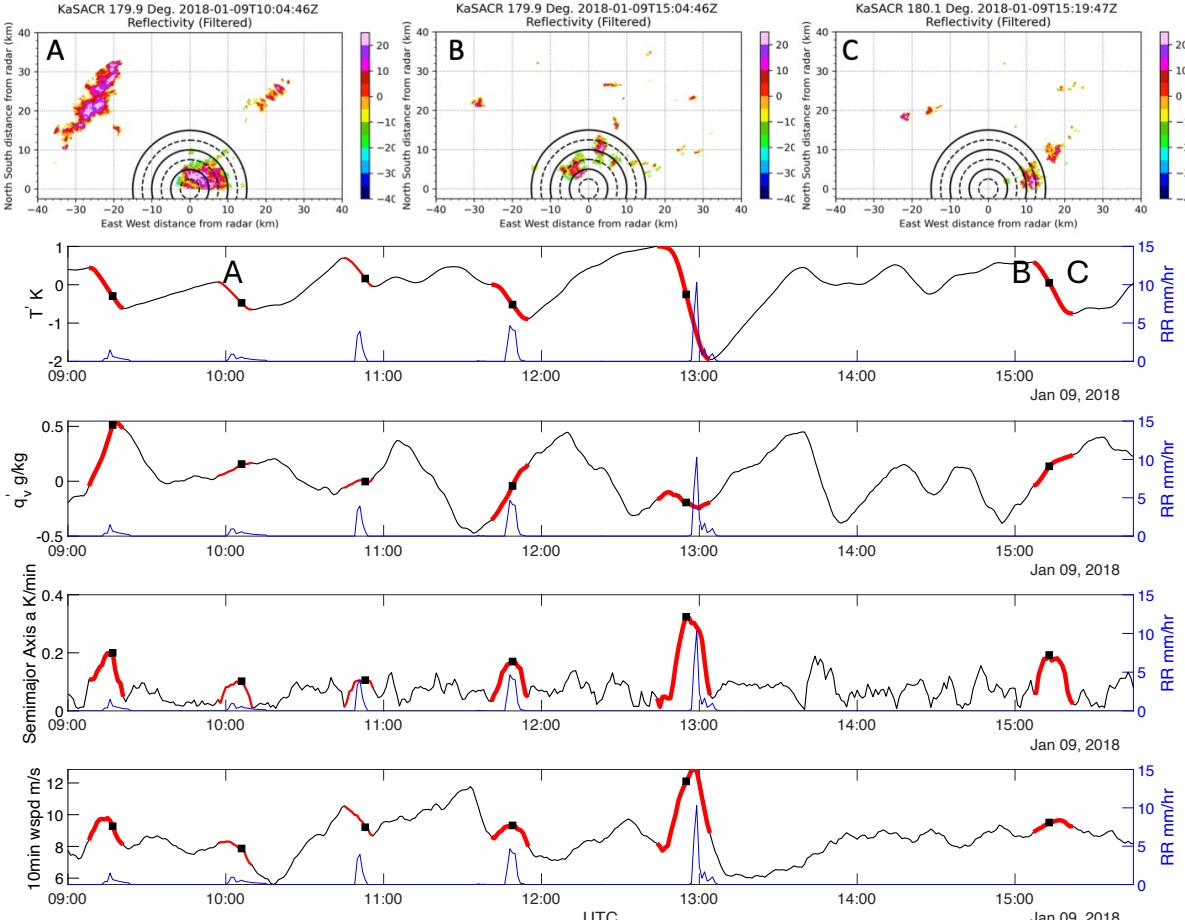

**Figure 6: Bottom rows: Black lines show time series of T', q$_v$', candidate semimajor axis, and wind speed wspd' for a particularly active time period characterized by several cold pool candidates (red). The thicknesses of the red lines correspond to that candidate's weight found via Eq. 2 and Eq. 3. The right y-axes (blue) show the rain rate observed by the ORG/PWD/LD merged precipitation time series. Black dots correspond to the location of maximum semimajor axis in each CP candidate. Top row: Ka-band scanning radar reflectivity for three time points, which are also labelled in the top-most time series.**

To further support the algorithm, we utilize the KaSACR that was in place for the ACE-ENA experiment (Wang et al., 2019) and the surface rain and wind measurements that are a part of the surface metrological station. The scanning radar scanned northward of 287° and 90° (Hunzinger et al., 2020) and was only active for a short period of time and made PPI scans only approximately every 15 minutes. However, not all of these scans (and accompanying ENA observations) can be confidently labeled as containing a cold pool or not because not all rain events cause measureable cold pools, as discussed above. We therefore focus on a limited number of scans when we can be very confident that a cold pool should exist (ConfidentYes) and when a cold pool should not exist (ConfidentNo). The average weights of the cold pool candidates during these scans can be compared to the average overall weights for the time period to identify whether or not the



algorithm is successful in diagnosing these cases. There are a total of 4,026 scans investigated here and the average CP
       candidate weight during this time is 0.077. We expect the ConfidentNo value to be much lower than this average and the
       ConfidentYes value to be much higher.

              We define the ConfidentNo scans as those that have a wind direction from the directions observed by the KaSACR,
       rain fractions less than 1% in all six 2.5 km radial groups between the KaSACR and 15 km range, and no detected surface
rain within ±6 hours of the scan. We must allow for some (albeit small) rain fraction due to random noise in the radar
       backscatter measurements. There are 103 scans that we can be very confident do not contain cold pools. The average cold
       pool weight for these scans is 0.002, indicating a strong rejection of the cold pool candidates during the ConfidentNo scans.

              We define the ConfidentYes scans as those that have surface rain that starts within ±7.5 minutes of the radar scan, a
       maximum observed surface rain rate greater than 1 mm/hr, and minimum surface relative humidity in the preceding 15
minutes less than 75 %. There are 18 ConfidentYes scans, with an average cold pool candidate weight of 0.734, indicating a
       strong preference for diagnosing prominent cold pools as distinct from the background turbulence during the ConfidentYes
       scans.

              The combination of the proof-of-concept time series (Fig. 6) and the desired behavior of the algorithm weights in
       cases we are very confident should and should not contain cold pools provides assurance that the algorithm is accurately
retrieving cold pools distinctly from the background turbulence.

## 3 Cold pools at ENA

              We now examine the 2016-2023 record of cold pools observed at the ENA site. In total, the algorithm identifies
       197,599 cold pool candidates throughout the 8-year analysis period. However, once we account for the time-dependent
       background turbulence, there are only 33,644 candidates with weights greater than zero (Fig. 5a). Those weights sum to a
total of 8,589 cold pool objects that are distinguishable from the background turbulence. For context, there are a total of
       18,569 distinct rain events during this period. So the sum of our cold pool candidate weights is less than half of the number
       of observed surface rain events. On the other hand, only 65% of CP candidates with weight equal to 1 are associated with
       observed surface rain. This section describes the characteristics of those 8,589 retrieved cold pools.

### 3.1 Cold pool properties

We now examine cold pool statistics from the entire population of cold pool candidates. Figure 7 reveals time series
       of temperature, moisture, and wind speed near the surface when cold pools pass over the ENA site. Stronger cold pool
       candidates (larger $a_{max}$) tend to have steeper decreases in temperature and stronger increases in specific humidity and are
       accompanied by gusts of wind. The weaker cold pool candidates have little, if any, wind gusts and could be associated with
       decaying cold pools. The strongest cold pools (defined in this work as those with the largest $a_{max}$) also tend to have the
deepest overall decrease in temperature and increase in moisture as the cold pool passes over ENA.



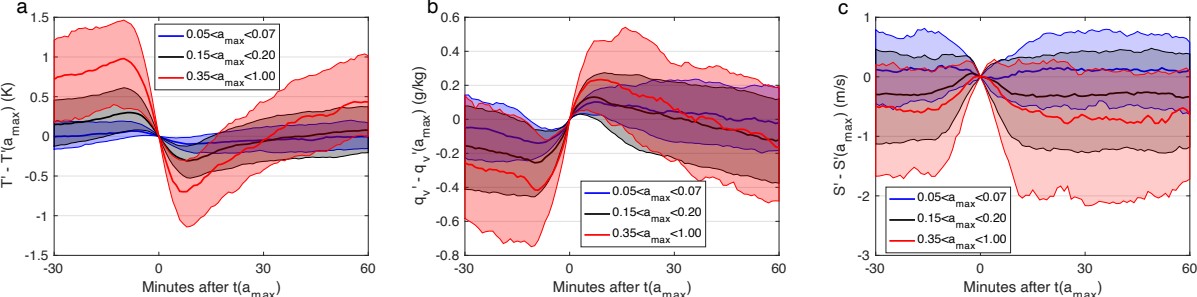

**Figure 7: Time series of (a) T', (b) qᵥ', and (c) horizontal wind speed S' during ENA cold pools, broken into CP strength regimes according to the maximum semimajor axis length (a_max) in each cold pool candidate. Bold lines denote the median and the outer ranges define the 25th percentile and 75th percentile bounds.**

Figure 8 reveals the retrieved size distribution, where CP sizes have been estimated from the cold pool candidate durations with the following arguments. As explained in Sect. 2, we define the start and finish of a given cold pool object as the preceding local maximum T' and following local minimum T', respectively. The along-wind length of the part of the cold pool that passes over the ENA site can be estimated by the elapsed time multiplied by the wind speed. However, because the cold pools are expected to intersect the met station at a random location along the across-wind direction of the cold pool, this simple calculation is likely to underestimate the true cold pool radius. By assuming that (i) cold pools are circular (with the $\Delta T' < 0$ portion forming the leading semicircle), (ii) intersections with the ENA instrumentation are uniformly random, and (iii) the average wind speed during the cold pool temperature decrease represents the cold pool propagation speed, we create a distribution of possible radii ($r$) for a given cold pool from the procedure detailed in Supplementary Fig. 2. In short, the procedure involves computing a distribution of possible radii for each cold pool candidate and adding the possible radii for all cold pools, accounting for each CP candidate's weight.

The weighted mean cold pool radius is found to be 8.29 km (Fig. 8) though most individual objects are likely to have a smaller size. About 10.2% of retrieved cold pools have radii less than 1 km, while about 29.7% have radii greater than 10 km. We note that some of the cold pool objects with the smallest cord lengths are associated with very low wind speeds, leaving our estimates of their cord lengths and true cold pool sizes less certain.



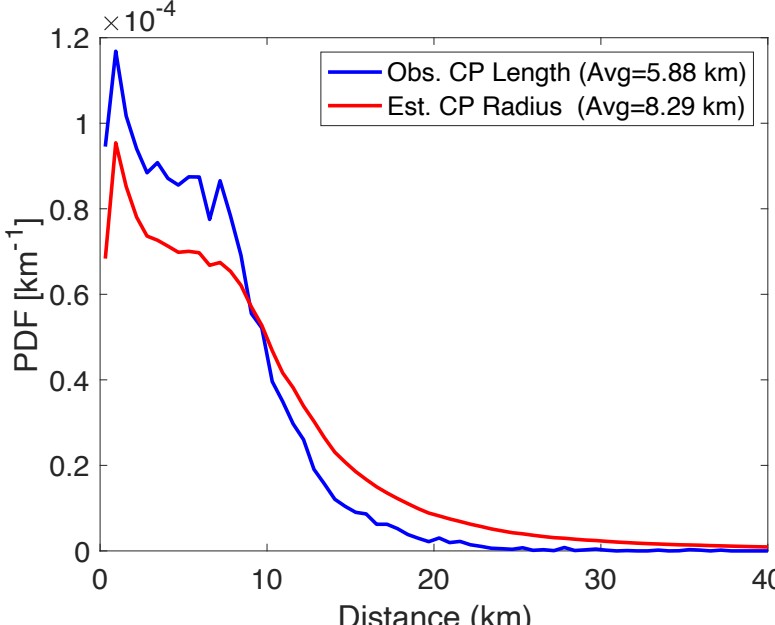

**Figure 8: PDFs of observed cord length (blue) and estimated cold pool radius (red).**


      The association between larger semimajor axis and more cumulus-like scenes is shown in Fig. 9. The discrimination between stratocumulus (Sc) and shallow cumulus (Cu) cloud regimes was introduced in Jeong et al. (2022) from vertically pointing Ka-band radar and Doppler lidar at ENA during a more limited time period of 20160101 to 20170901. As CP intensity increases, the fraction of candidates independently classified as Cu scenes also increases. Contrastingly, the fraction

classified as Sc remains largely indistinguishable from the overall average rate of Sc, with a likely decrease in the prevalence of stronger CPs during the Sc regime.





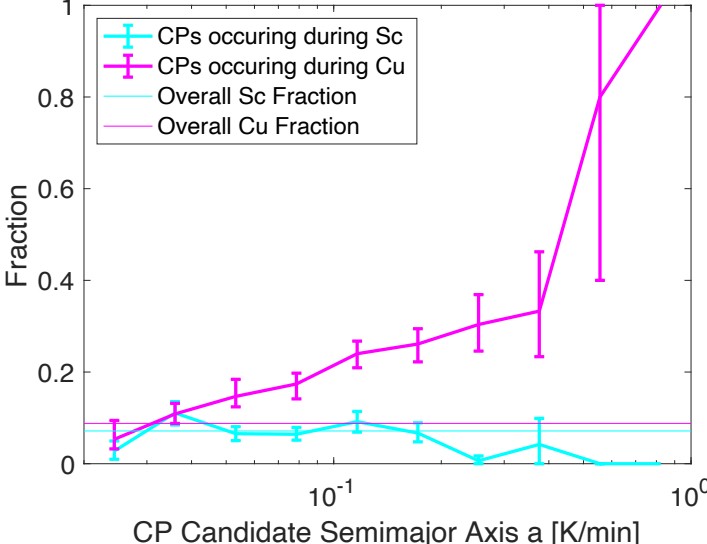

**Figure 9: Fraction of retrieved cold pools that are associated with Sc and Cu cloud types as a function of CP candidate semimajor axis. Background values are computed as the fraction of all 1-minute time points from 20151001 to 20170901 that are classified as either Sc or Cu.**

Figure 10 reveals the association between time series of lower tropospheric vertical motion and $a_{max}$. In the strongest cold pool candidates (large $a_{max}$; Fig. 10c), coherent downward motion exists during and following the strongest portion of the cold pools, representative of precipitation-driven downdrafts from stronger evaporation that then forms stronger cold pools. This downward motion is preceded by coherent upward motion, indicating that these strong cases are likely associated with cumulus precipitation. On the other hand, very weak vertical motions, likely not significantly different from 0 m/s, are present in the weak cold pool candidates (Fig. 10a), suggesting that these may be associated with light rains under well-mixed, stratocumulus-topped boundary layers or with rain events that have dissipated but the weak cold pool signatures remain.





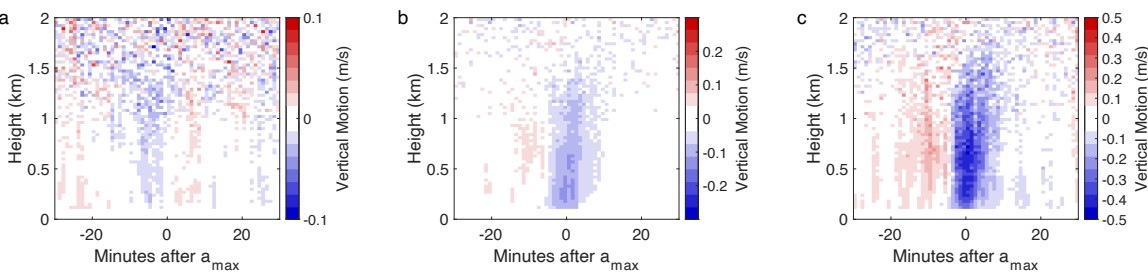

**Figure 10: Average vertical air motions from the zenith-pointing Doppler lidar during weak (a; $0.05<a_{max}<0.07$), medium (b; $0.15<a_{max}<0.20$, and strong (c; $0.35<a_{max}<1.00$ cold pool candidates. Each panel is computed from weighted averages of the**
**Doppler time series curtains belonging to that $a_{max}$ range.**

## 3.2 Cold pool climatology and synoptic setting

The strong annual and weaker diurnal cycles in ENA cold pools are shown in Fig. 11. The annual cycle (Fig. 11a) is more pronounced than the diurnal cycle (Fig. 11b), with cold pools being observed most frequently in the cooler months
when the background turbulence is relatively quiescent. These increased winter cold pools are associated with increased contiguous rain events (Fig. 11d). In contrast to the fixed-$\Delta T/\Delta t$ method and when accounting for seasonal and diurnal changes in background turbulence, we find that cold pools occur 318 % more frequently during NearRain periods than FarFromRain periods.






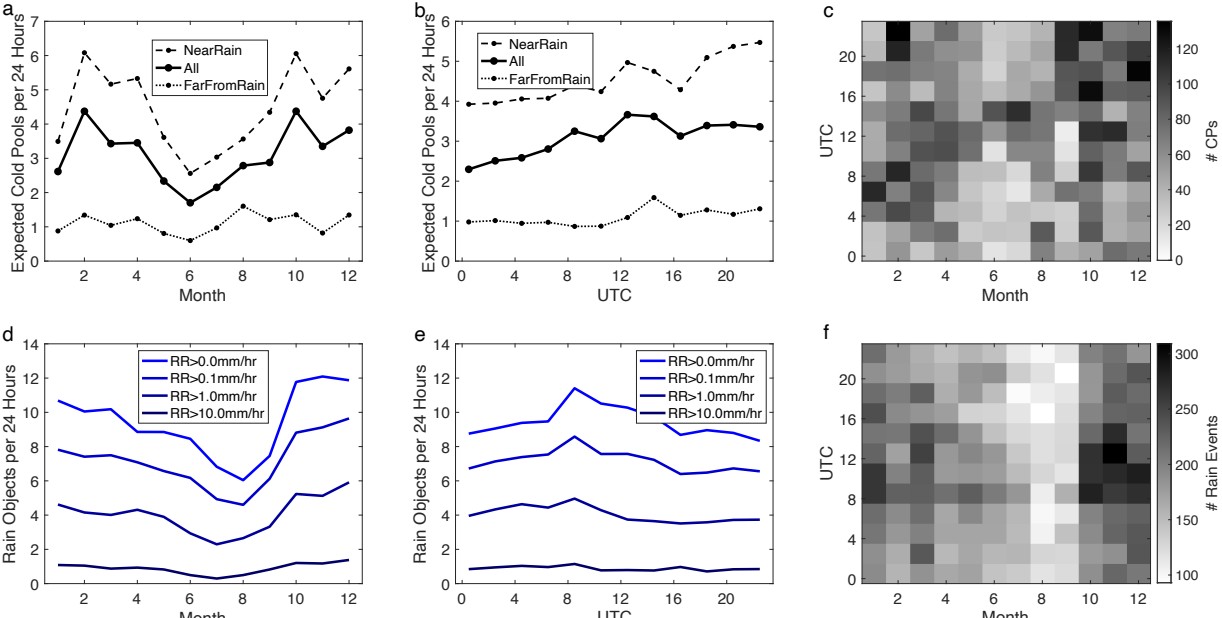

**Figure 11: Rate of retrieved cold pools as a function of (a) month, (b) UTC, and (c) month/UTC. Number of distinct rain events as a function of (d) month, (e) UTC, and (f) month/UTC.**

Figure 12 illustrates the synoptic setting for days with a high and low number of cold pools during cold months (DJF) and in warm months (JJA). In both seasons, days with more than five cold pools tend to be associated with sea level

pressure (SLP) depressions, especially centered to the east of ENA. In contrast, days with few cold pools are characterized by ridging near the Azores islands. Despite the low number of JJA days with more than five cold pools, the SLP deviations appear similar to those in DJF days with more than five cold pools except the SLP deviations from seasonal average need to be greater than during the winter months.





**Figure 12: (a, c, e, g)** Sea level pressure maps for days with high and low daily cold pools in DJF and JJA. **(b, d, f, h)** same as (a, c, e, g) but as a deviation from the DJF and JJA mean SLP maps. In (a), (c), (e), and (g), there are 172, 204, 63, and 259 days contributing to the average SLP maps. In each panel, the purple "x" designates the location of the ENA site at 39° 5′ 29.76″ N, 28° 1′ 32.52″ W.





## 4 Characteristics of Rain Events Leading to Observable Cold Pools

The previous section revealed that the number of cold pool objects at ENA is less than half of the total number of rainy events, a fraction that becomes even smaller when recalling that cold pool objects do not require the co-temporal observation of rain by the ENA instrumentation. So why do most rain events not lead to an observable cold pool at the surface as the rain passes over the ENA instrumentation? This section explores which rainy events result in cold pools that are distinct from the background turbulence.

Figure 13 shows the frequency distribution of mean CP candidate weights as a function of variables relevant to cold pools. As expected, rain events with heavier rain in the beginning of the event are accompanied by more frequent strong cold pool signatures (Fig. 13a), with rain rates greater than about 0.5 mm/hr resulting in a significant elevation of cold pool weight compared to the average of all rain events. Uncertainty increases at the highest rain rates (Fig. 13a), where there are fewer samples (Supplemental Fig. 4a). Frequency distributions of each variable during rain events and regardless of rain are shown in Supplemental Fig. 4.

While surface relative humidity greater than 80% is quite common during rain events (Supplemental Fig. 4b), cold pools are more likely to occur when the surface relative humidity is lower (Fig. 13b). In high-humidity events, less evaporation occurs, leading to diminished cold pool signatures.

We also examined the rate of surface pressure change while rain objects when rain is first observed at the station (Fig. 13c). Cold pool weights are greater than the background average when surface pressure increases over the course of an hour by 100-225 Pa, which we speculate is related to post-frontal scattered showers. However, these events are not frequent (Supplemental Fig. 4c), leading to most retrieved cold pools occurring when the surface pressure is only slightly increasing in time (Fig. 14c).

Boundary layer motions during rain events are also connected to cold pools. Rain events with surface horizontal wind speed greater than 7 m/s are associated with cold pools, while calm conditions are not (Fig. 13d). Because the strongest wind speeds are rare (Supplemental Fig. 4d), most cold pools are associated with wind speeds between 7 – 11 m/s (Fig. 14d). Low-level vertical wind speeds also play a role, with either strongly positive or negative ascent rates associated with higher chances of cold pools (Fig. 13e). These are likely associated with coherent updrafts/downdrafts common in shallow-convective circulations, depending on the exact timing of the rain and the updrafts, downdrafts, and cold pool leading edges (Fig. 10). These stronger low-level upward or downward motions are indeed associated with the majority of retrieved cold pools at ENA (Fig. 14e).

Elevated inversion heights (computed as the 50 m layer with the greatest increase in potential temperature in the lowest 4 km) are also associated with cold pools, especially inversion heights greater than 1900 m. We note that the simple inversion height method is less certain in boundary layers that lack a sharp inversion (i.e. more convective boundary layers). This uncertainty is evident in Fig. 13f for inversion heights greater than ~2800 km, where the mean CP weight is



indistinguishable from the average for all rainy events. Most values are between 1.0 and 2.5 km (Supplemental Fig. 4f), corresponding to the clearest signal in Fig. 13f and the most total cold pools in Fig. 14f.

The boundary layer decoupling index and shear exhibit reduced signals compared to some other variables. Severely decoupled cases (DEC>10 K) tend to have smaller cold pool weights and most rain-associated CPs occur when 2<DEC K<7 (Fig. 13g). Very low values of PBL shear are associated with slightly increased CP weights (Fig. 13h). PBL shear at ENA is usually less than 10 m/s and CPs rarely occur above this value.

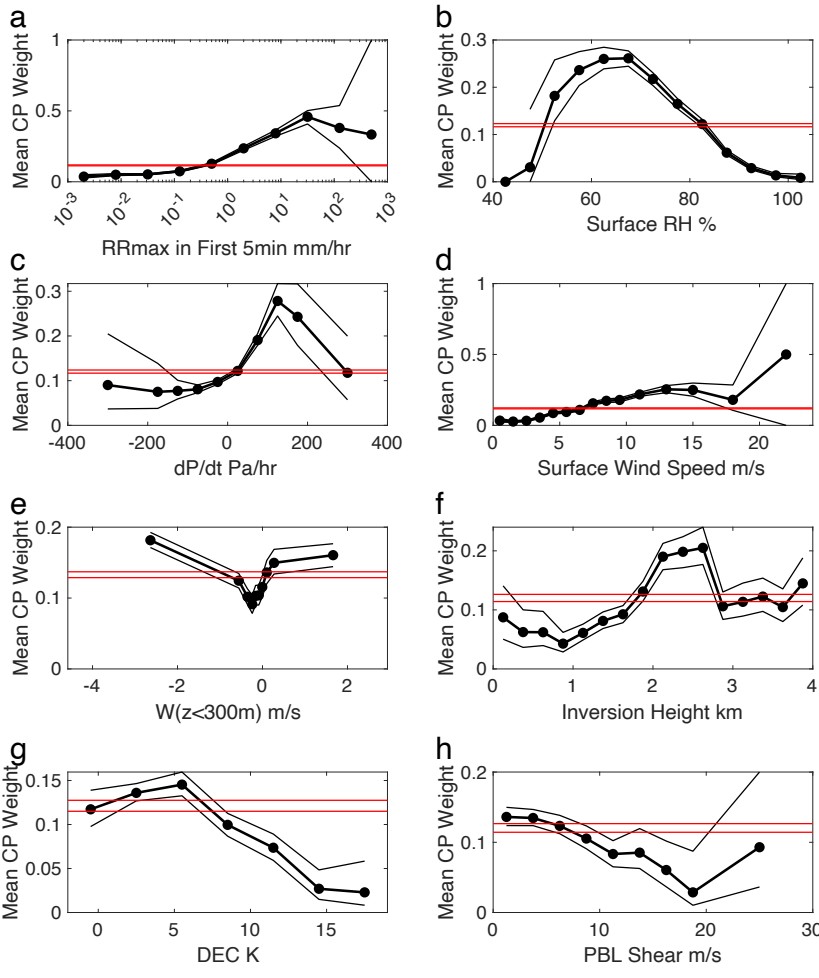

**Figure 13: Black lines show the average CP weight (including zeros) for cold pools that occur during the start of a rain event, as a function of variables obtained from the met station (a, b, d), vertically pointing Doppler lidar (e), and balloon sondes (c, f, g, h).**





**Error bars denote the 95% confidence intervals computed via bootstrap resampling with $10^3$ resamples. Red lines provide the corresponding confidence intervals for the overall overage CP weight during all rain events.**

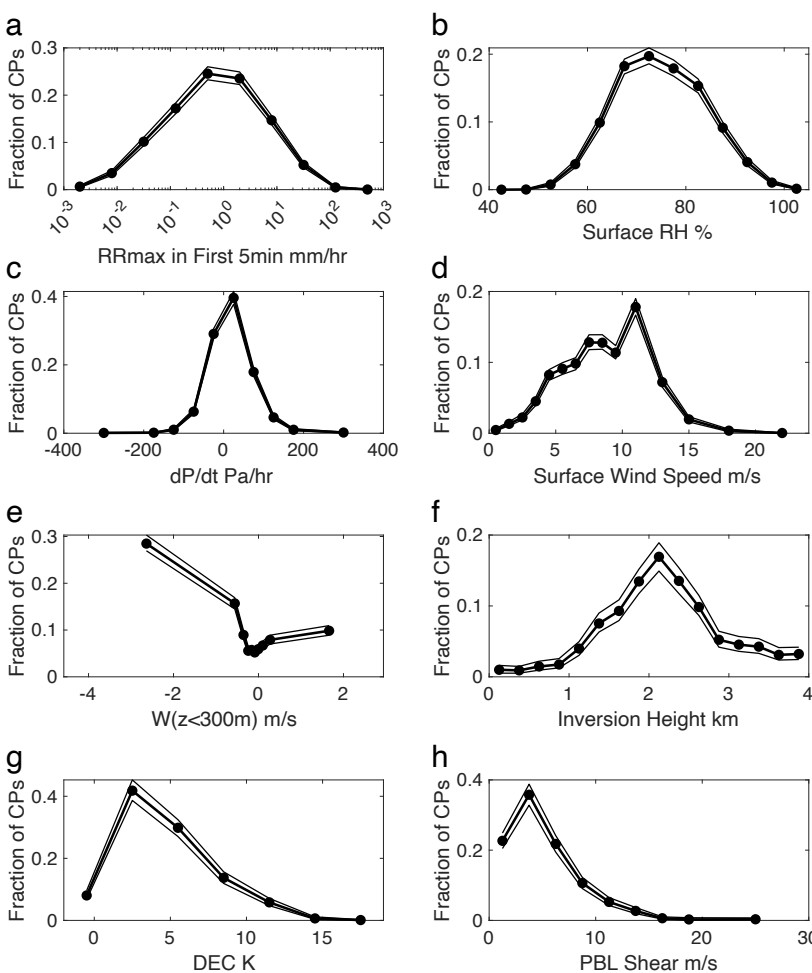

**Figure 14: Like Figure 13, but the sum of CP weights instead of the mean indicating the number of CPs in each bin.**

**5 Conclusions**

This work presented 8 years of cold pool statistics as measured by the common surface instrumentation at

525 the ARM ENA site on Graciosa Island in the Azores. Seeking to avoid a combination of false and missed detections that necessarily result from fixed temperature change methods, we developed a methodology to (1) characterize the bivariate



temperature and moisture background turbulence that exists in the absence of precipitation-induced cold pools by separating observations into diurnal/seasonal groups, (2) reduce the dimensionality of those bivariate deviations to a single dimension represented by the semimajor axis of an ellipse with the same rotation and eccentricity as the background turbulence, and (3) assign a weight to each cold pool candidate according to its strength relative to the frequency that a deviation of equal strength occurs in the background turbulence, thus preserving the estimated number and strength distribution of true cold pools in each month/hour group.

Despite having no "truth" benchmark with which to validate our cold pool detections, we demonstrated expected cold pool behavior related to the fraction of retrieved cold pools that contain surface-reaching precipitation, a time series of surface-reaching rain coupled with scanning radar of passing precipitation systems, vertical motion in the boundary layer and a prior subjectively determined low cloud morphology database, and cases we are confident did or did not contain cold pools during ACE-ENA. We presented statistics on cold pools over eight years, including their number and temporal cycles, their relationship to synoptic weather regimes, and their physical size. We then analyzed the characteristics of the boundary layer associated with the observed cold pools during surface-reaching rain events. Our findings can be summarized as follows:

1. Identifying ENA cold pools using a simple 1-minute temperature change threshold from meteorological station observations results in both missed detections and false alarms due to not accounting for changes in the background turbulence throughout the day and year, regardless of the exact threshold value used. This simple threshold method tends to overcount cold pools (i.e., detects false alarms) during the afternoon hours due to increased boundary layer turbulence while simultaneously missing weaker cold pools that occur during nighttime precipitation, leading to an overly-strong representation of the cold pool diurnal cycle. We also showed that the fixed temperature threshold method retrieves too many cold pools during times more than 6 hours from the nearest observed surface rain (false alarms) compared to times within 1 hour from the nearest observed surface rain.

2. Inclusion of 1-minute changes in moisture enhanced the retrieval, specifically during times when temperature decreases and moisture increases, a reversal from the usual positive correlation between temperature and moisture in background turbulence. Allowing moisture to decrease slightly included cases in which evaporation of rain occurred in dry mid-PBL layers but, upon descent to the surface, were not moist enough to overcome the additional moisture in the lowest few meters of the atmosphere.

3. Assigning weights to each cold pool candidate based on the frequency of similar bivariate deviation intensities occurring in the background turbulence allowed us to retain the total number of true cold pools and their strength distribution while not overcounting the very weak cold pool candidates that are mostly indistinguishable from the background turbulence.

4. Summing the candidate weights, we identify 8,589 cold pools that are distinct from the background turbulence from 2016 to 2023. The seasonal cycle of ENA cold pools peaks in the colder months, with ~4 expected cold pools per 24 hours compared to ~2.5 expected cold pools per 24 hours in the summer months. The cold pool diurnal cycle is less pronounced than the annual cycle and also less pronounced than the diurnal cycle derived with fixed ΔT/Δt thresholds.



Although the frequency of contiguous rain events peaks in the morning and continues to be elevated during the daytime, the temperature and moisture signals of these afternoon cold pools are often obscured by elevated background turbulence. Summer days with more than five cold pools are less frequent than in the winter but are also generally associated with synoptic troughs, especially the post-frontal region.

5.   We identify fewer than half the number of cold pools as the number of rainy objects, as observed by contiguous 1-minute rain rates from surface instrumentation. When rain is first measured at the surface, we find that the following properties result in increased chances of cold pools forming: leading-edge rain rates greater than 0.5 mm/hr, near-surface relative humidity less than 80%, strong1-hour increases in surface pressure, strong horizontal near-surface wind speed, strongly positive or strongly negative vertical motion in the lowest 300 m, and inversion heights between 2 and 2.75 km.

Though we are confident that the methodologies detailed here represent improvements in retrieving cold pool properties from simple near-surface time series observations from a met station, we also note the following unknowns, limitations, and potential future improvements to this methodology:

1.   Figures 6, 7, and 13 demonstrates a connection between cold pools and wind gusts, though wind information is not directly included in our procedure.

2.   Cold pools candidates that are similar in magnitude to the background turbulence are included in the statistics, although they are weighted appropriately for lack of confidence.

3.   While we endeavored to demonstrate expected behavior of the algorithm in Sect. 2, we do not have a "truth" dataset that can provide a full validation of cold pool retrievals.

        Although this algorithm was developed to examine small cold pool signatures resulting from precipitation that is most-often confined to the boundary layer, we note that the algorithm could also be employed at other locations with weak or strong cold pool signatures as-is, without the need for tuned and fixed thresholds, as long as the location has a long enough record to adequately characterize the background turbulence that is not associated with cold pools.

**Code Availability**

Relevant algorithm and analysis code will be made available prior to publication.

**Data Availability**

ARM ENA observations may be downloaded from the ARM data archive at https://www.arm.gov/data/.



## Author Contribution

MAS led this work, from the acquisition of the observations from the ARM data archive, the development and refinement of the cold pool detection algorithm, the figure generation, and the writing of the manuscript. MKW led the proposal that

funded this work and the general direction of this work. MJC provided context regarding future large eddy simulation work building off these results. JHJ created the subjective Cu/Sc classification during previous work and organized the ACE-ENA scanning radar. All co-authors contributed to the discussion of the techniques and results during biweekly meetings and to editing the manuscript.

## Competing Interests

The authors declare that they have no conflict of interest.

## Acknowledgements

This work was supported by the U.S. Department of Energy's Atmospheric System Research, an Office of Science Biological and Environmental Research program, under awards DE-SC0022992 and 89243022SSC000094. The analyses presented here relied upon accessible observations from the ARM Data Archive, for which we are grateful. This work was

performed at the Jet Propulsion Laboratory, California Institute of Technology, under a contract with the National Aeronautics and Space Administration. The authors offer their sincere thanks to ARM director Jim Mather and instrument mentor Jenni Kyrouac for their helpful insights about the surface sensor noise characteristics at ENA.

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
