# Peer review of "A climatology of cold pools distinct from background turbulence at the Eastern North Atlantic observations site"

_EGUsphere, 2024_

## Author Comment (AC1)

The authors thank the reviewers for their helpful comments. Although the observational instruments and general goals remain the same, we have made many changes to the manuscript, including a fundamentally different approach to identifying cold pools using the surface station time series.

**Response to Reviewer #1: https://doi.org/10.5194/egusphere-2024-1098-RC1**

Review of the article titled "A climatology of cold pools distinct from background turbulence at the Eastern North Atlantic observations site" by Smalley and coauthors for publication in Atmospheric Chemistry and Physics. The authors have used data collected at an ARM site to come up with a new technique for identifying cold pools. The technique applies Singular Value Decomposition (SVD) on bivariate distributions of temporal changes in the surface air temperature and moisture from their background values. The technique is then applied to data collected over several years. The main findings are that the cold pools at the site are weak, peak in the winter months, and tough to distinguish from the background turbulence. The article is overall well-written, and easy to follow. The results will be useful for scientists studying cold pools and those using data from the ARM site. However, the article falls short in many ways as mentioned below. Hence, I recommend this article for publication only after the authors have addressed the concerns listed below.

**Major Comments:**

The authors have criticized past studies that used a fixed delta-t and/or delta-r threshold to identify cold pools, especially calling out studies of Terai and Wood (2013), Vogel et al. (2021) Willibanks et al. (2015) and Ghate et al. (2020). Although these studies used a fixed threshold for identifying cold pools, they utilized data from several instruments including satellites, cloud radar, lidars etc. So although they might not have used a sophisticated technique to identify cold pools, by utilizing data from these other instruments they restrict their analysis to raining low cloud conditions only. The authors however have only used data from surface met station and rain gage. As cold pools travel away from the rain shaft, there should be at least some rain in the vicinity of the observed cold pools. In addition, possibly due to the imposition of positive delta-q threshold in the proposed technique, some cold pools associated with weak drizzle might have been lost. So, I suggest you either used data from variety of instruments in your technique, or rephrase the sentences in Line 49-71. Please see Zuidema et al. 2017 for further discussion on this topic.

The authors want to clarify that we are not criticizing those previous works. Each had its own science goals and limitations based on their own instruments and observational environment. We now make a greater effort to communicate in the paper that we are showing evidence that the constant-threshold method is not ideal *at ENA*, which experiences characteristically different weather and climate than Barbados (Vogel et al., 2021) and the Southeast Pacific (Terai and Wood, 2013). The science goals of Ghate et al. (2020) are simply different than our goals. We seek to understand diurnal and seasonal patterns of cold pools and estimate the total expected rate of cold pools at ENA, while Ghate et al. (2020) presented a more focused picture of a smaller number of cold pools.

We now utilize a wider variety of instruments during validation, including satellite-based IMERG precipitation estimates to understand the environment surrounding the precipitation and cold pools to provide context for the new method of detecting cold pools. We continue using the simple 1-minute time series of *in-situ* measurements from the surface observations because we want this technique to be suitable for use in other locations with long-standing temperature and wind speed time series but not necessarily other more expensive instruments that are often deployed for shorter observational periods.

The technique is validated using a single 7-hour case-study and only data from the scanning cloud radar is used. To further solidify the results, I suggest the authors use data from the vertically pointing instruments during few cases and try to understand how things evolved during those. I highly recommend picking up these cases based on the histograms shown in Figure-1 and Figure-4. So few cases spanning from each of the four quadrants would be ideal. Author Jeong's past paper suggests that the team has height resolved rain properties for several years Thank you.

Thank you for this suggestion. The lack of a "truth" or benchmark dataset makes validation quite difficult. In addition to a proof-of-concept time series (Figure 7), we now provide (i) scanning Ka band radar for context in the proof-of-concept time series (Supplemental Figure 2), (ii) the connection of cold pools to observed surface rain fraction and intensity composite time series for four regions of the Depth/Gust metric space (Figure 8), (iii) a new analysis of the ratio of the number of detected cold pools made near observed rain events to the number of detected cold pools during times when no rain is observed ($E_{NR}/E_{FFR}$), and (iv) a comparison of that ratio to the values obtained when using constant temperature change thresholds as implemented in other works.

We note that Terai and Wood (2013) provided only a single time series for proof-of-concept (similar to our Figure 7) as validation. Vogel et al. (2021) similarly provide only time series of the relevant surface observations with added profiles of zenith-pointing Doppler lidar vertical velocity and Ka-band radar reflectivity as validation of their retrieval. Both of those papers were published in ACP.

The main finding of the paper is that most of the cold pools are too weak to be identified from the background turbulence. However, the authors have made very limited attempts on diagnosing the origins of these weak cold pools or the high background turbulence. As it is a marine location, it is expected to have many precipitation induced cold pools. Is the higher number of cold pools during the winter months related to increased precipitation as reported by Wu et al. (2020 J. Climate), Ghate et al. (2021 JAMC), and Lamer et al. (2020 JGR) or they are associated with topography induced drainage flows? Is the increased background turbulence in summer months related to the island heating as reported by Ghate et al. (2021 JAMC). In addition, the winter months also encounter higher number of frontal passages as shown in Figure 12 that is similar to the findings of Ilotoviz et al. (2021 JGR). It seems that your technique only identifies cold pools associated with heavy rain only and hence the finding.

The authors disagree with the statement that "most of the cold pools are too weak to be identified from the background turbulence". The reviewer would be correct to say that most cold pool *candidates* are too small to be distinguishable from background variability. We have tried to be more diligent about distinguishing between candidates and detected cold pools in the new

manuscript. The goals of this work can be summarized with the following list: (i) introduce a new methodology that leverages a more objective method of detecting cold pools and avoids subjective constant thresholds or manually-selected cases, (ii) present the seasonal and diurnal cycles, sizes, and other properties of those detected cold pools, and (iii) present an analysis of which rain events lead to detected cold pools. The main body of the paper (Sections 1-6) already contain more than 12 thousand words and 15 figures, 13 of which have multiple panels. Adding additional analyses regarding the specific sources of cold pools beyond analyses already present in Figures 14 and 15 would both dilute the current results and make the manuscript quite long for the reader. We have added the Wu, Ghate, and Lamer references to the manuscript but leave those investigations to other work.

Lastly, I understand that the authors have proposed a new technique for identifying cold pools. But for this technique to be applicable to other studies, and used by other researchers, the authors should make an attempt to put their results in the context of previous work. So for example, if one would have used a fixed delta-T threshold for identifying cold pools, would they have also been able to produce Figures similar to 7 to 11. Or how would the Figure 7-11 would have looked if the cold pools would have been identified from a fixed delta-T threshold.
Thank you for this suggestion. We now first show the results obtained when using a constant threshold (Figure 1) both in terms of the expected rate of cold pools at ENA and their seasonal/diurnal cycles.

**Minor Comments:**
Line 13: DOE is an acronym that needs to be defined, especially as ACP is an international journal. Thank you.
Done

Line 33: Jiang et al. 2021 is not in the references. Please define.
It is now added to the references list, thank you.

Line 428: probably better to use the phrase boundary layer rather than lower tropospheric.
The wording has been changed according to your suggestion.

Line 685: I wonder why the authors chose to cite the campaign report rather than the BAMS article. I suggest citing the BAMS article as it is easy to find and more relevant. Thank you.
References to the campaign report have been changed to the BAMS paper.

**A climatology of cold pools distinct from background thermodynamic variability at the Eastern North Atlantic observations site**

Mark A. Smalley[1,2], Mikael K. Witte[1,2,3], Jong-Hoon Jeong[1], Maria J. Chinita[1,2]

[1]Joint Institute for Regional Earth System Science and Engineering, University of California, Los Angeles, Los Angeles, California
[2]Jet Propulsion Laboratory, California Institute of Technology, Pasadena, California
[3]Naval Postgraduate School, Monterey, California

*Correspondence to*: Mark A. Smalley (mark.a.smalley@jpl.nasa.gov)

© 2025. California Institute of Technology. Government sponsorship acknowledged.

**Abstract**

We identify cold pools at the US Department of Energy (DOE) Atmospheric Radiation Measurement (ARM) Eastern North Atlantic (ENA) facility on Graciosa Island in the Azores and examine the statistics of retrieved cold pools from 2016 to 2024. The retrieval leverages 1-minute deviations in near-surface temperature and wind speed from the ENA surface meteorological station time series to identify cold pool events that exceed the association between background thermodynamic variability and observed distinct precipitation events. Cold pools at ENA exhibit a prominent annual cycle, peaking in the winter months. Although there is a slight increase in rain events during the daytime, we find a decrease in daytime cold pools that are separable from background variability compared to nighttime because of the increased background variability during sunlit hours. Often, surface-reaching rain events are *not* associated with cold pools due to factors including but not limited to high background thermodynamic variability, reduced surface wind speed, high boundary layer humidity, fully overcast skies, and weak rain rate. Understanding the factors that lead to the formation of measurable cold pools will lead to a greater understanding of the dynamics of the marine boundary layer and their influence on cloud morphological structures.

**Short Summary**

Evaporation of falling rain leads to temporarily cooler and sometimes windier surface conditions (cold pools), which can lead to further convection that alters convective, cloud, precipitation, and radiation properties. We introduce a new method of measuring cold pools from simple surface-based measurements of temperature and wind speed and then then apply it to 9 years of surface station observations in the north Atlantic Ocean. Cold pools at ENA exhibit a prominent annual cycle, peaking in the winter months. Often, surface-reaching rain events are *not* associated with cold pools due high background thermodynamic variability, reduced surface wind speed, high boundary layer humidity, fully overcast skies, and weak rain rate.
* * *
**1 Introduction**

State of the art earth system models continue to struggle to simulate the geometrically thin but highly reflective stratocumulus (Sc) clouds that frequently cover the eastern portion of subtropical oceans (Jiang et al., 2021), with much of the uncertainty in the magnitude and sign of global cloud feedbacks being traced to the representation of Sc clouds (Klein et al., 2017; Scott et al., 2020; Myers et al., 2021; Ceppi and Nowack, 2021). The transition of boundary layer clouds from fully overcast stratocumulus to lower cloud fraction shallow cumulus (Cu) following the trade winds results in a strong increase in the amount of radiation absorbed by the ocean surface (Goren and Rosenfeld, 2014) and a reduction of solar radiation reflected (Hartmann and Short 1980; Wood 2012). The breakup of these expansive stratocumulus clouds is largely determined by meteorology and sea surface temperature (Bretherton and Wyant, 1997; Wyant et al., 1997), but recent works (Eastman and Wood, 2016; Yamaguchi et al., 2017; Goren et al., 2019; Blossey et al., 2021; Smalley et al., 2021) have emphasized the role of precipitation in modulating the timing of the cloud regime transition. It is therefore imperative that we understand the physical connections between precipitation and marine boundary layer cloud regime change.

Falling precipitation begins to evaporate as it encounters subsaturated air below the cloud layer. This evaporation cools the air, reducing its buoyancy and leading to downdrafts. Once these downdrafts reach the surface, they spread horizontally and form what is defined as a cold pool (CP) or density current (Wilbanks et al., 2015). Compared to the surrounding near-surface air, CPs are characterized by decreases in temperature and increases in wind speed. Horizontally propagating CPs may induce upward motion by displacing more buoyant air or by colliding with other outward-propagating CPs, mechanically forcing air upwards and potentially leading to further convection that can destabilize the cloud layer and lead to stronger rain rates, reduced cloud fraction, and lower scene reflectivity (Feingold et al. 2010).

While much of the existing literature on CPs has been in the context of impacts on deep convection (Engerer et al., 2008; Feng et al., 2015; de Szoeke et al., 2017), here we observe CPs passing over the Azores archipelago, which is a maritime environment frequented by boundary layer clouds and their precipitation in addition to occasional deeper and organized systems (Giangrande et al., 2019). These lighter rains, largely warm rain from shallow Cu and Sc, are expected to produce weak temperature and wind signals. Several previous studies have examined the CPs that form when rain falls primarily within the boundary layer during shallow convection. Terai and Wood (2013) leveraged time series of potential temperature from sub-cloud aircraft measurements over the subtropical southeastern Pacific Ocean to retrieve CP signatures. They found that small decreases in potential temperature (stronger than -0.36 K) were associated with weak gust fronts and increases in the concentrations of both coarse mode aerosols and dimethyl sulfide, which is relevant for secondary particle formation in the atmosphere. Wilbanks et al. (2015) used measurements of air density instead of potential temperature and similarly found that temperature decreases were strongly associated with their retrieved density currents; their constant threshold for changes in air density roughly corresponds to a total change in temperature of -0.24 K throughout the CP duration. In manually selected cases of precipitating post-cold frontal marine stratocumulus clouds, Ghate et al. (2020) used vertically pointing Doppler radar and lidars to analyze 76 drizzle shafts. They found that downdraft strength correlated with cloud-base drizzle intensity. Vogel et al. (2021) analyzed near-surface temperature variations in

Barbados, identifying CPs based on a fixed temperature change threshold of -0.05 K/min. They demonstrated associations between precipitation duration, retrieved CP strength, and cloud regime categorizations.

In this work we seek to estimate the total number of CPs and their diurnal and seasonal patterns. To do so, we must distinguish between thermodynamic variability caused by CPs and thermodynamic variability caused by other phenomena. Though Terai and Wood (2013), Wilbanks et al. (2015), and Vogel et al. (2021) removed the influence of small-scale variability that is not associated with precipitation-driven CPs by time-averaging surface meteorological station observations and requiring perturbations to exceed constant thresholds, ENA boundary layer variability spans a spectrum of spatial/temporal scales and varies strongly with time of day and season. Using a constant threshold at ENA can therefore lead to false CP detections during periods of enhanced variability, particularly during daytime hours when solar heating drives stronger mixing. Cold pool signatures also exist with a spectrum of intensities, which leaves methods that employ constant thresholds both unable to capture the weakest CPs and at the same time unable to remove all the signatures of the background thermodynamic variability, which we define as any naturally occurring fluctuations in thermodynamics and winds that are *not* due to CP activity. In addition, the background thermodynamic variability can be expected to exhibit annual and diurnal cycles, which are themselves variable with geographic location. We therefore seek a method to retrieve CPs with surface instrumentation in the presence of, but separable from, the background thermodynamic variability in a way that is flexible to diurnal and annual cycles.

Here, we present a method to measure CPs from surface-based observation sites with an emphasis on removing the signals of the background thermodynamic variability that are *not* associated with precipitation-driven CPs. While existing methods detect CPs by simply requiring the near-surface temperature to decrease by more than a subjectively prescribed amount over the source of a single minute, the new method expands upon that approach by (1) using three temperature and wind metrics to characterize CP candidates, (2) assigning thresholds for each metric based on statistical association between the frequency of distinct rain events and increasing magnitudes of the three metrics, and (3) allowing those thresholds to vary throughout the annual and diurnal cycles, thus accounting for time-varying background thermodynamic variability that is not associated with CPs. We will show the results to this method when applied to 9 years of surface observations from the Azores archipelago, which experiences diverse sub-tropical and mid-latitude weather conditions (Giangrande et al., 2019), as well as this method's advantages over method that employ a single constant threshold on a single metric.

We first describe the relevant observations in Section 2. In Section 3 we introduce and validate the algorithm using scanning radar, surface rain observations, and satellite precipitation observations. We then apply the algorithm to 9 years of surface station observations representing a short climatology in Section 4. Section 5 provides an analysis of which confirmed contiguous rain events lead to CPs, and Section 6 provides a discussion of our findings.

**2 Observations from the ARM ENA site**

The goal of this work is to confidently diagnose cold pools (CPs) via one-dimensional temperature and wind speed measurements from surface-based *in-situ* observations. We select the United States Department of Energy (DOE) Atmospheric Radiation Measurement (ARM) program's Eastern North Atlantic (ENA; 39° 5′ 29.76″ N, 28° 1′ 32.52″ W) observatory on

Graciosa Island in the Azores, which is situated at the northern edge of a subtropical marine weather regime but is also impacted by mid-latitude synoptic systems (Remillard and Tselioudis, 2015; Mechem et al., 2018; Giangrande et al., 2019). The site is situated in the northwestern portion of the island, about 30 m above sea level and about 400 m to the south of the nearest coastline, which is composed of rocky cliffs. The surface near the site is relatively flat and containing fields, residential areas, and the Aeródromo da Graciosa airstrip. The Graciosa Island time zone is UTC-1.

The ENA site contains a comprehensive suite of surface-based instrumentation, but this work mainly utilizes observations of near-surface temperature and wind speed from the surface meteorological station (MET; *enametC1.b1* datastream; Kyrouac and Shi 2011) and the Meteorological Automated Weather Station (MAWS; *enamawsC1.b1* datastream; Keeler et al., 2017) to retrieve CPs. The temperature and wind speed time series are derived from a serialized combination of the MET and MAWS systems due to elevated noise in the MET temperature deviations during extended time periods. Preceding about 2015 Oct 01 and between about 2017 May 01 and 2024 June 01, the minute-to-minute variations in the 1-minute MET temperature data (shown in Supplemental Figure 1) exhibit a considerable increase in variability that we believe is unlikely to be related to natural phenomena and likely represents anomalous instrument noise. This noise is too large to be useful in diagnosing weak changes due to CPs from light rain and virga. Therefore, in this work we utilize the 1-minute temperatures and wind speeds from MET between 2016 Jan 01 to 2019 April 01 and then MAWS from 2019 April 01 until 2024 Dec 31.

We note that the MAWS temperatures are provided at 0.1 K intervals, considerably greater than the MET temperature intervals of 0.01 K. Such coarse MAWS intervals is large compared to some of the CP signatures examined here. Similar to other previous works (Vogel et al., 2021), we perform temporal averaging to many of the surface station time series observations (explained later). This averaging reduces the influence of instrument noise while simultaneously eliminating small-scale variability that is unlikely to be related to CP signatures. As an additional benefit to our work with the MAWS temperature data, this smoothing allows the MAWS data to take values at finer intervals than the original 0.1 K intervals. The general conclusions of this work are not affected by switching to MAWS on 2017 May 01, indicating that the 0.1 K MAWS intervals does not strongly affect the CP detections.

Observations of surface precipitation at ENA are taken from a combination of three sources: the (1) optical rain gauge (ORG), (2) present weather detector (PWD), and (3) laser disdrometer measurements (LDIS; *enaldC1.b1* datastream; Wang et al., 2023). The ORG and PWD rain measurements are included in the MET data product referenced above. A given minute is labeled as raining if any of the three sensors report a positive rain rate. The three sensors have extended non-intersecting data gaps, supporting their combined usage during the entire 2016-2024 period.

The time series begins 2016 Jan 01 and continues to provide observations at the time of this manuscript's preparation, resulting in 9 years of observations from which to diagnose both background thermodynamic variability and precipitation-induced cold pools. The dataset holds a total of 4,643,927 individual valid one-minute observations (approximately 98% of the 9-year data record), within which we search for CPs. While we do not directly require the presence of surface-reaching precipitation to detect a specific CP, we construct the algorithm and aid interpretation of CP results by constructing "distinct rain events" from contiguous raining minutes, as determined by the any of the ORG, PWD, or LDIS sensors reporting any rain rate

greater than 0 mm/hr. We identify 29,269 distinct rain events during the 9-year period of record, where rain detections separated by less than 5 minutes were merged into single events to account for the high spatial and temporal variability of surface reaching precipitation.

Here we assume that the annual and diurnal cycles of CP frequency should be correlated with the analogous cycles of distinct rain events. Figure 1a and Figure 1b show these cycles for different observed maximum rain rates during each distinct rain event. The annual cycle peaks between October-March, with reduced frequency of rain events during the boreal summer. Similar results for ENA were found by Lamer et al. (2020), Wu et al. (2020), and Ghate et al. (2021). The diurnal cycle has a less-pronounced peak from about 0800 UTC to 1500 UTC in the late morning and early afternoon. While not all rain events lead to CPs, we expect these general annual and diurnal cycles to be represented in the CP statistics, with small discrepancies based on the separability of CP signatures from non-CP sources of background variability, which varies throughout the year and day.

Throughout this work, we refer to time series variables with an apostrophe to indicate that the quantity is anomaly of the 11-minute moving mean minus the 61-minute moving mean of that variable, where the means are computed symmetrically around each minute. For example, the temperature anomaly $T' = T_{11}-T_{61}$, where $T_{11}$ and $T_{61}$ are the 11-minute and 61-minute means, respectively. Similarly, we define the wind speed anomaly $S' = S_{11}-S_{61}$. The 11-minute moving mean is intended to remove spurious signals resulting from very small-scale variability and instrument noise, while the subtraction of the 61-minute mean is to remove short-term manifestations of the diurnal cycle and provide a common ground for larger-scale changes in temperature and wind speed anomalies that occur during different weather regimes (i.e., frontal passage). Changes in temperature anomaly over the course of one minute are denoted as $\Delta T'/\Delta t$ and carry the units K/min.

We also utilize the vertically-pointing Doppler lidar (Newsom et al. DLFPT), which provides retrievals of vertical air motion at about 1 s/30 m spacings, and balloon sondes (Keeler et al., SONDEWNPN). To match the MET and MAWS time points, we average the DLFPT vertical motions to 1-minute/30 m spacings. Sonde observations of temperature and specific humidity are averaged to 10 m vertical spacings. For validation, we examine reflectivity factors from the Ka-band scanning ARM cloud radar (KaSACR; Kollias et al., 2016), which was implemented during the ACE-ENA campaign (Wang et al., 2022) and provided plan-position indicator (PPI) and range-height indicator (RHI) scans. Here we use the PPI scans (between azimuthal angles of 287° and 90°) at the lowest elevation (0.5°) below the cloud base to show the location and intensity of falling precipitation.

To support the algorithm, we utilize a combination of the three surface precipitation estimates described above (ORG, PWD, and LD), in conjunction with precipitation retrievals from the 0.1° gridded half-hourly IMERG Final (Huffman et al., 2020; Huffman et al., 2023), which gathers precipitation observations primarily from microwave radiometers but is supplemented/interpolated with infrared when no microwave observations are available. We use only the IMERG Final values that are within ~30 km of the ENA site, a plausible range of influence for precipitation and CPs. We focus on times when rain is observed either at or near the surface instrumentation (NearRain) versus times when we can be very confident that a CP *should not* exist (FarFromRain). For NearRain times, rain is observed by the surface instrumentation within ±1 hour, indicating rain in the immediate vicinity that could lead to CP signatures. For FarFromRain times, zero surface rain must be observed within ±6 hours by the ORG, PWD, and LD and no rain can be reported by IMERG Final within ±2 hours. We assume that the

FarFromRain category should not contain CP signatures. However, we acknowledge this assumption's limitations, as propagating CPs from terminated rain events plausibly travel distances greater than 30 km with the mean boundary layer flow and shallow and isolated light rain near the surface instrumentation may go undetected by IMERG Final. Both possibilities would contaminate the FarFromRain category with larger temperature changes and gusts. In total, there are 286,766 individual FarFromRain minutes (6.2 % of the valid record) and 995,134 NearRain minutes (21.4 % of the valid record).

**3 Cold Pool Detection Algorithms**

**3.1 Cold pool detection using a constant temperature change threshold**

Previous works have utilized changes in temperature T alone to diagnose the existence of CPs from surface measurements at the Barbados Cloud Observatory (Vogel et al., 2021) and airborne data from the VOCALS field campaign (Terai and Wood, 2013). But are these constant-threshold methods effective at ENA? Figure 1 demonstrates considerable shortcomings of the constant-$\Delta T/\Delta t$ method when implemented at ENA by reproducing the Vogel et al. (2021) algorithm for a variety of different constant-$\Delta T/\Delta t$ thresholds. In Figures 1c and 1d, retrieved CP frequencies are plotted against month and UTC for comparison against Figures 1a and 1b. The disparities in the annual and diurnal cycles between CPs obtained using constant thresholds and observed rain event frequencies are evident, especially when the threshold $\Delta T/\Delta t > -0.10$ K/min (Figure 1c and 1d). These detections peak strongly during the sunlit hours (Figure 1c), when solar radiation leads to increased variability in the boundary layer. For thresholds stricter than -0.10 K/min, the retrieved annual cycle of CPs (Figure 1d) is also out of phase with the observed annual cycle of rain events (Figure 1a). For stricter thresholds, there are too few retrieved CPs to suggest that all CPs are detected (0.93 CPs/24 hours when and 0.47 CPs/24 hours for the -0.11 K/min and -0.14 K/min thresholds, respectively) when comparing to the number of observed rain events. When detecting CPs at ENA using a constant threshold for 1-minute changes in temperature, the resulting annual and diurnal cycles of CPs are reflections of the annual and diurnal cycles of the background thermodynamic variability instead of the desired precipitation-driven CPs.

Finally, Figures 1e and 1f demonstrate the method's sensitivity to the subjective choice of threshold value. Figure 1e illustrates that the estimated total number of detected CPs is strongly and smoothly sensitive to the constant $\Delta T/\Delta t$ threshold, with no obvious choice of the best value. We also examined the ratio of the expected rates of NearRain CPs to FarFromRain CPs, represented in Figure 1f as $E_{NR}/E_{FFR}$. $E_{NR}$ is computed as the number of retrieved NearRain CPs per 24 hours of valid NearRain times and *vice versa* for $E_{FFR}$. We prefer the ratio $E_{NR}/E_{FFR}$ to be high, analogous to a combination of high hit rates and low false alarm rates. The $E_{NR}/E_{FFR}$ ratio increases steeply for thresholds stricter than about -0.1 K/min, indicating there is little non-CP thermodynamic variability stronger than about -0.1 K/min at ENA. However, CPs using these strict thresholds still do not adequately capture the diurnal cycle (Figure 1d) and likely report too few CPs compared to distinct rain events, as described above.

It is clear from Figures 1e and 1f that there is no single ideal constant-$\Delta T/\Delta t$ threshold and that subjectively assigning any reasonable constant threshold will result in both missed detections of weak CPs especially in the winter nighttime *and* false detections of heightened

background thermodynamic variability especially during the summer daytime. Due to the shortcomings of constant thresholds shown in Fig. 1, we conclude that use of a constant temperature change threshold is unable to adequately distinguish between CPs and background thermodynamic variability, at least at ENA where the CP signature is relatively weak and often of similar strength to the background variability. We therefore seek additional information with which to confidently discriminate between precipitation-driven CPs and background thermodynamic variability. We note that use of a constant threshold may be appropriate when applied to identification of CPs in stronger convection over land (Redl et al, 2015; Provod et al, 2016; Kirsch et al, 2021; Kruse et al, 2022) and oceanic deep convection (e.g., de Szoeke et al. 2017), when drops in temperature exceed those typically observed at ENA.

[Figure]

Figure 1: (a) Annual and (b) diurnal cycles in the frequency of distinct surface-reaching rain events measured by ENA instrumentation. (c) Annual and (d) diurnal cycles of retrieved CPs using 4 different choices of constant $\Delta T/\Delta t$ thresholds. (e) The number of retrieved CPs as a function of constant $\Delta T/\Delta t$ threshold. (f) The ratio of the expected rates of CPs during NearRain times to the expected rates of CPs during FarFromRain times, $E_{NR}/E_{FFR}$ as defined in the text, as a function of constant $\Delta T/\Delta t$ threshold.

**3.2 Cold pool observational metrics from *in-situ* surface instrumentation**

[Figure]

**Figure 2: Temporal evolution of (a) T', (b) q$_v$', and (c) S' around the start of ENA rain events. Results are split into three ranges of the maximum surface rain rate within the first 5 minutes of the rain event. Panels (b), (c), and (d), show the mean T, q$_v$, and relative humidity RH for combinations of summer/winter and day/night launches.**

Inherent to this work is the assumption that CPs form due to negatively buoyant air resulting from evaporation of falling rain. This is demonstrated in Figure 2, which illustrates the temporal evolution of T', S', and q$_v$' relative to the observed onset of surface rain in panels a, c, and e, respectively. The drop in average T' is clear when the maximum rain rate in the first five minutes (RRmax) is greater than 1.0 mm/hr but the T' decreases are not as obvious for weaker rain rates. The sub-cloud rain evaporation must also produce an increase in the relative water vapor mixing ratio q$_v$' *where evaporation occurs*, though that increase in q$_v$' aloft is often unable to overcome higher q$_v$' values near the surface in boundary layers that are not well-mixed, which occurs frequently at ENA (Figures 2b, 2d, and 2f). As a result, the near-surface q$_v$' often decreases when upper-PBL air descends to the surface with the downdraft (Figure 2c). We therefore omit q$_v$' from the algorithm.

After reaching the surface, the negatively buoyant CP air spreads laterally, altering the near-surface wind speed and potentially its direction. Assuming the mean wind direction at the surface is similar to that of the raining cloud at the top of the boundary layer, the leading CP

edge should be associated with anomalously high near-surface wind speed (Figure 2e) that is approximately coincident with the decrease in temperature. Thus, CPs must be associated with a decrease in T' and a likely increase in S', as measured by the surface meteorological instrumentation (MET and MAWS). It is important to note that the existence of CPs does not require the existence of co-located rain at the surface (CP area is greater than rain area and likely longer-lived), nor does the existence of a CP require any rain to reach the surface (Jeong et al., 2023). Conversely, surface-reaching rain may not produce a CP at the time the airmass intersects the surface instrumentation (e.g., in the presence of a cool/moist surface layer or a nearly saturated boundary layer). We therefore do not require surface-detected rain for the identification of specific CPs. We expect that recently-formed rain events whose first surface-reaching drops fall upon the surface instrumentation have not yet had enough time to develop a CP, leaving them unobservable by the surface instrumentation. On the other hand, we expect CPs to survive longer than the rain falling from an individual cloud. Figure 2 illustrates some of the challenges in detecting CPs formed by boundary layer precipitation.

When looking for CPs in the temperature and wind speed time series, we first define CP "candidates" as a contiguous decrease in T' of any duration from the crest to the next trough. Most of these 353,338 candidates are manifestations of the background thermodynamic variability and will be rejected by the CP detection algorithm. We utilize three metrics to detect CPs from the surface station time series at ENA, each of which will produce a single value for each CP candidate. The first is the "Depth" metric, which is simply the accumulated drop in T' during the candidate: $Depth = \max(T') - \min(T')$. This is most similar to the methods employed by Wilbanks et al. (2015). The second metric is the "Rate", which is the absolute value of the strongest 1-minute decrease in T' during the candidate: $Rate = |min(\Delta T'/\Delta t)|$. The Rate metric is similar to what is used in many previous works (Terai and Wood, 2013; Vogel et al., 2021). Note that although the correlation between Depth and Rate is 0.92, indicating a high degree of shared information, we retain the use of both complementary metrics because the Depth metric is less susceptible to instrument noise and small-scale variability, and the Rate metric is less susceptible to longer duration non-CP signatures. The third metric is the "Gust", which is the difference between the maximum and minimum S' within 10 minutes before and after the time associated with the steepest drop in temperature ($t_{Rate}$): $Gust = \max(S'(t_{Rate} - 10: t_{Rate} + 10)) - \min(S'(t_{Rate} - 10: t_{Rate} + 10))$. The Gust metric is defined over a longer period because the measured peak in S' associated with surface-reaching rain is not always coincident with $t_{Rate}$.

**3.3 Time-varying metric thresholds**

At the heart of our CP retrieval algorithm is the requirement for the Depth, Rate, and Gust metrics to exceed values characteristic of the background thermodynamic variability, which varies throughout the year and day. We therefore set separate thresholds for each metric as a function of time of day and time of year. Broadly, the thresholds are determined by identifying the metric value for which the fraction of candidates associated with rain event starts ($F_{RE}$) deviates significantly from that group's average $F_{RE}$. The technique is detailed as follows.

The candidate Depth, Rate, and Gust metrics are collected into one-month/two-hour groups for a total of 144 groups. For a given month/hour group, candidates belonging to the 8 neighboring month/hour groups are aggregated to improve counting statistics. Figure 3a and Figure 3d demonstrate that the Depth distributions during January 0000-0200 UTC and July

1200-1400 UTC peak strongly at very weak values. The vast majority of candidates will be rejected in favor of relatively few confidently-detected CPs. The fraction of candidates that are associated with rain event starts ($F_{RE}$) increases with increasing Depth in both groups (Figure 3b and 3e), eventually exceeding the average for the group. We then perform $\chi^2$ tests for dependence between the $F_{RE}$ from the entire group and the $F_{RE}$ as a function of Depth, with the null hypothesis being that the distribution of $F_{RE}$ for the entire group is not different from the distribution of $F_{RE}$ for candidates with a given Depth value. These subsets are defined using a moving window approach, allowing for a dynamic assessment of variations in $F_{RE}$ over different segments of the group. The p-value from the test indicates whether there is a significant difference between the full-group average and the subset values. The weakest metric value for which $p \leq 0.01$ and $F_{RE}>mean(F_{RE})$ is selected as the metric's threshold for that month/hour group. Figures 2c and 2f demonstrate that the Depth threshold for daytime summer is larger than for nighttime winter, reflecting the different levels of variability and rain event frequency between those regimes.

[Figure]

**Figure 3: The process of defining Depth thresholds (red dashed lines) for January 0000-0200 UTC (a-c) and July 1200-1400 UTC (d-f). (a and d) Histogram of the Depth in black. (b and e) Rain event frequency as a function of Depth metric in black with the group-average $F_{RE}$ in solid-red. (c and f) $\chi^2$ and associated p-value for a test of dependence between $F_{RE}$ as a function of Depth and the grou-average $F_{RE}$. Thresholds are defined when the p-value diminishes below 0.01.**

Figure 4a shows the full annual/diurnal cycle of candidate frequency at ENA. The distribution peaks during the winter and spring evening and night a secondary peak during the summer daytime. However, we do not expect Figure 4a to reflect the occurrence frequency of actual CPs due to time-varying background thermodynamic variability and rain event frequency. Figure 4b shows that the frequency of distinct rain events is actually at a minimum in the summer daytime and evening, with its peak during the winter daytime. The thresholds for Depth (Figure 4c) and Rate (Figure 4e) are highly correlated, as expected, with the strictest thresholds found during the summer daytime when the influence of solar heating-driven boundary layer variability is greatest. The Gust thresholds peak during the summer nighttime, with a minimum

1000-1400 UTC from winter to late spring. Figure 4 also shows the fraction of candidates that exceed their respective thresholds (Figure 4d, 4f, and 4h). Fractionally, more candidates exceed their thresholds during the fall and winter seasons, especially during the night hours. In summary, Figure 4 demonstrates this technique's ability to allow the Depth, Rate, and Gust thresholds to reflect expected seasonal and diurnal patterns of background thermodynamic variability and rain event frequency.

[Figure]

**Figure 4: a) Total number of candidates belonging to each month/hour group. (b) Expected frequency of rain events. (c, e, and g) Depth, Rate, and Gust thresholds found at ENA. (d, f, and h) The fraction of candidates that exceed the respective Depth, Rate, and Gust thresholds. The time zone at ENA is UTC-1.**

**3.4 Detecting cold pools from the metrics and time-varying thresholds**

Now we turn to the designation of CPs based on a candidate's metrics exceeding the relevant thresholds. First, we group the temperature-related metrics (Depth and Rate) together. If either the Depth or Rate metrics do not exceed their thresholds, the candidate is not considered to be a CP and its weight is set to 0.0. If all three Depth, Rate, and Gust metrics exceed their

thresholds, the candidate is given a weight of 1.0, indicating that we are fully confident that it is a CP and not background thermodynamic variability. We also find cases where the Depth and Rate thresholds are exceeded but the Gust threshold is not. Some of these cases are accompanied by surface precipitation, but some are clearly not. With a lack of further information, we set the weight to 0.5 for these cases, reflecting an uncertainty in their identify while including them with an appropriate lower confidence. These cases weighted 0.5 have less influence on computed statistics than cases for which all three thresholds are exceeded.

Following these rules, we assign a weight to each candidate in the record, constituting our "Best" estimate of the CPs at ENA. Alternative estimates produce qualitative uncertainty ranges; the "Exclusive" estimate requires all three metrics to exceed their thresholds and the "Permissive" estimate requires only the Depth and Rate metrics to exceed their thresholds. Unless otherwise noted, results are shown for this Best estimate. Because the choices in this final step are subjective, we later examine alternative choices to gain an understanding of qualitative uncertainties. First we proceed with a discussion of the general properties of CPs retrieved by our Best estimates.

[Figure]

Figure 5: (a) Number of candidates, (b) $F_{RE}$, (c) sum of candidate weights, and (d) the average candidate weight as a function of the Depth and Gust metrics.

Figure 5 provides an understanding of where retrieved CPs inhabit the bivariate Depth and Gust space, with the understanding that Depth and Rate provide similar information. Most candidates have very weak Depth and Gust values (Figure 5a). However, those weak-value candidates are usually given a weight of zero (Figure 5c and 5d), consistent with the very low $F_{RE}$ in that space (Figure 5b). Note that Depth and Gust are not directly correlated, meaning each

is providing new information to the algorithm. Figure 5b shows that extreme cases in which either Depth≫0 and Gust≈0 or Depth≈0 and Gust≫0 are associated with an increase in the fraction of candidates that contain rain compared to when both Depth and Gust are weak, supporting their inclusion in the algorithm. Most of the summed candidate weights are associated with Depth≈0.4 K and Gust≈1 m/s. Figure 5d communicates that, while these cases are quite frequent at ENA, they are usually assigned a weight of 0.5. This points to the challenge of separating CPs from background thermodynamic variability at ENA, especially using simple 1-minute changes in temperature and wind speed. Regardless, the reduced weights of these candidates reduce their overall influence on the statistics presented in the CP climatology we present in the next section. Overall, Figure 5 shows that most candidates have weak Depth and Gust metrics are generally discarded by the algorithm and that higher candidate weights are associated with increased frequency of observed surface rain.

Figure 6 provides an alternative view, where the fraction of candidates that are designated as CPs is displayed as a function of candidate Depth. The candidate frequency distribution (log-spaced bins) sharply decreases for stronger Depth values greater than about 0.2 K. However, many of those weak-Depth candidates are likely background thermodynamic variability and are therefore not associated with CPs. Note again that a constant threshold in Depth would result in both missed and false detections because the CPs at ENA tend to be weak and their Depth distribution merges with the background variability. Weak candidates (e.g. Depth <0.2 K) are almost never diagnosed as CPs, while strong candidates (Depth>2.0 K) are almost always diagnosed as CPs.

[Figure]

**Figure 6: Histogram of the strength of candidates (dotted black) and retrieved cold pools (solid black) with the fraction of candidates that are estimated to be cold pools in blue.**

**3.5 Algorithm justification**

Figure 7 provides a proof of concept for CP identification through a series of relatively heavy and frequency rain events occurring during the ACE-ENA winter intensive observation period (Wang et al., 2022). CP candidates are highlighted with red lines, where the line thickness

scales with the confidence weights assigned by the detection algorithm. The candidates with thick red lines received a weight of 1.0, indicating strong confidence that they are CPs. Indeed, the first five of these CPs coincide with surface-reaching precipitation. Later, two CP candidates received a weight of 0.5 (shown with thin red lines) because their Gust values did not exceed their thresholds. The KaSACR scans spanning these final candidates are provided in Supplementary Figure 2 and reveal the horizontal structure of evolving precipitation passing near the ENA site. Although the KaSACR did not observe rain near the instrumentation during the 13:45 UTC candidate determined to be a CP, we suspect that the steep drop in T' represents a CP *remnant* from a terminated rain event that continues to propagate with the boundary layer flow. The KaSACR scans around 14:20 UTC and 15:04 UTC indicate scattered precipitation near the site and KaSACR scans at 15:15 UTC reveal that rain did indeed pass quite near but not directly over the ENA instrumentation (Supplemental Figure 2), suggesting that the CP from that nearby rain expanded laterally after descending to the surface. We suspect that the gust front intensity decreases as the CP expands farther from the downdraft, but we leave that investigation to future work. Figure 7 demonstrates that the "Best" CP detection algorithm can identify both CPs with and without associated surface rain observed by the ORG, PWD, and LD.

[Figure]

**Figure 7: (a) T', (b) 1-minute T' changes, and (c) S' for a selected period during ACE-ENA. Thick and thin red lines denote candidates for which weight W=1.0 W=0.5, respectively. Candidates with W=0.0 are not labeled. At the top of each panel for each candidate, the top number reflects the relevant metric value and the bottom number reflects the threshold for that month/hour group.**

Although there is no truth dataset against which we can judge the algorithm's performance, we executed the $E_{NR}/E_{FFR}$ analysis previously shown for the constant threshold technique (Figure 1f). The "Best" algorithm produces $E_{NR}/E_{FFR}$=2.65, indicating CPs are 2.65 as likely to be found in the NearRain periods as the FarFromRain periods. This value is coincidentally only slightly above the $E_{NR}/E_{FFR}$=2.35 value found for the constant threshold of -0.05 K/min used in Vogel et al. (2021) applied to ENA observations. The coincidence further

extends to the total expected rates of CPs between our algorithm (5.95 CPs per 24 hours) and the Vogel et al. (2021) algorithm (5.96 CPs per 24 hours). However, the similarities end there, as the diurnal and annual cycles will be shown to be quite different between the two techniques (Section 4.2). The Exclusive and Permissive estimates produce a qualitative uncertainty range of 3.57 to 8.34 CPs per 24 hours of valid observations. Note that while this qualitative uncertainty range is large, we will show that *the seasonal and diurnal cycles are not strongly affected by the choice of algorithm, as long as the thresholds vary throughout the year and day.*

To justify the Best estimate's performance, Figure 8 shows the evolution of surface rain for four regimes in the Depth/Gust metric space. The regime boundaries were chosen to separate the candidates by their metric values while maintaining adequate CP sampling and have the following sampling counts listed as the number of estimated CPs divided by the number of candidates: Weak-Gusty (340/1600), Strong-Gusty (1096/1102), Weak-Calm (166/42373), and Strong-Calm (460/948). At $t_{Rate}$, the Weak-Calm candidates have a rain fraction (8.6 %), slightly higher than the overall average (7.9 %), supporting the reduced average CP weight of the Weak-Calm candidates (0.053). Conversely, The Strong-Gusty and Strong-Calm candidates have elevated rain fraction, conditional rain rate, and average rain rate surrounding $t_{Rate}$, supporting their much higher average weight (0.995 and 0.485, respectively). While the rain fraction within ±1 hour surrounding $t_{Rate}$ is near-constant at 4.3 times the overall average for Weak-Gusty candidates, the rain rates are both elevated and variable, suggesting variability in rain rates and wind speeds within longer-lived rain events, possibly frontal systems or mesoscale systems.

[Figure]

**Figure 8: Mean evolution of surface rain for candidates falling within four regions of the Depth/Gust bivariate metric space. "Weak" and "Strong" candidates have 0.1<Depth<0.3 K and 1.0<Depth<5.0 K, respectively. "Calm" and "Gusty" candidates have 0.0<Gust<1.0 m/s and 2.0<Gust<13.0 m/s, respectively. Uncertainty envelopes represent 95% confidence intervals around the mean, obtained from $10^3$ bootstrap resamples with replacement at each time point. Panels depict (a) rain fraction, (b) conditional rain rate excluding non-raining times, and (c) average rain rate including non-raining times.**

**4 Annual and diurnal cycles of cold pools at ENA**

We now examine the 2016-2024 record of cold pools observed at the ENA site. In total, the algorithm identifies 308,701 CP candidates throughout the 9-year analysis period. However, once we account for the time-dependent background variability, there are only 23,509 candidates with weights greater than zero (Figure 5a). Those weights sum to an expected total of 19,198 CPs that are distinguishable from the background variability. Recall that the candidates for which the Depth and Rate thresholds *are* exceeded but the Gust threshold *is not* exceeded are assigned a weight of 0.5 and therefore have a reduced influence on the CP statistics, reflecting the reduced confidence that those cases are CPs. For context, there are a total of 27,596 distinct rain events during this period. So the sum of the CP candidate weights is only about 61.3 % of the number of

observed surface rain events. On the other hand, only 33.7 % of CP candidates with weight equal to 1 are associated with observed surface rain during the candidate. This section describes the characteristics of those 19,198 retrieved CPs.

**4.1 Cold pool properties**

We now examine CP statistics from the entire population of CP candidates. To understand the typical cloud scene associated with CPs, we show the connection between stronger candidates and more cumulus-like scenes in Figure 9. The discrimination between stratocumulus (Sc) and shallow cumulus (Cu) cloud regimes is based on Zheng and Miller (2022), who diagnosed cloud structure in 6-hour segments with three cloud/precipitation variables gleaned from the ARM Ka-band zenith-pointing radar (KaZR) and ceilometer at ENA. Their "Thickness Index" (TI in Figure 9) broadly represents the fraction of the boundary layer inhabited by clouds, their Drizzle Index (DI in Figure 9) increases for segments with more intense drizzle, and their Complexity Index (CI in Figure 9) increases in time segments characterized by variable cloud base heights. See Zheng and Miller (2022) for details. Rather than perform a k-means categorization to define Cu and St segments as done in Zheng and Miller (2022), we simply assign thresholds to the TI, DI, and CI values as detailed in the Figure 9 caption. Not all segments are assigned as Sc or Cu. Figure 9c shows that as Depth increases, the fraction of candidates independently classified as Cu scenes also increases. Contrastingly, the fraction classified as Sc begins slightly under the overall average decreases to zero in candidates with stronger Depth, indicating that stronger CPs are not associated with Sc cloud types at ENA. Figure 9 demonstrates that CPs are more associated with cumulus-topped boundary layers instead of stratocumulus regimes, especially for strong CPs.

[Figure]

Figure 9: (a) Scatterplot of categorized 6-hour segments. Segments are labeled as Sc if they have TI<0.20, DI<0.10, CI<0.30, and more than 80 % cloud cover. Segments are labeled as Cu if they have TI<1.00, DI>0.05, CI>0.20, and less than 70 % cloud cover. Not all segments are assigned as Sc or Cu. A rotated perspective of (a) is shown in (b). (c) Fraction of **cold pool detections** that are associated with Sc and Cu cloud types as a function of CP candidate Depth. Overall average values (thin horizontal lines) are computed as the fraction of all valid 1-minute time points that are classified as either Sc or Cu.

[Figure]

**Figure 10: Average vertical air motions from the zenith-pointing Doppler lidar during candidates for which (a) RR==0 and Depth>1.0 K, (b) 0<maximum RR<10 mm/hr, and (c) RR>10 mm/hr. The color bar limits in (b) and (c) are saturated to show details of the areas of both upward and downward motions. The strongest downward motion in (b) and (c) is -0.23 m/s and -0.59 m/s, respectively.**

CPs at ENA are coincident with downward motion that is preceded by upward motion. Figure 10 shows this evolution by compositing time series of sub-cloud vertical motion retrieved by the zenith-pointing Doppler lidar during CP candidates. In all panels, upward motion precedes downward motion surrounding $t_{Rate}$. These patterns show the updrafts and downdrafts coincide with the CPs observed by the surface station. In all panels, ascending motion precedes descending motion coinciding with $t_{Rate}$, likely representing a combination of the updraft and mechanical lifting of boundary layer air by dense CP air. About 20-30 minutes after $t_{Rate}$, it appears that another updraft tends to pass over the lidar, with similar timing to the increases in rain rate shown in Figure 8. In Figure 10a, which includes strong Depth candidates with no observed surface rain, there still exists some downward motion preceded by upward motion, indicating that the rain system may have passed nearby but not directly over the site, as was demonstrated in Figure 7 and Supplementary Figure 2. Although the vertical motions in Figure 10 appear to be weak, note that CP sizes can be small (shown later in Figure 12), so averaging vertical motions that are slightly offset in time will result in a diminished picture of the individual in-downdraft descent rate.

**4.2 Cold pool climatology and synoptic setting**

The CP annual and diurnal cycles in ENA CPs are shown in Figure 11. The annual cycle (Figure 11a) is more pronounced than the diurnal cycle (Figure 11b), with CPs being observed most frequently in the boreal winter months and at night when the background variability is

relatively calm (Figure 3 and Figure 4). In contrast to the constant-threshold method (Figure 1c), retrieved CPs using the Depth, Rate, and Gust metrics with dynamic thresholds (Figure 11a) closely match the annual cycle of precipitation events at ENA (Figure 1a). Note that the diurnal cycle of CPs (Figure 11b) peaks between 0000-0500 UTC, while the peak in the diurnal cycle of rain events peaks at about 0800 UTC and is slightly elevated during until about 1400 UTC (Figure 1b). This discrepancy is explained by the increased background thermodynamic variability in the boundary layer during daylight hours, requiring increases in the threshold value that is used to identify CPs (as shown in Figures 4c and e). Rain events during the peak times (~0800 UTC) certainly lead to cool, negatively buoyant air that descends to the surface. However, the signals from those events are embedded in stronger background variability, making their signals both more difficult to detect but also less meaningful regarding their effects on further convection and changes to the boundary layer structure and cloud morphology. The seasonal and diurnal cycles are not strongly affected by the choice of algorithm, as long as the thresholds vary throughout the season and day following background thermodynamic variability and the frequency of rain events (Figures 11a and 11b).

Figure 11d shows the monthly expected number of CPs and rain starts for the 2016-2024 record. Correlations between the number of expected monthly rain events of any minimum rain rate with the number of expected monthly CPs is 0.68. The correlation increases to 0.73 when including only rain events with maximum rain rate greater than 1 mm/hr. A simple regression of the time series of monthly expected CP counts yields no significant linear trend at the 95% confidence level, though more sophisticated techniques are warranted for a clearer picture of the dependence of CPs on atmospheric and oceanic trends and cycles affecting the boundary layer at ENA.

[Figure]

**Figure 11: Expected frequency of retrieved cold pools as a function of (a) month, (b) UTC, and (c) month/UTC. In (a) and (b), thick black lines represent our Best estimate (described in the text), while the dotted line represents the most restrictive estimate (Depth, Rate, and Gust required to exceed their thresholds; Exclusive) and the dashed line represents the most**

Figure 1d reveals bivariate histograms of ΔT'/Δt and Δqᵥ'/Δt for time periods that are associated with rain (NearRain; surface rain observed within 1 hour before or after) and not associated with rain (FarFromRain; no surface rain observed within 6 hours before or after). We assume that the FarFromRain category does not contain cold pool signatures and is therefore representative of the "background turbulence" experienced at the site, though we note the possibility of an isolated rain shower passing near the met station, thus contaminating the FarFromRain category with larger deviations. In total, there are 703,041 individual FarFromRain minutes, about 17.6% of the total observational record. Temperature and moisture deviations during FarFromRain periods are generally symmetric about the origin and are usually very small at the 1-minute time scale used in this work. In contrast, the temperature deviation extremes for NearRain periods are skewed towards ΔT'/Δt<0, indicating a tail in the distribution resulting from evaporation of rain. Note that the NearRain category is expected to contain most of the true cold pools but there is no distinct distribution of ΔT'/Δt and Δqᵥ'/Δt that can be clearly delineated from the non-cold pool distribution. Therefore, defining a strict threshold in ΔT'/Δt and/or Δqᵥ'/Δt would result in both missed detections of small cold pools and false alarms due to anomalously large background turbulence. ¶
In addition to these metrics that are observable with the surface meteorological instrumentation, we expect that rain events whose first surface-reaching drops fall upon the surface met station have not yet had enough time to develop a cold pool, leaving them unobservable by the surface instrumentation. On the other hand, we expect cold pools to survive longer than the rain falling from an individual cloud. Figure 1 illustrates some of the challenges in detecting cold pools formed by boundary layer precipitation.¶
¶                                                                    ... [4]

**permissive estimate (Depth and Rate required to exceed their thresholds; Permissive). Expected rates of distinct rain events (all events and only those with maximum rain rate stronger than 1 mm/hr) along with the expected cold pools for each month during 2016-2024 (d). The time zone at ENA is UTC-1.**

Figure 12 presents the retrieved CP size distribution, where the sizes have been estimated from the CP candidate time durations with the following arguments. As explained in Section 3, we define the start and finish of a given CP candidate as the preceding local maximum T' and following local minimum T', respectively. The along-wind length of the part of the CP that passes over the ENA site can be estimated by the elapsed time multiplied by the propagation speed, as long as the CP evolution is slow compared to the time the CP requires to pass over the instrumentation. However, because the CPs are expected to intersect the surface instrumentation at a random location along the across-wind direction of the CP, this simple calculation is likely to underestimate the actual CP radius. By assuming that (1) CPs are circular (with the ΔT'<0 portion forming the leading semicircle), (2) intersections with the ENA instrumentation are uniformly random, and (3) the average wind speed during the CP temperature decrease represents the CP propagation speed, we create a distribution of possible radii (*r*) for a given CP from the procedure detailed in Supplementary Figure 3. In short, the procedure involves computing a distribution of possible radii for each CP candidate and adding the possible radii for all CPs, accounting for each CP candidate's weight. The weighted mean CP radius is found to be 8.0 km (Figure 12) though most CPs have a smaller size. The CP radius distribution peaks at about 4.5 km. About 5.1% of retrieved CPs have radii less than 1 km, while about 26.1% have radii greater than 10 km. For comparison with the southeast Pacific Ocean, Terai and Wood (2014) found a median CP size of about 6 km but Wilbanks et al. (2015) found CP lengths ranging from 5-40 km with a mean length of 15 km.

[Figure]

**Figure 12: PDFs of observed chord length (blue) and estimated CP radius (red).**

Figure 13 shows the synoptic setting for days with a high versus low number of CPs. Sea level pressure values are taken from the Modern-Era Retrospective Analysis for Research and Applications, Version 2 (MERRA-2; Gelero et al., 2017). We compute the daily-average of the 3-hourly instantaneous values provided by the "inst3_3d_asm_Nv" product for days that either have a large number of CPs in that day (277 days) or a small number of CPs in that day (206

[Figure]

To characterize the background bivariate deviations separately in each month/hour group, we apply singular value decomposition (SVD) to the ΔT'/Δt and Δq_v'/Δt belonging to that group and the eight neighboring groups. This produces the eigenvalues and eigenvectors that represent the relative magnitude of variabili... [5]

days). Days with many CPs tend to be associated with sea level pressure (SLP) depressions (Figure 13a and 13b), especially those centered to the North and East of the Azores such that Graciosa Island is likely experiencing a post-frontal environment with frequent shallow precipitation events due to subsidence in the lee of a passing trough (Lamer et al. 2020). In contrast, days with few CPs are characterized by slight ridging near and to the East of the Azores islands (Figure 13c and 13d).

[Figure]

Figure 13: (a) Sea level pressure (SLP) maps for days with frequent cold pools. (b) Same as (a) but as a deviation the average SLP at each location. Panels (c) and (d) are the same as (a) and (b) but for days with few cold pools. In each panel, the black "x" designates the location of the ENA site. Sea level pressure values are taken from MERRA-2

**5 Which Rain Events Produce Observable Cold Pools?**

Our Best estimate for CP frequency at ENA is 5.95 CPs per 24 hours but the expected rate of rain events (F_{RE}) is 9.08 rain events per 24 hours. Given that we detect CPs that are not associated with observed rain at the site, why do so many rain events *not produce* an observable CP at the surface as the rain passes over the ENA instrumentation? This section explores which rain events result in CPs that are distinct from the background thermodynamic variability.

Figure 14 shows the frequency distribution of mean CP candidate weights as a function of variables relevant to boundary layer structure and CP formation, while Figure 15 shows the frequency distributions of retrieved CPs compared to the full distributions and the distributions during rain event starts. As expected, rain events with heavier rain in the beginning of the event are more likely to be accompanied by measurable CPs (Figure 14a). Rain events starting with rain rates greater than about 0.5 mm/hr result in a significant elevation of CP weight compared to

4 Characteristics of

the average of all rain events. Indeed, CPs tend to form in association with higher rain rates than the typical rain event (Figure 15a).

While only 38.4 % of rain events occur when the surface relative humidity is less than 80 %, 56 % of CPs occur in these relatively dry conditions (Figure 15b). A given rain event is more likely to create a measurable CP if the relative humidity immediately preceding the rain event start is between 60-80 % (Figure 14b). There are multiple plausible explanations for this, including that (1) less evaporation occurs when the sub-cloud layer is humid, leading to diminished downdrafts and therefore diminished CP signatures and (2) a previous CP conditioned the near-surface layer with moist air, effectively creating a low-level inversion that traps moisture near the surface. The boundary layer integrated saturation deficit exhibits clear associations with mean CP weights, as well (Figure 14h). We define the integrated saturation deficit as $H_{CB} \sum (100 - RH(z))$ in all evenly-spaced sub-cloud layers, where $H_{CB}$ is the cloud base height from ceilometer and RH(z) is the relative humidity [%] in a given layer. This metric is intended to summarize both the relative humidity of the sub-cloud layer and the distance the drops need to fall through sub-saturated air. Rain events with very low integrated saturation deficit (either high humidity or low cloud base heights or both) have a strongly reduced chance of being associated with CPs. CPs tend to occur when the saturation deficit is higher than typical for rain events (Figure 15h).

We also examined the rate of surface pressure change while rain events when rain is first observed at the station (Figure 14c and Figure 15c). The 1-hour pressure change was computed as P(t+30) – P(t-30), where P is the air pressure measured by either MET or MAWS and t is the time of the rain event start. CP weights tend to be slightly greater than the average when surface pressure increases by more than about 0.50 hPa over the course of an hour, which we speculate is related to post-frontal scattered showers and potentially cold air outbreaks. These cases represent the upper tail of the 1-hour pressure change distribution (Figure 15c).

Near-surface horizontal winds during rain events are also shown to be connected to CP passages at ENA. Rain events with surface horizontal wind speed greater than 7 m/s are associated with elevated chances of CPs, while calm conditions are not (Figure 14d). Because the strongest wind speeds are less frequent, most CPs are associated with wind speeds between 5 – 11 m/s (Figure 15d). Increased wind speed variability is also associated with a higher chance that a rain event will lead to a measurable CP (Figure 14f). While this CP detection algorithm utilizes the change in wind speed within ±10 minutes of $t_{Rate}$, here we assess the wind speed variability over a longer period of two hours, better representing the boundary layer characteristics outside of the immediate vicinity of the rain event. Figure 15f shows that, although rain events are associated with a higher wind speed variability than the distribution for all times, rain events that lead to CPs exist within environments with further increased dynamical variability.

CPs are associated with 2-hour low cloud cover (defined as the fraction of the time that the ceilometer retrieves a cloud base lower than 4 km from 60 minutes prior to 60 minutes after a given time) between about 0.35 and 0.90 (Figure 14e). While rain events are associated with low cloud fraction near 1.0 (Figure 15e), CPs are not as closely associated with complete cloud coverage. This is consistent with the findings of Figure 9, where the stratocumulus cloud type is less frequently associated with CPs in candidates with larger Depth metric.

The mean CP weight is positively correlated with estimated inversion height, especially inversion heights greater than 2 km (Figure 14g). In the shallowest boundary layers, precipitation experiences less time in the sub-saturated sub-cloud layer, resulting in less total evaporation. We also speculate that the rain rates tend to be weaker in very shallow boundary layers than in

deeper layers where convection has developed further. Inversion heights are taken from the Heffter method (Heffter, 1980), except in cases when the Heffter inversion height is less than 300 m, in which case we use layer with the greatest increase in potential temperature below 4 km. Figure 15g shows that the boundary layer depth distribution is increased towards higher values when CPs occur than for other rain events.

[Figure]

The boundary layer decoupling index and shear exhibit reduced signals compared to some other variables. Severely decoupled cases (DEC>10 K) tend to have smaller cold pool weights and most rain-associated CPs occur when 2<DEC K<7 (Fig. 13g). Very low values of PBL shear are associated with slightly increased CP weights (Fig. 13h). PBL shear at ENA is usually less than 10 m/s and CPs rarely occur above this value.¶
¶                                                    ... [15]

Figure 14: Blue lines show the uncertainty ranges for the average CP weight (including zeros for rejected candidates) for candidates that occur during the start of a rain event, as a function of variables obtained from the *in-situ* surface instrumentation (a, b, c, d, f), ceilometer (e), and balloon sondes (g, h). Error bars denote the 95% confidence intervals computed via bootstrap resampling with $10^3$ resamples with replacement. Red lines provide the corresponding confidence intervals for the overall average candidate weight during all rain events.

[Figure]

[Figure]

**Figure 15: Like Figure 14, but the fraction of retrieved cold pools (blue), the during rain starts regardless of cold pool detection (black), and the overall distributions regardless of cold pools or rain events (grey).**

**6 Conclusions**

This work presents cold pool (CP) statistics from 9 years of surface measurements at the ARM ENA site on Graciosa Island in the Azores. Seeking to avoid a combination of false alarms and missed detections that necessarily result from constant temperature change methods, we developed a new methodology that: (1) uses three metrics (Depth, Rate, and Gust) to characterize CP candidates, (2) assigns thresholds for each metric based on statistical association between the frequency of distinct rain events and increasing magnitudes of the three metrics, and (3) allows those thresholds to vary throughout the annual and diurnal cycles, thus accounting for time-varying background thermodynamic variability that is not associated with CPs.

Despite having no "truth" benchmark with which to validate the CP detections, we demonstrated expected CP behavior related to a time series of surface-reaching rain coupled with scanning radar of passing precipitation systems, surface-reaching frequency and intensity of rain for candidates belonging to four regions of the Depth/Gust bivariate space, and the frequencies of CPs in cases we are confident did or did not contain CPs using a combination of surface-observed precipitation and satellite-based IMERG precipitation retrievals. We presented statistics on CPs over 9 years, including their total number and temporal cycles, their relationships to synoptic weather regimes, and their physical sizes. We then analyzed the characteristics of the boundary layer associated with the observed CPs during surface-reaching rain events. Our findings can be summarized as follows:

1. Identifying ENA CPs using a simple 1-minute temperature change threshold from meteorological station observations results in both missed detections and false alarms because a fixed threshold does not account for changes in the background variability throughout the year and day, regardless of the exact threshold value used. This simple constant threshold method tends to overcount CPs (i.e., detects false alarms) during the afternoon hours due to increased boundary layer thermodynamic variability while simultaneously missing weaker CPs that occur during winter and nighttime precipitation, leading to an overly-strong and reversed representation of the CP diurnal cycle. We also demonstrated the strong and undesirable sensitivity of the number of CPs retrieved to the choice of constant temperature threshold value.

2. The main advance presented in this work is the development of a method to assign time-varying thresholds determined by statistical associations between the frequency of distinct rain events and increasing magnitudes of three CP candidate metrics. The three metrics, each receiving one value for each CP candidate, permitted capture of diverse CP types, which may be targeted for investigation in future work. These types were presented in Figure 8 and include strong CPs with strong gusts (near or directly under the heavier precipitation source), strong CPs with weak gusts (likely CPs farther from the rain source), and weak CPs with strong gusts (associated with longer-lived rain events, potentially frontal systems).

3. Summing the candidate weights for our Best estimate, we identify 19,198 individual CPs that are distinct from the background variability from 2016 to 2024. The seasonal cycle of ENA CPs peaks in the colder months, with ~7-8 CPs per 24 hours compared to ~4 expected CPs per 24 hours in the summer months. The CP diurnal cycle is less pronounced than the annual cycle, consistent the muted diurnal cycle in the frequency of distinct rain events. Although the frequency of rain events peaks from 0800-1400 UTC, the temperature and wind gust signals

Deleted: <#>Inclusion of 1-minute changes in moisture enhanced the retrieval, specifically during times when temperature decreases and moisture increases, a reversal from the usual positive correlation between temperature and moisture in background turbulence. Allowing moisture to decrease slightly included cases in which evaporation of rain occurred in dry mid-PBL layers but, upon descent to the surface, were not moist enough to overcome the additional moisture in the lowest few meters of the atm... [17]

of these afternoon CPs are often obscured by elevated background thermodynamic variability, leaving them less likely to affect the boundary layer and leaving them statistically indistinguishable from non-CP processes. Days with more than 8 retrieved CPs are associated with the post-frontal sector of synoptic troughs.

4. The number of CPs detected here represents only about 66 % of the number of distinct rain events, as observed by contiguous 1-minute rain rates from surface instrumentation. When rain is first measured by the in-situ surface instrumentation, detected CPs tend to have the following properties: leading-edge rain rates greater than 0.5 mm/hr, near-surface relative humidity less than 80% and large values of integrated saturation deficit, deeper boundary layers, stronger horizontal near-surface wind speed and variability, and broken fractional low cloud cover between 0.35 and 0.90.

Though we are confident that the methodologies detailed here represent improvements in retrieving CP properties from simple near-surface time series observations from surface instrumentation, we also note the following unknowns, limitations, and potential future improvements to this methodology:

1. CP candidates that are similar in magnitude to the background thermodynamic variability are included in the statistics, though many of them are likely not actually CPs.

2. While we endeavored to demonstrate expected behavior of the algorithm in Section 3, we do not have a "truth" dataset that can provide a full validation of CP retrievals.

Although this algorithm was developed to examine weak CP signatures resulting from precipitation that is most-often confined to the boundary layer, we note that the metric thresholds could also be recomputed at other locations without the need for subjectively-tuned or constant thresholds, as long as the location has a long enough record to adequately characterize the background thermodynamic variability that is not associated with CPs and varies throughout the time of year and day.

**Code Availability**

Relevant algorithm and analysis code will be made publicly available on Zenodo prior to publication.

**Data Availability**

ARM ENA observations may be downloaded from the ARM data archive at https://www.arm.gov/data/. Half-hourly IMERG Final gridded precipitation retrievals can be accessed from NASA's GES DISC at https://disc.gsfc.nasa.gov/datasets/GPM_3IMERGHH_07/summary?keywords="IMERG final". MERRA-2 reanalysis sea level pressure fields may be accessed from NASA's GED Disc at

https://disc.gsfc.nasa.gov/datasets/M2I3NVASM_5.12.4/summary?keywords=inst3_3d_asm_Nv.

**Author Contribution**

MAS led this work, from the acquisition of the observations from the ARM data archive, the development and refinement of the cold pool detection algorithm, the figure generation, and the writing of the manuscript. MKW led the proposal that funded this work and the general direction of this work. MJC provided context regarding ongoing large eddy simulation work building off these results. JHJ organized the ACE-ENA scanning radar figures and provided advice for the Doppler lidar analyses. All co-authors contributed to the discussion of the techniques and results during biweekly meetings and to editing the manuscript.

**Competing Interests**

The authors declare that they have no conflict of interest.

**Acknowledgements**

This work was supported by the U.S. Department of Energy's Atmospheric System Research, an Office of Science Biological and Environmental Research program, under awards DE-SC0022992 and 89243022SSC000094. The analyses presented here relied upon accessible observations from the ARM Data Archive, for which we are grateful. This work was performed at the Jet Propulsion Laboratory, California Institute of Technology, under a contract with the National Aeronautics and Space Administration. The authors offer their sincere thanks to ARM director Jim Mather and instrument mentor Jenni Kyrouac for their helpful insights about the surface sensor noise characteristics at ENA.

**References**

Blossey, P. N., Bretherton, C. S., & Mohrmann, J. (2021). Simulating Observed Cloud Transitions
in the Northeast Pacific during CSET, Mon. Wea. Rev., 149(8), 2633-2658,
https://journals.ametsoc.org/view/journals/mwre/149/8/MWR-D-20-0328.1.xml.

Bretherton, C. S., & M. C. Wyant (1997), Moisture transport, lower-tropospheric stability, and decoupling of cloud-topped boundary layers, J. Atmos. Sci., 54(1), 148–167.

Ceppi, P., & Nowack, P. (2021). Observational evidence that cloud feedback amplifies global warming. Proceedings of the National Academy of Sciences, 118(30).

de Szoeke, S. P., Skyllingstad, E. D., Zuidema, P., & Chandra, A. S. (2017). Cold Pools and Their Influence on the Tropical Marine Boundary Layer. Journal of the Atmospheric Sciences, 74(4), 1149–1168. https://doi.org/10.1175/JAS-D-16-0264.1
* * *
Eastman, R., & Wood, R. (2018). The Competing Effects of Stability and Humidity on Subtropical Stratocumulus Entrainment and Cloud Evolution from a Lagrangian Perspective, *J. Atmos. Sci.*, **75**(8), 2563-2578, https://journals.ametsoc.org/view/journals/atsc/75/8/jas-d-18-0030.1.xml

Engerer, N.A., Stensrud, D.J. and Coniglio, M.C. (2008) Surface characteristics of observed cold pools. *Monthly Weather Review*, **136**(12), 4839–4849.

Feingold, G., I. Koren, H. Wang, H. Xue, and W. Brewer, 2010: Precipitation-generated oscillations in open cellular cloud fields. Nature, 466, 849–852.

Feng, Z., Hagos, S., Rowe, A. K., Burleyson, C. D., Martini, M. N., & de Szoeke, S. P. (2015). Mechanisms of convective cloud organization by cold pools over tropical warm ocean during the AMIE/DYNAMO field campaign. Journal of Advances in Modeling Earth Systems, 7(2), 357–381. https://doi.org/https://doi.org/10.1002/2014MS000384

Gelaro, R., and Coauthors, 2017: The Modern-Era Retrospective Analysis for Research and Applications, Version 2 (MERRA-2). J. Climate, 30, 5419–5454, https://doi.org/10.1175/JCLI-D-16-0758.1

Ghate, V. P., Cadeddu, M. P., & Wood, R. (2020). Drizzle, turbulence, and density currents below post cold frontal open cellular marine stratocumulus clouds, J. Geophys. Res. Atmos., 125, e2019JD031586, https://doi.org/10.1029/2019JD031586.

Ghate, V. P., M. P. Cadeddu, X. Zheng, and E. O'Connor, 2021: Turbulence in the Marine Boundary Layer and Air Motions below Stratocumulus Clouds at the ARM Eastern North Atlantic Site. J. Appl. Meteor. Climatol., 60, 1495–1510, https://doi.org/10.1175/JAMC-D-21-0087.1.

Giangrande, S. E., Wang, D., Bartholomew, M. J., Jensen, M. P., Mechem, D. B., Hardin, J. C., & Wood,R. (2019). Midlatitude oceanic cloud and precipitation properties as sampled by the ARM Eastern North Atlantic Observatory. Journal of Geophysical Research: Atmospheres,124, 4741–4760.https://doi.org/10.1029/2018JD029667

Goren, T., Kazil, J., Hoffmann, F., Yamaguchi, T., & Feingold, G. (2019). Anthropogenic air pollution delays marine stratocumulus breakup to open cells. Geophys. Res. Lett., 46(23), 14135–14144. https://doi.org/10.1029/2019GL085412.

Hartman, D. L., and D. Short, 1980: On the use of earth radiation budget statistics for studies of clouds and climate. J. Atmos. Sci., 37, 1233–1250.

Heffter, J. L.: Transport Layer Depth Calculations, Second Joint Conference on Applications of Air Pollution Meteorology, 24–27 March 1980, New Orleans, Louisiana, https://doi.org/10.1175/1520-0477-61.1.65, 1980.

Huffman, G.J. et al. (2020). Integrated Multi-satellite Retrievals for the Global Precipitation Measurement (GPM) Mission (IMERG). In: Levizzani, V., Kidd, C., Kirschbaum, D.B., Kummerow, C.D., Nakamura, K., Turk, F.J. (eds) Satellite Precipitation Measurement. Advances in Global Change Research, vol 67. Springer, Cham. https://doi.org/10.1007/978-3-030-24568-9_19

Huffman, D. T. Bolvin, R. Joyce, E. J. Nelkin, J. Tan, D. Braithwaite, K. Hsu, O. A. Kelley, P. Nguyen, S. Sorooshian, D. C. Watters, B. J. West, and P. Xie, 2023: NASA Global Precipitation Measurement (GPM) Integrated Multi-satellite Retrievals for GPM (IMERG) Version 07. Algorithm Theoretical Basis Doc., version 7, 47 pp.

Hunzinger, A., J. C. Hardin, N. Bharadwaj, A. Varble, and A. Matthews, 2020: An extended radar relative calibration ad- justment (eRCA) technique for higher-frequency radars and range–height indicator (RHI) scans. Atmos. Meas. Tech., 13, 3147–3166, https://doi.org/10.5194/amt-13-3147-2020.

Jeong, J.-H., Witte, M. K., Glenn, I. B., Smalley, M., Lebsock, M. D., Lamer, K., & Zhu, Z. (2022). Distinct dynamical and structural properties of marine stratocumulus and shallow cumulus clouds in the Eastern North Atlantic. Journal of Geophysical Research: Atmospheres, 127, e2022JD037021. https://doi.org/10.1029/2022JD037021

Jiang, J. H., Su, H., Wu, L., Zhai, C., & Schiro, K. A. (2021). Improvements in cloud and water vapor simulations over the tropical oceans in CMIP6 compared to CMIP5. Earth and Space Science, 8, e2020EA001520, https://doi.org/10.1029/2020EA001520.

Keeler, E., Kyrouac, J., & Ermold, B. Automatic Weather Station (MAWS), 2017. Atmospheric Radiation Measurement (ARM) User Facility. https://doi.org/10.5439/1162061

Keeler, E., Burk, K., & Kyrouac, J. Balloon-Borne Sounding System (SONDEWNPN), 2013-09-28 to 2025-04-08, Eastern North Atlantic (ENA), Graciosa Island, Azores, Portugal (C1). Atmospheric Radiation Measurement (ARM) User Facility. https://doi.org/10.5439/1595321

Kirsch, B., F. Ament, and C. Hohenegger, 2021: Convective Cold Pools in Long-Term Boundary Layer Mast Observations. Mon. Wea. Rev., 149, 811–820, https://doi.org/10.1175/MWR-D-20-0197.1.

Klein, S. A., Hall, A., Norris, J. R., & Pincus, R. (2017). Low-cloud feedbacks from cloud-controlling factors: A review. Shallow clouds, water vapor, circulation, and climate sensitivity, 135-157.

Kollias, P., and Coauthors, 2016: Development and applications of ARM millimeter-wavelength cloud radars. The Atmospheric Radiation Measurement (ARM) Program: The First 20 Years, Meteor. Monogr., No. 57, Amer. Meteor. Soc., https://doi.org/10.1175/AMSMONOGRAPHS-D-15-0037.1

Kruse, I.L., Haerter, J.O. & Meyer, B. (2022) Cold pools over the Netherlands: A statistical study from tower and radar observations. *Q J R Meteorol Soc*, 711–726, https://doi.org/10.1002/qj.4223

Kyrouac, Jenni, and Yan Shi. "Surface Meteorological Instrumentation (MET)." Atmospheric Radiation Measurement (ARM) User Facility, doi:10.5439/1786358. Accessed 3 May 2023.

Lamer, K., Naud, C. M., & Booth, J. F. (2020). Relationships between precipitation properties and large-scale conditions during subsidence at the Eastern North Atlantic observatory. *Journal of Geophysical Research: Atmospheres*, 125. https://doi.org/10.1029/2019JD031848

Mechem, D. B., C. S. Wittman, M. A. Miller, S. E. Yuter, and S. P. de Szoeke, 2018: Joint Synoptic and Cloud Variability over the Northeast Atlantic near the Azores. J. Appl. Meteor. Climatol., 57, 1273–1290, https://doi.org/10.1175/JAMC-D-17-0211.1.

Myers, T.A., et al. (2021). Observational constraints on low cloud feedback reduce uncertainty of climate sensitivity. Nat. Clim. Chang. 11, 501–507, https://doi.org/10.1038/s41558-021-01039-0.

Newsom, R., Shi, Y., & Krishnamurthy, R. Doppler Lidar (DLFPT). Atmospheric Radiation Measurement (ARM) User Facility. https://doi.org/10.5439/1025185

Provod, M., J. H. Marsham, D. J. Parker, and C. E. Birch, 2016: A Characterization of Cold Pools in the West African Sahel. *Mon. Wea. Rev.*, **144**, 1923–1934, https://doi.org/10.1175/MWR-D-15-0023.1.

Redl, R., A. H. Fink, and P. Knippertz, 2015: An Objective Detection Method for Convective Cold Pool Events and Its Application to Northern Africa. *Mon. Wea. Rev.*, **143**, 5055–5072, https://doi.org/10.1175/MWR-D-15-0223.1.

Rémillard, J., and G. Tselioudis, 2015: Cloud Regime Variability over the Azores and Its Application to Climate Model Evaluation. J. Climate, 28, 9707–9720, https://doi.org/10.1175/JCLI-D-15-0066.1.

Ritsche MT. 2011. ARM Surface Meteorology Systems Handbook. U.S. Department of Energy. DOE/SC-ARM/TR-086. 10.2172/1007926.

Scott, R. C., Myers, T. A., Norris, J. R., Zelinka, M. D., Klein, S. A., Sun, M., & Doelling, D. R. (2020). Observed sensitivity of low-cloud radiative effects to meteorological perturbations over the global oceans. J. Cli., 33(18), 7717-7734.

Smalley, K. M., Lebsock, M. D., Eastman, R., Smalley, M., and Witte, M. K.: A Lagrangian analysis of pockets of open cells over the southeastern Pacific, Atmos. Chem. Phys., 22, 8197–8219, https://doi.org/10.5194/acp-22-8197-2022, 2022.

Terai, C. R., & Wood, R. (2013). Aircraft observations of cold pools under marine stratocumulus. Atmos. Chem. Phys., 13, 9899–9914, doi:10.5194/acp-13-9899-2013.

Vogel, R., Konow, H., Schulz, H., and Zuidema, P.: A climatology of trade-wind cumulus cold pools and their link to mesoscale cloud organization, Atmos. Chem. Phys., 21, 16609–16630, https://doi.org/10.5194/acp-21-16609-2021, 2021.

Wang, J., and Coauthors, 2022: Aerosol and Cloud Experiments in the Eastern North Atlantic (ACE-ENA). Bull. Amer. Meteor. Soc., 103, E619–E641, https://doi.org/10.1175/BAMS-D-19-0220.1.

Wang, D., Bartholomew, M. J., Zhu, Z., & Shi, Y. Laser Disdrometer (LD). Atmospheric Radiation Measurement (ARM) User Facility. https://doi.org/10.5439/1973058

Wilbanks, M. C., Yuter, S. E., de Szoeke, S. P., Brewer, W. A., Miller, M. A., Hall, A. M., & Burleyson, C. D. (2015). Near-Surface Density Currents Observed in the Southeast Pacific Stratocumulus-Topped Marine Boundary Layer. Mon. Wea. Rev., 143(9), 3532-3555, https://journals.ametsoc.org/view/journals/mwre/143/9/mwr-d-14-00359.1.xml.

Wood, R., 2012: Stratocumulus Clouds. Mon. Wea. Rev., 140, 2373–2423, https://doi.org/10.1175/MWR-D-11-00121.1.

Wu, P., X. Dong, and B. Xi, 2020: A Climatology of Marine Boundary Layer Cloud and Drizzle Properties Derived from Ground-Based Observations over the Azores. J. Climate, 33, 10133–10148, https://doi.org/10.1175/JCLI-D-20-0272.1.

Wyant, M. C., Bretherton, C. S., Rand, H. A., & Stevens, D. E. (1997). Numerical simulations and a conceptual model of the stratocumulus to trade cumulus transition. J. Atmos. Sci., 54, 168–192, https://doi.org/10.1175/1520-0469(1997)054<0168:NSAACM>2.0.CO;2.

Yamaguchi, T., Feingold, G., & Kazil, J. (2017). Stratocumulus to cumulus transition by drizzle. J. Adv. Model. Earth Syst., 9(6), 2333–2349, https://doi.org/10.1002/2017MS001104.

Zheng, Q., and M. A. Miller, (2022). Summertime Marine Boundary Layer Cloud, Thermodynamic, and Drizzle Morphology over the Eastern North Atlantic: A Four-Year Study. J. Climate, 35, 4805–4825, https://doi.org/10.1175/JCLI-D-21-0568.1.

| Page 7: [1] Deleted | Smalley, Mark A (US 329J-Affiliate) | 4/18/25 12:56:00 PM |
|---|---|---|

| Page 7: [2] Deleted | Smalley, Mark A (US 329J-Affiliate) | 4/18/25 12:56:00 PM |
|---|---|---|

| Page 7: [3] Deleted | Smalley, Mark A (US 329J-Affiliate) | 4/18/25 12:56:00 PM |
|---|---|---|

| Page 23: [4] Deleted | Smalley, Mark A (US 329J-Affiliate) | 4/18/25 12:56:00 PM |
|---|---|---|

| Page 24: [5] Deleted | Smalley, Mark A (US 329J-Affiliate) | 4/18/25 12:56:00 PM |
|---|---|---|

| Page 24: [5] Deleted | Smalley, Mark A (US 329J-Affiliate) | 4/18/25 12:56:00 PM |
|---|---|---|

| Page 24: [6] Deleted | Smalley, Mark A (US 329J-Affiliate) | 4/18/25 12:56:00 PM |
|---|---|---|

| Page 24: [6] Deleted | Smalley, Mark A (US 329J-Affiliate) | 4/18/25 12:56:00 PM |
|---|---|---|

| Page 24: [6] Deleted | Smalley, Mark A (US 329J-Affiliate) | 4/18/25 12:56:00 PM |
|---|---|---|

| Page 24: [6] Deleted | Smalley, Mark A (US 329J-Affiliate) | 4/18/25 12:56:00 PM |
|---|---|---|

| Page 24: [6] Deleted | Smalley, Mark A (US 329J-Affiliate) | 4/18/25 12:56:00 PM |
|---|---|---|

| Page 24: [6] Deleted | Smalley, Mark A (US 329J-Affiliate) | 4/18/25 12:56:00 PM |
|---|---|---|

| Page 24: [6] Deleted | Smalley, Mark A (US 329J-Affiliate) | 4/18/25 12:56:00 PM |
|---|---|---|

| Page 24: [6] Deleted | Smalley, Mark A (US 329J-Affiliate) | 4/18/25 12:56:00 PM |
|---|---|---|

| Page 24: [6] Deleted | Smalley, Mark A (US 329J-Affiliate) | 4/18/25 12:56:00 PM |
|---|---|---|

| Page 24: [6] Deleted | Smalley, Mark A (US 329J-Affiliate) | 4/18/25 12:56:00 PM |
|---|---|---|

| Page 24: [6] Deleted | Smalley, Mark A (US 329J-Affiliate) | 4/18/25 12:56:00 PM |
|---|---|---|

| Page 24: [6] Deleted | Smalley, Mark A (US 329J-Affiliate) | 4/18/25 12:56:00 PM |
|---|---|---|

| Page 24: [6] Deleted | Smalley, Mark A (US 329J-Affiliate) | 4/18/25 12:56:00 PM |
|---|---|---|
| Page 24: [6] Deleted | Smalley, Mark A (US 329J-Affiliate) | 4/18/25 12:56:00 PM |
| Page 24: [6] Deleted | Smalley, Mark A (US 329J-Affiliate) | 4/18/25 12:56:00 PM |
| Page 24: [6] Deleted | Smalley, Mark A (US 329J-Affiliate) | 4/18/25 12:56:00 PM |
| Page 24: [6] Deleted | Smalley, Mark A (US 329J-Affiliate) | 4/18/25 12:56:00 PM |
| Page 24: [6] Deleted | Smalley, Mark A (US 329J-Affiliate) | 4/18/25 12:56:00 PM |
| Page 24: [6] Deleted | Smalley, Mark A (US 329J-Affiliate) | 4/18/25 12:56:00 PM |
| Page 24: [6] Deleted | Smalley, Mark A (US 329J-Affiliate) | 4/18/25 12:56:00 PM |
| Page 24: [6] Deleted | Smalley, Mark A (US 329J-Affiliate) | 4/18/25 12:56:00 PM |
| Page 24: [6] Deleted | Smalley, Mark A (US 329J-Affiliate) | 4/18/25 12:56:00 PM |
| Page 24: [6] Deleted | Smalley, Mark A (US 329J-Affiliate) | 4/18/25 12:56:00 PM |
| Page 24: [6] Deleted | Smalley, Mark A (US 329J-Affiliate) | 4/18/25 12:56:00 PM |
| Page 24: [6] Deleted | Smalley, Mark A (US 329J-Affiliate) | 4/18/25 12:56:00 PM |
| Page 24: [6] Deleted | Smalley, Mark A (US 329J-Affiliate) | 4/18/25 12:56:00 PM |
| Page 24: [6] Deleted | Smalley, Mark A (US 329J-Affiliate) | 4/18/25 12:56:00 PM |
| Page 24: [6] Deleted | Smalley, Mark A (US 329J-Affiliate) | 4/18/25 12:56:00 PM |
| Page 24: [7] Deleted | Smalley, Mark A (US 329J-Affiliate) | 4/18/25 12:56:00 PM |
| Page 24: [7] Deleted | Smalley, Mark A (US 329J-Affiliate) | 4/18/25 12:56:00 PM |
| Page 24: [7] Deleted | Smalley, Mark A (US 329J-Affiliate) | 4/18/25 12:56:00 PM |

| Page 24: [8] Deleted | Smalley, Mark A (US 329J-Affiliate) | 4/18/25 12:56:00 PM |
| Page 24: [8] Deleted | Smalley, Mark A (US 329J-Affiliate) | 4/18/25 12:56:00 PM |
| Page 24: [9] Deleted | Smalley, Mark A (US 329J-Affiliate) | 4/18/25 12:56:00 PM |
| Page 24: [9] Deleted | Smalley, Mark A (US 329J-Affiliate) | 4/18/25 12:56:00 PM |
| Page 25: [10] Deleted | Smalley, Mark A (US 329J-Affiliate) | 4/18/25 12:56:00 PM |
| Page 25: [10] Deleted | Smalley, Mark A (US 329J-Affiliate) | 4/18/25 12:56:00 PM |
| Page 25: [10] Deleted | Smalley, Mark A (US 329J-Affiliate) | 4/18/25 12:56:00 PM |
| Page 25: [10] Deleted | Smalley, Mark A (US 329J-Affiliate) | 4/18/25 12:56:00 PM |
| Page 25: [11] Deleted | Smalley, Mark A (US 329J-Affiliate) | 4/18/25 12:56:00 PM |
| Page 25: [11] Deleted | Smalley, Mark A (US 329J-Affiliate) | 4/18/25 12:56:00 PM |
| Page 25: [11] Deleted | Smalley, Mark A (US 329J-Affiliate) | 4/18/25 12:56:00 PM |
| Page 25: [11] Deleted | Smalley, Mark A (US 329J-Affiliate) | 4/18/25 12:56:00 PM |
| Page 25: [11] Deleted | Smalley, Mark A (US 329J-Affiliate) | 4/18/25 12:56:00 PM |
| Page 25: [11] Deleted | Smalley, Mark A (US 329J-Affiliate) | 4/18/25 12:56:00 PM |
| Page 25: [11] Deleted | Smalley, Mark A (US 329J-Affiliate) | 4/18/25 12:56:00 PM |
| Page 25: [11] Deleted | Smalley, Mark A (US 329J-Affiliate) | 4/18/25 12:56:00 PM |
| Page 25: [11] Deleted | Smalley, Mark A (US 329J-Affiliate) | 4/18/25 12:56:00 PM |

| Page 25: [12] Deleted | Smalley, Mark A (US 329J-Affiliate) | 4/18/25 12:56:00 PM |
| Page 25: [12] Deleted | Smalley, Mark A (US 329J-Affiliate) | 4/18/25 12:56:00 PM |
| Page 25: [12] Deleted | Smalley, Mark A (US 329J-Affiliate) | 4/18/25 12:56:00 PM |
| Page 25: [12] Deleted | Smalley, Mark A (US 329J-Affiliate) | 4/18/25 12:56:00 PM |
| Page 25: [12] Deleted | Smalley, Mark A (US 329J-Affiliate) | 4/18/25 12:56:00 PM |
| Page 25: [12] Deleted | Smalley, Mark A (US 329J-Affiliate) | 4/18/25 12:56:00 PM |
| Page 25: [12] Deleted | Smalley, Mark A (US 329J-Affiliate) | 4/18/25 12:56:00 PM |
| Page 25: [13] Deleted | Smalley, Mark A (US 329J-Affiliate) | 4/18/25 12:56:00 PM |
| Page 25: [13] Deleted | Smalley, Mark A (US 329J-Affiliate) | 4/18/25 12:56:00 PM |
| Page 25: [13] Deleted | Smalley, Mark A (US 329J-Affiliate) | 4/18/25 12:56:00 PM |
| Page 25: [13] Deleted | Smalley, Mark A (US 329J-Affiliate) | 4/18/25 12:56:00 PM |
| Page 25: [13] Deleted | Smalley, Mark A (US 329J-Affiliate) | 4/18/25 12:56:00 PM |
| Page 25: [14] Deleted | Smalley, Mark A (US 329J-Affiliate) | 4/18/25 12:56:00 PM |
| Page 25: [14] Deleted | Smalley, Mark A (US 329J-Affiliate) | 4/18/25 12:56:00 PM |
| Page 25: [14] Deleted | Smalley, Mark A (US 329J-Affiliate) | 4/18/25 12:56:00 PM |
| Page 25: [14] Deleted | Smalley, Mark A (US 329J-Affiliate) | 4/18/25 12:56:00 PM |
| Page 25: [14] Deleted | Smalley, Mark A (US 329J-Affiliate) | 4/18/25 12:56:00 PM |
| Page 27: [15] Deleted | Smalley, Mark A (US 329J-Affiliate) | 4/18/25 12:56:00 PM |
| Page 28: [16] Deleted | Smalley, Mark A (US 329J-Affiliate) | 4/18/25 12:56:00 PM |

| Page 29: [17] Deleted | Smalley, Mark A (US 329J-Affiliate) | 4/18/25 12:56:00 PM |

| Page 29: [18] Deleted | Smalley, Mark A (US 329J-Affiliate) | 4/18/25 12:56:00 PM |

1.

---

## Author Comment (AC2)

The authors thank the reviewers for their helpful comments. Although the observational instruments and general goals remain the same, we have made many changes to the manuscript, including a fundamentally different approach to identifying cold pools using the surface station time series.

Response to Reviewer #2: https://doi.org/10.5194/egusphere-2024-1098-RC2

**General Comments**

The study "A climatology of cold pools distinct from background turbulence at the Eastern North Atlantic observations site" by Smalley and co-authors in consideration for publication in ACP introduces a new method to detect passages of oceanic cold pools in observations of a surface-based meteorological station. The detection algorithm is developed and validated using an 8-year measurement record of the Eastern North Atlantic site on Graciosa Island (Azores) and is based on bivariate fluctuations in temperature-moisture space above a background state that depends on the annual and diurnal cycle. The authors spent a substantial share of the manuscript on describing the concepts and derivation of the method and its distinction from a simpler threshold-based method, followed by a discussion on the properties of the detected cold pools by analyzing various meteorological parameters.

The study introduces a cold pool detection method that utilizes a (to my knowledge) novel concept that is of potential interest for the cold pool and convective organization community. However, the study has several substantial weaknesses, both with respect to the scientific quality and the presentation style, which can be summarized as follows:

- The concept of "background turbulence" is unclear. Although it is of central importance for the study, the authors do not state what spatial and temporal scales this term refers to (especially in the context of typical cold pool sizes and lifetimes) and which processes contribute to it. Furthermore, I am skeptical that the accuracy of the temperature data (0.1 K) allows to reliably distinguish between cold pool signals and background noise and that the temporal resolution of the data (1-min data smoothed over an 11-minute period) is sufficient to capture circulations that are typically relevant for boundary layer turbulence.

The term "background turbulence" has been changed to "background variability" or "background thermodynamic variability" and refers to any fluctuations in temperature that are not attributable to cold pools. We note that the Vogel et al. (2021) method also employs an 11-minute moving average (the same as our $T_{11}$) to avoid small-scale fluctuations that are likely not attributable to cold pools. We now make that more clear in the manuscript.

- The introduced detection method is insufficiently described. After reading through the methods section several times, it is still unclear to me how exactly the distinction between cold pool and background values in temperature-moisture space is done and what the role of the weight parameter is compared to the semimajor axis of the derived ellipse.

We have devised a new method, which is simpler and relies on thresholds for three metrics that vary throughout the time of day and time of year. The new method is described in Section 3.2-3.4.

- The benefit of the detection method is unclear. Although the authors correctly acknowledge the lack of a benchmark data set for validation of the cold pool detections, any statements on the performance of the algorithm, especially compared to existing and much simpler methods, remain vague.

Thank you for this suggestion. The lack of a "truth" or benchmark dataset makes validation quite difficult. In addition to a proof-of-concept time series (Figure 7), we now provide (i) scanning Ka band radar for context in the proof-of-concept time series (Supplemental Figure 2), (ii) the connection of cold pools to observed surface rain fraction and intensity composite time series for four regions of the Depth/Gust metric space (Figure 8), (iii) a new analysis of the ratio of the number of detected cold pools made near observed rain events to the number of detected cold pools during times when no rain is observed ($E_{NR}/E_{FFR}$), and (iv) a comparison of that ratio to the values obtained when using constant temperature change thresholds as implemented in other works.

We note that Terai and Wood (2013) provided only a single time series for proof-of-concept (similar to our Figure 7) as validation. Vogel et al. (2021) similarly provide only time series of the relevant surface observations with added profiles of zenith-pointing Doppler lidar vertical velocity and Ka-band radar reflectivity as validation of their retrieval. Both of those papers were published in ACP.

- The study lacks integration into the existing literature. It does not state the open scientific problem that it aims at, does not motivate why this problem is relevant, does not discuss the results in the context of previous studies, and does not explore possible implications of the presented work.

The introduction was expanded and the revised version addresses this comment. In addition, Figure 1 in the revised manuscript visually demonstrates the "problem" of using a constant threshold to identify cold pools at ENA.

- The reader is poorly guided through the manuscript. The authors rarely motivate why a specific analysis is performed and reveal the overall concept of the detection method only at the end of the (lengthy) methodology section.

The motivation for the paper (overall) as well as the aims of each individual section are clarified in the revised manuscript.

In the light of the listed deficiencies, I see the manuscript on the borderline between major revisions and rejection, leaning towards the latter. If the editor decides to not reject the manuscript, the authors need to satisfactorily address all of these critical issues in a revised version of the manuscript. More specific comments are listed below.

**Specific Comments**

1. Line 1: I suggest to replace the term "background turbulence" with something like "background variability/fluctuations in temperature and humidity".

We have implemented your suggestion, thank you.

2. L29: Please add a sentence on the findings of the study.

The Short Summary now reads, "Evaporation of falling rain leads to temporarily cooler and sometimes windier surface conditions (cold pools), which can lead to further convection that alters convective, cloud, precipitation, and radiation properties. We introduce a new method of measuring cold pools from simple surface-based measurements of temperature and wind speed and then then apply it to 9 years of surface station observations in the north Atlantic Ocean. Cold pools at ENA exhibit a prominent annual cycle, peaking in the winter months. Often, surface-reaching rain events are not associated with cold pools due high background thermodynamic variability, reduced surface wind speed, high boundary layer humidity, fully overcast skies, and weak rain rate.".

3. L45: The authors should mention that the sign of the moisture signal is more uncertain than for the other variables (as discussed later in the study).

During the re-write, we delayed mentioning moisture/humidity in this context until the discussion of Figure 2.

4. L47: The "fundamental change in the character of the boundary layer clouds" need to be specified.

The sentence now reads, "Horizontally propagating cold pools may induce upward motion by displacing more buoyant air or by colliding with other outward-propagating cold pools, mechanically forcing air upwards and potentially leading to further convection that can destabilize the cloud layer and lead to stronger rain rates, reduced cloud fraction, and lower scene reflectivity (Feingold et al. 2010).".

Feingold, G., I. Koren, H. Wang, H. Xue, and W. Brewer, 2010: Precipitation-generated oscillations in open cellular cloud fields. Nature, 466, 849–852.

5. L49: Please state here or elsewhere that this study focuses on oceanic/marine cold pools. There is a whole body of literature on continental cold pools that is not mentioned at all but also relevant for the study.

We added a clause (L52-53) stating that "we observe cold pools passing over the Azores archipelago, which is a maritime environment frequented by…"

6. L50: Since the study does not specifically exclude strong cold pools or deep convective case from the analysis, it is not fair to say that the study focuses on weak cases. The presented method is rather capable to detect a wider range of cold pool intensities. This also applies to similar statements elsewhere in the manuscript.

The reviewer raises a good point here. There are certainly some strong cold pools from deeper and organized convection in the dataset. We have changed the wording to be, "While much of the existing literature on cold pools has been in the context of impacts on deep convection (Feng

et al., 2015; de Szoeke et al., 2017), here we observe cold pools passing over the Azores archipelago, which is a maritime environment frequented by boundary layer clouds and their precipitation in addition to occasional deeper and organized systems (Giangrande et al., 2019). These lighter rains, largely warm rain from shallow Cu and Sc, are expected to produce small temperature and wind signals."

7. L52: Please explain the term "boundary layer rain". Is it a synonym for weak convection?
Yes, it is a synonym for weak convection. We have updated the text to say, "Several previous studies have examined the cold pools that form when rain falls primarily within the boundary layer during shallow convection" (Lines 55-56).

8. L63-71: Since the term "(background) turbulence" is a central concept for the study, it is essential to properly define it here in the context of the following analyses. Which processes contribute to the background turbulence? What spatial and temporal scales are included? How do these scales compare to typical sizes and life times of (oceanic) cold pools? Clarifying these issues is critical to assess the usability and performance of the introduced method and to discuss the retrieved findings.
We now refer to this concept as "background thermodynamic variability" and we have added the following description to Lines 77-78, "…background thermodynamic variability, which we define as any naturally occurring fluctuations in thermodynamics and winds that are *not* due to cold pool activity".

9. L81: Further describe the measurement location (e.g., distance to coast line) as these information can be relevant for interpreting the observations.
We have added the following information about ENA to the first paragraph in Section 2. "The site is situated in the northwestern portion of the island, about 30 m above sea level and about 400 m to the south of the nearest coastline, which is composed of rocky cliffs. The surface near the site is relatively flat and containing fields, residential areas, and the Aeródromo da Graciosa airstrip."

10. L89: Based on Supplement Fig. 1a, the instrumental noise of the temperature measurements calculated over 30-min periods is close to 0.1 K/min. For shorter periods we can expect it to be even stronger and, therefore, of substantial magnitude compared to the cold pool temperature signals shown later. The authors should discuss if and to what extent the instrumental noise could impact the presented results.
The signals shown in Supplemental Figure 1a reflect measured variability from all sources, which includes but is not limited to measurement noise and true atmospheric variations. We have added text to Section 2 to explain this reasoning. The relevant portion is now, "The temperature and wind speed time series are derived from a serialized combination of the MET and MAWS systems due to elevated noise in the MET temperature deviations during extended time periods. Preceding about 2015 Oct 01 and between about 2017 May 01 and 2024 June 01, the minute-to-minute variations in the 1-minute MET temperature data (shown in Supplemental Figure 1) exhibit a considerable increase in variability that we believe is unlikely to be related to natural phenomena and likely represents anomalous instrument noise. This noise is too large to be useful in diagnosing small changes due to cold pools from light rain and virga. Therefore, in this work

we utilize the 1-minute temperatures and wind speeds from MET between 2016 Jan 01 to 2019 April 01 and then MAWS from 2019 April 01 until 2024 Dec 31.".

11. L93-96: As correctly stated by the authors, the measurement resolution of 0.1 K is relatively large compared to the studied cold pool signals. However, the applied temporal smoothing does not increase the information content of the data beyond the instrumental limits (even though the numerical values have a resolution below 0.1 K) but rather removes information on the short-term variability.

The text did not intend to imply that information was somehow added by averaging. The paragraph now reads, "We note that the MAWS temperatures are provided at 0.1 K intervals, considerably greater than the MET temperature intervals of 0.01 K. Such coarse MAWS intervals is large compared to some of the cold pool signatures examined here. Similar to other previous works (Vogel et al., 2021), we perform temporal averaging to many of the surface station time series observations (explained later). This averaging reduces the influence of instrument noise while simultaneously eliminating small-scale variability that is unlikely to be related to cold pool signatures. As an additional benefit to our work with the MAWS temperature data, this smoothing allows the MAWS data to take values at finer intervals than the original 0.1 K intervals. The general conclusions of this work are not affected by switching to MAWS on 2017 May 01, indicating that the 0.1 K MAWS intervals does not strongly affect the cold pool detections.". Also, please see our response to your Specific Comment #25, which addresses the effects of instrument sensitivity on the seasonal and diurnal cycles of detected cold pools.

12. L110: Describe how the gaps in the rain measurements are filled.

The sentence now reads, "…rain detections separated by less than 5 minutes were merged into single events to account for the high spatial and temporal variability of surface reaching precipitation".

13. L114: What is meant by "very small-scale turbulence"? From a meteorological perspective, I would probably think about spatial scales of 1 m or even below. With a temporal resolution of 1 min and a typical wind speed of 5 m/s, one can resolve flow features with a size of about 300 m, which would be referred to as a "large eddy" in the context of PBL turbulence. This again suggests to avoid the term "turbulence" in this study.

See our reply to the first Major Comment. We now refer more generally to "thermodynamic variability" throughout the revised manuscript.

14. L119/120: What is the original resolution of the Doppler Lidar data?

The relevant text now reads, "We also utilize the vertically-pointing Doppler lidar (Newsom et al. DLFPT), which provides retrievals of vertical air motion at about 1 s/30 m spacings. To match the MET and MAWS time points, we average the DLFPT vertical motions to 1-minute/30 m spacings.".

15. L132: At this point, it is unclear why the following analyses are performed. In the previous sentence, the authors state that rainfall data is not required for the detection algorithm but analyze rain events in the following paragraph.

This text was changed due to the change in method.

16. Fig. 1: I have several issues with this figure:
    o What temperature and moisture signals are shown in Fig.1? Every 1-min signal during the respective rain event or the first one, or the strongest one?
    o The data presentation in polar coordinates for panels a to c is quite confusing. Do the color-coded tiles refer to the actual area they cover? I suggest to use a scatter plot or 2D density plot (in Cartesian coordinates) instead.
    o It is very hard to compare the results of the panels due the different relative analysis times. It also unclear why these times where chosen.

This figure is no longer included due to the change in method.

17. L136-137: Another possible explanation for negative moisture signals is the downward transport of drier free-tropospheric air into the surface layer.

This portion was removed during the re-write. Figure 2 now shows persistent moisture increases near the surface from sondes.

18. L149: Since the data is smoothed, the time scale is larger than 1 min.

This sentence was removed during the re-write.

19. L149: How many NearRain minutes does the data set contain (compared to FarFromRain minutes)?

We now state this explicitly at the end of Section 2. "Both possibilities would contaminate the FarFromRain category with larger temperature changes and gusts. In total, there are 286,766 individual FarFromRain minutes (6.2 % of the valid record) and 995,134 NearRain minutes (21.4 % of the valid record)."

20. L152-153: The reader should be informed first about what is done rather than what is not.

We now include a paragraph near the end of the Introduction to briefly describe the algorithm basics.

21. L171: This statement is not justified so far.

This statement is justified by Figures 1c and 1d, which show that the diurnal cycle of background variability contaminates the annual and diurnal depictions of cold pool frequencies. Figure 1e shows there is no clear break between background variability and cold pools. If there were clear and consistent differences between the cold pools and the background variability, the number of cold pools obtained using a constant threshold would flatten in the region between the cold pool distribution and the background variability distribution. This is further shown (later) by the histograms of the Depth metric in Figure 3a and 3d.

22. L173: The authors should mention that there are several other studies on continental cold pools, that also used threshold-based detection methods often combined with other data sources (e.g., Engerer et al, 2008, MWR; Redl et al, 2015, MWR; Provod et al, 2016, MWR; Kirsch et al, 2021, MWR; Kruse et al, 2022, QJRMS)

We added the requested references to Line 213 to make the point that fixed thresholds may be more appropriate for more vigorous convection and over land, when observed temperature drops are stronger in magnitude than those typically observed at ENA. The Engerer et al., (2008) paper

manually defines the cases first by eye and then measures the temperature drop during those selected cases, rather than objectively determining the existence of a cold pool, so we add its reference to the Introduction.

23. L175: It is unclear how exactly the method of Vogel et al (2021) is applied to the present data set. According to the notation convention used throughout the manuscript, I assume that dT/dt denotes the perturbation of the raw temperature data (as opposed to the smoothed data T'). Moreover, does the threshold need to be continuously exceeded over 20-min periods as in Vogel et al (2021) or only for individual 1-min time steps (which would be very problematic)? The authors have to clarify this.

The Vogel algorithm does use T (11-minute moving window average temperature) instead of T' 11-minute moving window average temperature minus 121-minute moving window average temperature). This should be clearer now that we moved the Vogel results to Figure 1, before we introduce T'. Thank you for helping to clarify.

According to Vogel et al. (2021; their Section 2.3), there is no requirement of the temperature to decrease more quickly than the threshold for any amount of time more than a single minute. They identify candidates as times when T decreases faster than the threshold and then identify the beginning of the cold pool candidate as the most recent time that T increases. If that time is more than 20 minutes before the temperature decreases faster than the threshold, the cold pool start is defined as 20 minutes before that time.

24. L177-182: I understand the statistical argument that the authors make on the cold pool detection rate in rainy vs. rain-free periods, however, I think that it could be strengthened by directly comparing the detection rates of both methods for different periods (since there is no reference data set). Also, it is unclear how large the absolute numbers of (presumably) false detections are compared to the full data set, i.e. how strongly they actually impact the performance of the simple method.

We now assess the ratio of detection rates during NearRain periods to the detection rates during FarFromRain periods as the ratio $E_{NR}/E_{FFR}$ in Figure 1f and in the text just after Figure 7 and in Section 3.1. We prefer this ratio to be high, indicating frequency cold pools near rain and rare detections when there is no rain nearby.

25. Fig. 2: The detection thresholds are below or around the measurement resolution of the observations. How large is the impact of instrumental noise in this analysis?

Answering this fully would require a full investigation using instruments that do not exist at ENA. For example, we would like to perform the full analyses using the MAWS and MET measurements separately, but there is not enough data in either record to reliably define the month/hour thresholds for the three metrics. In absence of that data, we recompute our new "best" algorithm for four different scenarios, representing combinations of whether the station temperature series is averaged to 0.1 K intervals…

1. "FineCoarse Early" means we allow the MET and MAWS data to exist at its native intervals of 0.01 K and 0.1 K, respectively, while switching from MET to MAWS at 2017 May 05 0000 UTC
2. "FineCoarse Late" is the same as "FineCoarse Early" except we switch from MET to MAWS at 2019 April 01 0000 UTC

3. "CoarseCoarse Early" is the same as "FineCoarse Early" except all temperatures are averaged to 0.1 K intervals.
4. "CoarseCoarse Late" is the same as "CoarseCoarse Early" except we switch from MET to MAWS at 2019 April 01 0000 UTC

The figures below demonstrate that the estimated annual and diurnal cycles are not strongly affected by the choices made regarding instrument temperature intervals at ENA.

[Figure]

26. Fig. 3: Show the time of the day in local time rather than UTC, which eases the interpretation of the results.

The local time is UTC-1, a small enough difference that we do not believe interpretation of results is impacted. We now specifically state that the time zone is UTC-1 in Section 2.

27. L209-211: This sentence again refers to the method, which was not yet properly described. The reader can only guess how exactly the method actually works.

This section has been rewritten. The relevant discussion has been moved to the analysis of Figure 1, which is sufficiently separated from the discussion of our method and results.

28. L217-225: This paragraph is very hard to follow. For example, it is unclear to me what are eddy depth and eddy strength as opposed to eddy size. Since the described issue seems to be of minor importance, I suggest to delete the paragraph.

This paragraph is now omitted.

29. Fig. 4: Please explain the dots shown in panels a and b. I guess they correspond to the ellipses shown in panels c to f. Why were these examples selected?

The method has changed and this figure is no longer included.

30. L261: What does "greater than the line extending along the ellipse's major axis" mean? Does this mean that the point has to lie outside of the ellipse? How many of the points do actually lie outside of the ellipse (if at all)? Since this is the core of the introduced method, the authors need to be very clear on how it works!

The method has changed and this figure is no longer included.

31. L270: Explain the term "semimajor axis".

An ellipse's semimajor axis is the maximum length of a straight line segment that runs between the ellipse's center and any point on its perimeter. As the method has changed, we do not need to explain the meaning in the paper.

32. L275: The authors should explicitly state that the semimajor axis is a measure of the cold pool strength (as this is implicitly assumed later on).

The method has changed and the ellipse is no longer relevant.

33. L279: How do the authors know that these are actual cold pools?

See our answer to the relevant General Comment.

34. L280: Again, the authors do not sufficiently motivate why the following analyses are performed.

See our answer to the relevant General Comment.

35. L282: How are false alarms identified? What is the reference?

We use a combination of *in-situ* surface rain measurements from the ORG, PWD, and LD as well as IMERG Final precipitation retrievals. We now explain this in Lines 163-177.

36. L286: I suggest to call them "detected" rather than "true" cold pools, since there is no reference data set (as also stated later in the manuscript, L533)

Thank you. We now use the words "detected" or "retrieved" or "estimated" when referring to candidates that the algorithm determines are cold pools.

37. L290: Is "T" temperature or time?

This equation is no longer in the paper due to the change in cold pool detection method.

38. L302: What is "a" as opposed to "s"? Both seem to somehow represent the semimajor axis.

This equation is no longer in the paper due to the change in cold pool detection method.

39. L308-312: It is unclear to me why the weight W is needed (as it appears to correlate with the semimajor axis) and how it is used to identify cold pools.

The weight communicates how confident the algorithm is that a given candidate is a cold pool. Weights can take values of 0.0, 0.5, or 1.0, depending on which metrics exceed their respective thresholds.

40. L330: This does not become clear from the definition of W in the previous section. Is 1 the maximum value that W can take?

The method of defining weights has changed. The possible values are 0 (confidently *not* a cold pool), 0.5 (potentially a cold pool but not enough evidence to be fully confident), and 1.0 (confidently a cold pool.

41. L338-341: Delete this sentence as the wind analysis does not seem to be of significance for the method.

The method has been changed to include wind speed information.

42. L341-343: I don't understand this sentence.
This sentence was removed during the re-write.

43. Fig. 6, top panels: Explain the solid and dashed rings in the plots.
These panels are now in the supplementary material. The caption of Supplementary Figure 2 now explains that the concentric circles surround the ENA radar site.

44. Fig. 6, bottom panels: I suggest to code the weight as color rather than as line thickness since it is very hard to compare visually.
There are now only 3 possible values so the differences are now easy to see in the figure.

45. L346: Explain the top row first.
This figure no longer contains the radar scans.

46. L347: State the time period in the caption.
The time period is visible in the figure.

47. L354: Define the period.
This text has been changed and the full KaSACR observational period is no longer relevant.

48. L357-375: Given the unjustified and rather arbitrary choice of thresholds in different variable as well as the overall very small sample size, the presented analysis is not convincing in validating the cold pool detection algorithm.
The validation methods have been changed.

49. L379: Can the weight actually become negative?
No, W<0 was never possible. The method of defining weights has changed. The possible values are 0 (confidently *not* a cold pool), 0.5 (potentially a cold pool but not enough evidence to be fully confident), and 1.0 (confidently a cold pool).

50. L379-380: Why do the weights sum up to a number of events? I am afraid I still did not get the concept of the weight.
The weight communicates how confident the algorithm is that a given candidate is a cold pool. If 100 candidates all have a weight of 0.5, the algorithm is only half-confident that each candidate is a cold pool. We expect that about half are cold pools and half are not, given the absence of further information. From that idea, summing the weights leads to an expected number of cold pools (50 cold pools out of 100 candidates in this example).

51. L385: Does "the entire population of cold pool candidates" refer to the full sample (~200.000 cold pools) or only to the retrieved ones (~8500). Either the statement in L383 is wrong or the wording is misleading.
These analyses account for the candidate weights (confidence).

52. L388: Wind gusts are usually defined over 3-second time periods. The used measurements do not have the sufficient resolution to justify statements about gustiness.

We now define our definition of "Gust" as a new metric. When gust is used with a lower case "g", we are simply referring to an anomalous increase in wind speed, not a rigorously defined value.

53. L389-390: Isn't this a trivial statement since a_max measures the perturbation strength in temperature-moisture space?

The method has changed and this is no longer relevant to the manuscript.

54. L393: How are the cold pool strength categories chosen and what are the respective sample sizes?

This figure is no longer in the paper. The relevant new figure is Figure 8. The samples and number of cold pools are listed in the text describing Figure 8. Boundaries were chosen by eye to characterize the four regimes while maintaining a reasonable number of cold pools in each regime.

55. L397: The authors do not motivate why the cold pool size is estimated.

This paper presents a climatology of cold pools at ENA. The size of such objects are relevant to the general statistics that comprise a climatology, in addition to their seasonal and hourly frequency, association with synoptic conditions, and potential changes over time.

56. L408-411: How should the reader interpret this information? How do these numbers compare to the results found by other authors? There is certainly more literature that has already studied the cold size of oceanic cold pools. Why are these number relevant?

We now relate our results to those of Terai and Wood (2014) and Wilbanks et al. (2015) in L498-499.

57. L410: What is the "cord length"?

This was a misspelling of the word "chord", which is no longer relevant because the method changed.

58. L416-475: Figs. 8 to 12 are shown without any motivation and are insufficiently described and discussed. The authors either need to expand this part (see also the following comments) or delete the entire part.

These figures have been removed from the paper due to changes in methodology.

59. L418: What is a "more limited time period"?

The more limited time period was 2016 Jan 01 to 2017 Sep 01, as stated in that sentence. This sentence was removed when the manuscript was rewritten. We switched from the subjective classification to a more objective classification (Figure 9).

60. Fig. 9: It is unclear to me if this figure shows fraction of cold pool events or cloud types or both.

The old Figure 9 and the new Figure 9c show the fraction of cold pool detections that occurred during 6-hour chunks of time that were independently assigned as Sc or Cu. Not all 6-hour segments were assigned as either Sc or Cu, which is why the values for a given Depth do not add to 1.0.

61. L432-433: I would think that the upward motion signature before the arrival of the downdraft is rather caused by lifting ahead of the cold pool front.

This figure (now Figure 11) has been changed for the new manuscript. At your helpful suggestion, we have added the mechanical lifting as a potential driver of upward motion preceding the cold pool.

62. L442: This climatology should be better placed at the beginning of section 3.

We prefer to leave it here, after some characteristics of the cold pools have been presented (Figure 9). Also, note that the manuscript's structure has been revised due to changes in methodology.

63. L466-469: I don't understand this sentence.

This analysis was removed during the re-write.

64. Fig. 12: Indicate the data source for this analysis.

The MERRA-2 source has been added. It must have been accidentally removed during internal edits prior to submission.

65. L482-521: It is unclear why all these variables are discussed and what the conclusions are. Also, the results are not discussed in the context of existing literature.

The section in question has been retitled "Which Rain Events Produce Observable Cold Pools?" to give the reader a clearer idea of why we undertake the analysis that follows. We are not aware of a comparable analysis in the literature to which our results can be compared, as most analysis of cold pools in marine environments have been from either ship or aircraft, rendering them unable to produce a multi-year time series of cold pool occurrence as we do with the ground-based measurements.

66. L493: Further explain the argument about post-frontal scattered showers.

Mid-latitude cyclones over the ocean frequently feature a region of precipitating shallow convection in the cold sector of the storm that follows frontal passage. We added a citation of Lamer et al. (2020) to substantiate the validity of this claim at ENA.

67. L494-495: The passage of a cold pool usually leads to an increase in air pressure, meanings that this is not a condition for which cold pools are more likely to occur (as the wording suggests).

This pressure change is over the course of an hour. We added the sentence, "The 1-hour pressure change was computed as $P(t+30) – P(t-30)$, where P is the air pressure measured by either MET or MAWS and t is the time of the rain event start." to clarify.

68. L496: Does "boundary layer motions" refer to wind?

Yes, that paragraph exclusively analyzed the two wind speed panels. We have changed to language to, "Near-surface horizontal winds are also shown to be connected to cold pool passages at ENA.".

69. L510: Define the boundary layer decoupling index.

This variable is no longer used in the paper.

70. L517: The text does not provide any information about the use of balloon measurements.
That was our mistake. We now include the sentence, "We also utilize the vertically-pointing Doppler lidar (Newsom et al. DLFPT), which provides retrievals of vertical air motion at about 1 s/30 m spacings, and balloon sondes (Keeler et al., SONDEWNPN). To match the MET and MAWS time points, we average the DLFPT vertical motions to 1-minute/30 m spacings. Sonde observations of temperature and specific humidity are averaged to 10 m vertical spacings.".

71. Fig. 14: Panels a, b, g, and h are not mentioned in the text.
All panels for Figures 14 and 15 are now directly referenced in the text.

72. L526-532: Such a short summary of the method concept would be required much earlier in the manuscript.
Thank you for this great suggestion. We now include a paragraph near the end of Section 1 that summarizes the new method and explains what the paper will accomplish.

73. L531: As stated in the following sentence, there is no "truth" or reference data set.
The method has changed and this sentence is no longer included.

74. L545: How do the authors come to the conclusion of an "overly-strong representation of the cold pool diurnal cycle"? Is there any existing literature that justifies this statement?
We justify this statement with Figure 1d.

75. L567: The observed signals in the different variable are rather a consequence of cold pool passages than a condition for their formation.
The cold pools must have previously formed upwind in order to be pass over the site. We have reworded the text as follows, "When rain is first measured by the *in-situ* surface instrumentation, detected cold pools tend to have the following properties…".

76. L579-583: As already mentioned earlier, the authors miss a discussion on the broader implications of the developed algorithm for the convection and cloud organization community.
We appreciate the reviewer's suggestion to include a broader discussion on the implications of our developed algorithm for the convection and cloud organization community. After careful consideration, we have decided not to add such a paragraph for several reasons. First, the manuscript has already undergone substantial expansion during revision, including the development and explanation of an entirely new methodology, which has increased its length considerably. Second, our work is primarily focused on establishing a reliable detection methodology for cold pools at ENA, with the primary contribution being the time-varying threshold approach that can distinguish weak cold pools from background variability. We believe that speculating on broader implications beyond this methodological advancement would extend beyond the scope of our current study and potentially dilute our central findings.

Instead, we have strengthened our existing discussions in Sections 1 and 6 to better contextualize our work within the current literature. In the final paragraph of Section 6 (Conclusions), we note

that our approach could be applied at other locations without subjectively-tuned thresholds, which we feel appropriately indicates the transferability of our method without overreaching our empirical findings. We believe that a more extensive discussion of implications for cloud organization would be better suited for future work that specifically tests such applications across different environmental conditions.

77. L584: Since the development of the introduced detection method is the key innovation of the study, I strongly recommend to make the code publicly available.

The code will be made publicly available prior to publication, as required by the journal.

**Technical Corrections**

1. L13: Please explain "DOE"

We now state the Department of Energy with its acronym.

2. L13: Add a note on where Graciosa Island is located (Azores or geographical coordinates)

We added "in the Azores".

3. L14. Delete "entire"

The word "entire" has been removed.

4. L61: Name the Caribbean Island

Barbados is now named, thank you.

5. L78: I suggest to create a separate section for the data description part.

Done as suggested.

6. L79: Specify that "one-dimensional" refers to time

We now state that we use, "…one-dimensional temperature and wind speed measurements from surface-based in-situ observations.".

7. L80: Add "United States" to DOE

Done, thank you.

8. L81: Add the coordinates of the measurement location

The ENA coordinates have been added.

9. L87: Please shortly explain what a "serialized combination" is.

A serialized combination is one after another. Both are used but one at a time. The reader will easily understand this from the context of discussion switching from one instrument to the other.

10. L88: Change to "before ... and after ..."

"Preceding … and following …" carries the same meaning as the suggestion. Due to the recent reduction in MET noise, we have change the sentence to say, "Preceding about 2015 Oct 01 and between about 2017 May 01 and 2024 June 01…"

11. L109: Better call them "rain events" since the term "objects" is usually associated with spatial entities (and stick to the naming convention throughout the manuscript)

Good suggestion. We implemented it.

12. L112: Delete "often"

Good suggestion. We implemented it.

13. L116-117: The sentence is incomplete.

The sentence now reads, "For example, the temperature anomaly $T' = T_{11} - T_{61}$, where $T_{11}$ and $T_{61}$ are the 11-minute and 61-minute means, respectively."

14. L117: q is usually used for specific humidity. I suggest to use r for the mixing ration instead.

We changed $q_v$ to $w_v$ instead of $w$ to avoid confusion with vertical motion.

15. L117: Add the missing slash in "dq_v'/dt"

Good catch. This was removed due to the change in method.

16. L121: Define the "short period"

We removed "a short period".

17. L122: Which study does "this study" refer to?

We changed the wording to, "Here we use the PPI scans…"

18. L137: Delete "will"

The method changed and this sentence is no longer present.

19. L143: I suggest to rename the "FarFromRain" category to "NoRain"

We choose to retain FarFromRain due to the new need for the ratio $E_{NR}/E_{FFR}$, which is explained in the text. The NearRain acronym would interfere with the NoRain acronym.

20. L153: Change to "weak cold pools"

Done.

21. L155: Change to something like "rain cells that have formed right above the station "

We changed it to, "We expect that recently-formed rain events whose first surface-reaching drops fall upon the surface instrumentation have not yet had enough time to develop a cold pool, leaving them unobservable by the surface instrumentation." because rain falling from a rain cell that formed right above the station might not fall directly down onto the instrumentation due to horizontal winds.

22. L163: Include the shown temperature and moisture perturbations into the caption for Fig. 1a-c.

This figure is no longer in the paper.

23. L163: Change "probability" to "frequency"
This figure is no longer in the paper.

24. L164: Please remind the reader of the analysis period ("this period").
This figure is no longer in the paper.

25. L169: Introduce the "CP" acronym at first use of the term and use it consistently throughout the manuscript.
We have implemented this suggestion.

26. L178: Swap "year" and "day" (so that Fig. 2a is referenced before Fig. 2b)
Done.

27. Fig. 2: Change "expected" to "detected" (in y label)
We retain the use of "expected" because it is a mean frequency per 24 hours of valid observations, similar to an expectation rate.

28. Fig. 2b: Use local time instead of UTC
As addressed above in the response to Minor Comment 26, the ENA site is UTC-1 thus the conversion from UTC to local time is negligible. As a matter of style, we have chosen to keep the times in UTC.

29. L195: Change "desire" to "aim" or "goal"
This sentence was removed due to the change in method.

30. L208: Avoid the word "clear" in this context.
Changed to "evident".

31. L266/267: Delete "of course"
Done.

32. L273-274: Move this information to the respective figure caption.
This sentence was removed due to the change in method.

33. L276-277: Change to "sharply decreases beyond that value"
This sentence was removed due to the change in method.

34. L285: Add "(Eq. 1)"
This equation was removed due to the change in method.

35. L353: meteorological
This sentence was removed due to the change in method.

36. L354: Add "azimuth angle"
This was added to the Line 160.

37. L406: Supplementary Figure 3
Changed, thank you.

38. L418: Stick to one date format.
This sentence was removed due to the change in the observations used in the Sc/Cu discrimination.

39. L425: The period stated here is different from the text (L418).
This figure and sentence was removed due to the change in the observations used in the Sc/Cu discrimination.

40. L443: Delete "strong" and "weaker"
The sentence is now, "The CP annual and diurnal cycles in ENA cold pools are shown in Figure 11.".

41. L444: Replace "cooler months" with "winter months"
This sentence was changed during the re-write.

42. L460: Replace "UTC" with "time of day"
We now specifically state that the time zone at ENA is UTC-1.

43. L529: Add "ellipse in temperature-moisture space"
This sentence was removed due to the change in method.

44. L568: strong 1-hour
This sentence was changed during the re-write.

45. L579: change "small" to "weak"
Done throughout the paper when referring to cold pool signatures.

**A climatology of cold pools distinct from background thermodynamic variability at the Eastern North Atlantic observations site**

Mark A. Smalley[1,2], Mikael K. Witte[1,2,3], Jong-Hoon Jeong[1], Maria J. Chinita[1,2]

[1]Joint Institute for Regional Earth System Science and Engineering, University of California, Los Angeles, Los Angeles, California
[2]Jet Propulsion Laboratory, California Institute of Technology, Pasadena, California
[3]Naval Postgraduate School, Monterey, California

*Correspondence to*: Mark A. Smalley (mark.a.smalley@jpl.nasa.gov)

© 2025. California Institute of Technology. Government sponsorship acknowledged.

**Abstract**

We identify cold pools at the US Department of Energy (DOE) Atmospheric Radiation Measurement (ARM) Eastern North Atlantic (ENA) facility on Graciosa Island in the Azores and examine the statistics of retrieved cold pools from 2016 to 2024. The retrieval leverages 1-minute deviations in near-surface temperature and wind speed from the ENA surface meteorological station time series to identify cold pool events that exceed the association between background thermodynamic variability and observed distinct precipitation events. Cold pools at ENA exhibit a prominent annual cycle, peaking in the winter months. Although there is a slight increase in rain events during the daytime, we find a decrease in daytime cold pools that are separable from background variability compared to nighttime because of the increased background variability during sunlit hours. Often, surface-reaching rain events are *not* associated with cold pools due to factors including but not limited to high background thermodynamic variability, reduced surface wind speed, high boundary layer humidity, fully overcast skies, and weak rain rate. Understanding the factors that lead to the formation of measurable cold pools will lead to a greater understanding of the dynamics of the marine boundary layer and their influence on cloud morphological structures.

**Short Summary**

Evaporation of falling rain leads to temporarily cooler and sometimes windier surface conditions (cold pools), which can lead to further convection that alters convective, cloud, precipitation, and radiation properties. We introduce a new method of measuring cold pools from simple surface-based measurements of temperature and wind speed and then then apply it to 9 years of surface station observations in the north Atlantic Ocean. Cold pools at ENA exhibit a prominent annual cycle, peaking in the winter months. Often, surface-reaching rain events are *not* associated with cold pools due high background thermodynamic variability, reduced surface wind speed, high boundary layer humidity, fully overcast skies, and weak rain rate.

**1 Introduction**

State of the art earth system models continue to struggle to simulate the geometrically thin but highly reflective stratocumulus (Sc) clouds that frequently cover the eastern portion of subtropical oceans (Jiang et al., 2021), with much of the uncertainty in the magnitude and sign of global cloud feedbacks being traced to the representation of Sc clouds (Klein et al., 2017; Scott et al., 2020; Myers et al., 2021; Ceppi and Nowack, 2021). The transition of boundary layer clouds from fully overcast stratocumulus to lower cloud fraction shallow cumulus (Cu) following the trade winds results in a strong increase in the amount of radiation absorbed by the ocean surface (Goren and Rosenfeld, 2014) and a reduction of solar radiation reflected (Hartmann and Short 1980; Wood 2012). The breakup of these expansive stratocumulus clouds is largely determined by meteorology and sea surface temperature (Bretherton and Wyant, 1997; Wyant et al., 1997), but recent works (Eastman and Wood, 2016; Yamaguchi et al., 2017; Goren et al., 2019; Blossey et al., 2021; Smalley et al., 2021) have emphasized the role of precipitation in modulating the timing of the cloud regime transition. It is therefore imperative that we understand the physical connections between precipitation and marine boundary layer cloud regime change.

Falling precipitation begins to evaporate as it encounters subsaturated air below the cloud layer. This evaporation cools the air, reducing its buoyancy and leading to downdrafts. Once these downdrafts reach the surface, they spread horizontally and form what is defined as a cold pool (CP) or density current (Wilbanks et al., 2015). Compared to the surrounding near-surface air, CPs are characterized by decreases in temperature and increases in wind speed. Horizontally propagating CPs may induce upward motion by displacing more buoyant air or by colliding with other outward-propagating CPs, mechanically forcing air upwards and potentially leading to further convection that can destabilize the cloud layer and lead to stronger rain rates, reduced cloud fraction, and lower scene reflectivity (Feingold et al. 2010).

While much of the existing literature on CPs has been in the context of impacts on deep convection (Engerer et al., 2008; Feng et al., 2015; de Szoeke et al., 2017), here we observe CPs passing over the Azores archipelago, which is a maritime environment frequented by boundary layer clouds and their precipitation in addition to occasional deeper and organized systems (Giangrande et al., 2019). These lighter rains, largely warm rain from shallow Cu and Sc, are expected to produce weak temperature and wind signals. Several previous studies have examined the CPs that form when rain falls primarily within the boundary layer during shallow convection. Terai and Wood (2013) leveraged time series of potential temperature from sub-cloud aircraft measurements over the subtropical southeastern Pacific Ocean to retrieve CP signatures. They found that small decreases in potential temperature (stronger than -0.36 K) were associated with weak gust fronts and increases in the concentrations of both coarse mode aerosols and dimethyl sulfide, which is relevant for secondary particle formation in the atmosphere. Wilbanks et al. (2015) used measurements of air density instead of potential temperature and similarly found that temperature decreases were strongly associated with their retrieved density currents; their constant threshold for changes in air density roughly corresponds to a total change in temperature of -0.24 K throughout the CP duration. In manually selected cases of precipitating post-cold frontal marine stratocumulus clouds, Ghate et al. (2020) used vertically pointing Doppler radar and lidars to analyze 76 drizzle shafts. They found that downdraft strength correlated with cloud-base drizzle intensity. Vogel et al. (2021) analyzed near-surface temperature variations in

Barbados, identifying CPs based on a fixed temperature change threshold of -0.05 K/min. They demonstrated associations between precipitation duration, retrieved CP strength, and cloud regime categorizations.

In this work we seek to estimate the total number of CPs and their diurnal and seasonal patterns. To do so, we must distinguish between thermodynamic variability caused by CPs and thermodynamic variability caused by other phenomena. Though Terai and Wood (2013), Wilbanks et al. (2015), and Vogel et al. (2021) removed the influence of small-scale variability that is not associated with precipitation-driven CPs by time-averaging surface meteorological station observations and requiring perturbations to exceed constant thresholds, ENA boundary layer variability spans a spectrum of spatial/temporal scales and varies strongly with time of day and season. Using a constant threshold at ENA can therefore lead to false CP detections during periods of enhanced variability, particularly during daytime hours when solar heating drives stronger mixing. Cold pool signatures also exist with a spectrum of intensities, which leaves methods that employ constant thresholds both unable to capture the weakest CPs and at the same time unable to remove all the signatures of the background thermodynamic variability, which we define as any naturally occurring fluctuations in thermodynamics and winds that are *not* due to CP activity. In addition, the background thermodynamic variability can be expected to exhibit annual and diurnal cycles, which are themselves variable with geographic location. We therefore seek a method to retrieve CPs with surface instrumentation in the presence of, but separable from, the background thermodynamic variability in a way that is flexible to diurnal and annual cycles.

Here, we present a method to measure CPs from surface-based observation sites with an emphasis on removing the signals of the background thermodynamic variability that are *not* associated with precipitation-driven CPs. While existing methods detect CPs by simply requiring the near-surface temperature to decrease by more than a subjectively prescribed amount over the source of a single minute, the new method expands upon that approach by (1) using three temperature and wind metrics to characterize CP candidates, (2) assigning thresholds for each metric based on statistical association between the frequency of distinct rain events and increasing magnitudes of the three metrics, and (3) allowing those thresholds to vary throughout the annual and diurnal cycles, thus accounting for time-varying background thermodynamic variability that is not associated with CPs. We will show the results to this method when applied to 9 years of surface observations from the Azores archipelago, which experiences diverse sub-tropical and mid-latitude weather conditions (Giangrande et al., 2019), as well as this method's advantages over method that employ a single constant threshold on a single metric.

We first describe the relevant observations in Section 2. In Section 3 we introduce and validate the algorithm using scanning radar, surface rain observations, and satellite precipitation observations. We then apply the algorithm to 9 years of surface station observations representing a short climatology in Section 4. Section 5 provides an analysis of which confirmed contiguous rain events lead to CPs, and Section 6 provides a discussion of our findings.

**2 Observations from the ARM ENA site**

The goal of this work is to confidently diagnose cold pools (CPs) via one-dimensional temperature and wind speed measurements from surface-based *in-situ* observations. We select the United States Department of Energy (DOE) Atmospheric Radiation Measurement (ARM) program's Eastern North Atlantic (ENA; 39° 5′ 29.76″ N, 28° 1′ 32.52″ W) observatory on

Graciosa Island in the Azores, which is situated at the northern edge of a subtropical marine weather regime but is also impacted by mid-latitude synoptic systems (Remillard and Tselioudis, 2015; Mechem et al., 2018; Giangrande et al., 2019). The site is situated in the northwestern portion of the island, about 30 m above sea level and about 400 m to the south of the nearest coastline, which is composed of rocky cliffs. The surface near the site is relatively flat and containing fields, residential areas, and the Aeródromo da Graciosa airstrip. The Graciosa Island time zone is UTC-1.

The ENA site contains a comprehensive suite of surface-based instrumentation, but this work mainly utilizes observations of near-surface temperature and wind speed from the surface meteorological station (MET; *enametC1.b1* datastream; Kyrouac and Shi 2011) and the Meteorological Automated Weather Station (MAWS; *enamawsC1.b1* datastream; Keeler et al., 2017) to retrieve CPs. The temperature and wind speed time series are derived from a serialized combination of the MET and MAWS systems due to elevated noise in the MET temperature deviations during extended time periods. Preceding about 2015 Oct 01 and between about 2017 May 01 and 2024 June 01, the minute-to-minute variations in the 1-minute MET temperature data (shown in Supplemental Figure 1) exhibit a considerable increase in variability that we believe is unlikely to be related to natural phenomena and likely represents anomalous instrument noise. This noise is too large to be useful in diagnosing weak changes due to CPs from light rain and virga. Therefore, in this work we utilize the 1-minute temperatures and wind speeds from MET between 2016 Jan 01 to 2019 April 01 and then MAWS from 2019 April 01 until 2024 Dec 31.

We note that the MAWS temperatures are provided at 0.1 K intervals, considerably greater than the MET temperature intervals of 0.01 K. Such coarse MAWS intervals is large compared to some of the CP signatures examined here. Similar to other previous works (Vogel et al., 2021), we perform temporal averaging to many of the surface station time series observations (explained later). This averaging reduces the influence of instrument noise while simultaneously eliminating small-scale variability that is unlikely to be related to CP signatures. As an additional benefit to our work with the MAWS temperature data, this smoothing allows the MAWS data to take values at finer intervals than the original 0.1 K intervals. The general conclusions of this work are not affected by switching to MAWS on 2017 May 01, indicating that the 0.1 K MAWS intervals does not strongly affect the CP detections.

Observations of surface precipitation at ENA are taken from a combination of three sources: the (1) optical rain gauge (ORG), (2) present weather detector (PWD), and (3) laser disdrometer measurements (LDIS; *enaldC1.b1* datastream; Wang et al., 2023). The ORG and PWD rain measurements are included in the MET data product referenced above. A given minute is labeled as raining if any of the three sensors report a positive rain rate. The three sensors have extended non-intersecting data gaps, supporting their combined usage during the entire 2016-2024 period.

The time series begins 2016 Jan 01 and continues to provide observations at the time of this manuscript's preparation, resulting in 9 years of observations from which to diagnose both background thermodynamic variability and precipitation-induced cold pools. The dataset holds a total of 4,643,927 individual valid one-minute observations (approximately 98% of the 9-year data record), within which we search for CPs. While we do not directly require the presence of surface-reaching precipitation to detect a specific CP, we construct the algorithm and aid interpretation of CP results by constructing "distinct rain events" from contiguous raining minutes, as determined by the any of the ORG, PWD, or LDIS sensors reporting any rain rate

greater than 0 mm/hr. We identify 29,269 distinct rain events during the 9-year period of record, where rain detections separated by less than 5 minutes were merged into single events to account for the high spatial and temporal variability of surface reaching precipitation.

Here we assume that the annual and diurnal cycles of CP frequency should be correlated with the analogous cycles of distinct rain events. Figure 1a and Figure 1b show these cycles for different observed maximum rain rates during each distinct rain event. The annual cycle peaks between October-March, with reduced frequency of rain events during the boreal summer. Similar results for ENA were found by Lamer et al. (2020), Wu et al. (2020), and Ghate et al. (2021). The diurnal cycle has a less-pronounced peak from about 0800 UTC to 1500 UTC in the late morning and early afternoon. While not all rain events lead to CPs, we expect these general annual and diurnal cycles to be represented in the CP statistics, with small discrepancies based on the separability of CP signatures from non-CP sources of background variability, which varies throughout the year and day.

Throughout this work, we refer to time series variables with an apostrophe to indicate that the quantity is anomaly of the 11-minute moving mean minus the 61-minute moving mean of that variable, where the means are computed symmetrically around each minute. For example, the temperature anomaly $T' = T_{11} - T_{61}$, where $T_{11}$ and $T_{61}$ are the 11-minute and 61-minute means, respectively. Similarly, we define the wind speed anomaly $S' = S_{11} - S_{61}$. The 11-minute moving mean is intended to remove spurious signals resulting from very small-scale variability and instrument noise, while the subtraction of the 61-minute mean is to remove short-term manifestations of the diurnal cycle and provide a common ground for larger-scale changes in temperature and wind speed anomalies that occur during different weather regimes (i.e., frontal passage). Changes in temperature anomaly over the course of one minute are denoted as $\Delta T'/\Delta t$ and carry the units K/min.

We also utilize the vertically-pointing Doppler lidar (Newsom et al. DLFPT), which provides retrievals of vertical air motion at about 1 s/30 m spacings, and balloon sondes (Keeler et al., SONDEWNPN). To match the MET and MAWS time points, we average the DLFPT vertical motions to 1-minute/30 m spacings. Sonde observations of temperature and specific humidity are averaged to 10 m vertical spacings. For validation, we examine reflectivity factors from the Ka-band scanning ARM cloud radar (KaSACR; Kollias et al., 2016), which was implemented during the ACE-ENA campaign (Wang et al., 2022) and provided plan-position indicator (PPI) and range-height indicator (RHI) scans. Here we use the PPI scans (between azimuthal angles of 287° and 90°) at the lowest elevation (0.5°) below the cloud base to show the location and intensity of falling precipitation.

To support the algorithm, we utilize a combination of the three surface precipitation estimates described above (ORG, PWD, and LD), in conjunction with precipitation retrievals from the 0.1° gridded half-hourly IMERG Final (Huffman et al., 2020; Huffman et al., 2023), which gathers precipitation observations primarily from microwave radiometers but is supplemented/interpolated with infrared when no microwave observations are available. We use only the IMERG Final values that are within ~30 km of the ENA site, a plausible range of influence for precipitation and CPs. We focus on times when rain is observed either at or near the surface instrumentation (NearRain) versus times when we can be very confident that a CP *should not* exist (FarFromRain). For NearRain times, rain is observed by the surface instrumentation within ±1 hour, indicating rain in the immediate vicinity that could lead to CP signatures. For FarFromRain times, zero surface rain must be observed within ±6 hours by the ORG, PWD, and LD and no rain can be reported by IMERG Final within ±2 hours. We assume that the

FarFromRain category should not contain CP signatures. However, we acknowledge this assumption's limitations, as propagating CPs from terminated rain events plausibly travel distances greater than 30 km with the mean boundary layer flow and shallow and isolated light rain near the surface instrumentation may go undetected by IMERG Final. Both possibilities would contaminate the FarFromRain category with larger temperature changes and gusts. In total, there are 286,766 individual FarFromRain minutes (6.2 % of the valid record) and 995,134 NearRain minutes (21.4 % of the valid record).

**3 Cold Pool Detection Algorithms**

**3.1 Cold pool detection using a constant temperature change threshold**

Previous works have utilized changes in temperature T alone to diagnose the existence of CPs from surface measurements at the Barbados Cloud Observatory (Vogel et al., 2021) and airborne data from the VOCALS field campaign (Terai and Wood, 2013). But are these constant-threshold methods effective at ENA? Figure 1 demonstrates considerable shortcomings of the constant-$\Delta T/\Delta t$ method when implemented at ENA by reproducing the Vogel et al. (2021) algorithm for a variety of different constant-$\Delta T/\Delta t$ thresholds. In Figures 1c and 1d, retrieved CP frequencies are plotted against month and UTC for comparison against Figures 1a and 1b. The disparities in the annual and diurnal cycles between CPs obtained using constant thresholds and observed rain event frequencies are evident, especially when the threshold $\Delta T/\Delta t>$-0.10 K/min (Figure 1c and 1d). These detections peak strongly during the sunlit hours (Figure 1c), when solar radiation leads to increased variability in the boundary layer. For thresholds stricter than -0.10 K/min, the retrieved annual cycle of CPs (Figure 1d) is also out of phase with the observed annual cycle of rain events (Figure 1a). For stricter thresholds, there are too few retrieved CPs to suggest that all CPs are detected (0.93 CPs/24 hours when and 0.47 CPs/24 hours for the -0.11 K/min and -0.14 K/min thresholds, respectively) when comparing to the number of observed rain events. When detecting CPs at ENA using a constant threshold for 1-minute changes in temperature, the resulting annual and diurnal cycles of CPs are reflections of the annual and diurnal cycles of the background thermodynamic variability instead of the desired precipitation-driven CPs.

Finally, Figures 1e and 1f demonstrate the method's sensitivity to the subjective choice of threshold value. Figure 1e illustrates that the estimated total number of detected CPs is strongly and smoothly sensitive to the constant $\Delta T/\Delta t$ threshold, with no obvious choice of the best value. We also examined the ratio of the expected rates of NearRain CPs to FarFromRain CPs, represented in Figure 1f as $E_{NR}/E_{FFR}$. $E_{NR}$ is computed as the number of retrieved NearRain CPs per 24 hours of valid NearRain times and *vice versa* for $E_{FFR}$. We prefer the ratio $E_{NR}/E_{FFR}$ to be high, analogous to a combination of high hit rates and low false alarm rates. The $E_{NR}/E_{FFR}$ ratio increases steeply for thresholds stricter than about -0.1 K/min, indicating there is little non-CP thermodynamic variability stronger than about -0.1 K/min at ENA. However, CPs using these strict thresholds still do not adequately capture the diurnal cycle (Figure 1d) and likely report too few CPs compared to distinct rain events, as described above.

It is clear from Figures 1e and 1f that there is no single ideal constant-$\Delta T/\Delta t$ threshold and that subjectively assigning any reasonable constant threshold will result in both missed detections of weak CPs especially in the winter nighttime *and* false detections of heightened

background thermodynamic variability especially during the summer daytime. Due to the shortcomings of constant thresholds shown in Fig. 1, we conclude that use of a constant temperature change threshold is unable to adequately distinguish between CPs and background thermodynamic variability, at least at ENA where the CP signature is relatively weak and often of similar strength to the background variability. We therefore seek additional information with which to confidently discriminate between precipitation-driven CPs and background thermodynamic variability. We note that use of a constant threshold may be appropriate when applied to identification of CPs in stronger convection over land (Redl et al, 2015; Provod et al, 2016; Kirsch et al, 2021; Kruse et al, 2022) and oceanic deep convection (e.g., de Szoeke et al. 2017), when drops in temperature exceed those typically observed at ENA.

[Figure]

**Figure 1: (a) Annual and (b) diurnal cycles in the frequency of distinct surface-reaching rain events measured by ENA instrumentation. (c) Annual and (d) diurnal cycles of retrieved CPs using 4 different choices of constant ΔT/Δt thresholds. (e) The number of retrieved CPs as a function of constant ΔT/Δt threshold. (f) The ratio of the expected rates of CPs during NearRain times to the expected rates of CPs during FarFromRain times, $E_{NR}/E_{FFR}$ as defined in the text, as a function of constant ΔT/Δt threshold.**

**3.2 Cold pool observational metrics from *in-situ* surface instrumentation**

[Figure]

**Figure 2: Temporal evolution of (a) T', (b) q$_v$', and (c) S' around the start of ENA rain events. Results are split into three ranges of the maximum surface rain rate within the first 5 minutes of the rain event. Panels (b), (c), and (d), show the mean T, q$_v$, and relative humidity RH for combinations of summer/winter and day/night launches.**

Inherent to this work is the assumption that CPs form due to negatively buoyant air resulting from evaporation of falling rain. This is demonstrated in Figure 2, which illustrates the temporal evolution of T', S', and q$_v$' relative to the observed onset of surface rain in panels a, c, and e, respectively. The drop in average T' is clear when the maximum rain rate in the first five minutes (RRmax) is greater than 1.0 mm/hr but the T' decreases are not as obvious for weaker rain rates. The sub-cloud rain evaporation must also produce an increase in the relative water vapor mixing ratio q$_v$' *where evaporation occurs*, though that increase in q$_v$' aloft is often unable to overcome higher q$_v$' values near the surface in boundary layers that are not well-mixed, which occurs frequently at ENA (Figures 2b, 2d, and 2f). As a result, the near-surface q$_v$' often decreases when upper-PBL air descends to the surface with the downdraft (Figure 2c). We therefore omit q$_v$' from the algorithm.

After reaching the surface, the negatively buoyant CP air spreads laterally, altering the near-surface wind speed and potentially its direction. Assuming the mean wind direction at the surface is similar to that of the raining cloud at the top of the boundary layer, the leading CP

edge should be associated with anomalously high near-surface wind speed (Figure 2e) that is approximately coincident with the decrease in temperature. Thus, CPs must be associated with a decrease in T' and a likely increase in S', as measured by the surface meteorological instrumentation (MET and MAWS). It is important to note that the existence of CPs does not require the existence of co-located rain at the surface (CP area is greater than rain area and likely longer-lived), nor does the existence of a CP require any rain to reach the surface (Jeong et al., 2023). Conversely, surface-reaching rain may not produce a CP at the time the airmass intersects the surface instrumentation (e.g., in the presence of a cool/moist surface layer or a nearly saturated boundary layer). We therefore do not require surface-detected rain for the identification of specific CPs. We expect that recently-formed rain events whose first surface-reaching drops fall upon the surface instrumentation have not yet had enough time to develop a CP, leaving them unobservable by the surface instrumentation. On the other hand, we expect CPs to survive longer than the rain falling from an individual cloud. Figure 2 illustrates some of the challenges in detecting CPs formed by boundary layer precipitation.

When looking for CPs in the temperature and wind speed time series, we first define CP "candidates" as a contiguous decrease in T' of any duration from the crest to the next trough. Most of these 353,338 candidates are manifestations of the background thermodynamic variability and will be rejected by the CP detection algorithm. We utilize three metrics to detect CPs from the surface station time series at ENA, each of which will produce a single value for each CP candidate. The first is the "Depth" metric, which is simply the accumulated drop in T' during the candidate: $Depth = \max(T') - \min(T')$. This is most similar to the methods employed by Wilbanks et al. (2015). The second metric is the "Rate", which is the absolute value of the strongest 1-minute decrease in T' during the candidate: $Rate = |min(\Delta T'/\Delta t)|$. The Rate metric is similar to what is used in many previous works (Terai and Wood, 2013; Vogel et al., 2021). Note that although the correlation between Depth and Rate is 0.92, indicating a high degree of shared information, we retain the use of both complementary metrics because the Depth metric is less susceptible to instrument noise and small-scale variability, and the Rate metric is less susceptible to longer duration non-CP signatures. The third metric is the "Gust", which is the difference between the maximum and minimum S' within 10 minutes before and after the time associated with the steepest drop in temperature ($t_{Rate}$): $Gust = \max(S'(t_{Rate} - 10 : t_{Rate} + 10)) - \min(S'(t_{Rate} - 10 : t_{Rate} + 10))$. The Gust metric is defined over a longer period because the measured peak in S' associated with surface-reaching rain is not always coincident with $t_{Rate}$.

**3.3 Time-varying metric thresholds**

At the heart of our CP retrieval algorithm is the requirement for the Depth, Rate, and Gust metrics to exceed values characteristic of the background thermodynamic variability, which varies throughout the year and day. We therefore set separate thresholds for each metric as a function of time of day and time of year. Broadly, the thresholds are determined by identifying the metric value for which the fraction of candidates associated with rain event starts ($F_{RE}$) deviates significantly from that group's average $F_{RE}$. The technique is detailed as follows.

The candidate Depth, Rate, and Gust metrics are collected into one-month/two-hour groups for a total of 144 groups. For a given month/hour group, candidates belonging to the 8 neighboring month/hour groups are aggregated to improve counting statistics. Figure 3a and Figure 3d demonstrate that the Depth distributions during January 0000-0200 UTC and July

Moved down [1]: require any rain to reach the surface (Jeong et al., 2023).
Moved (insertion) [1]

1200-1400 UTC peak strongly at very weak values. The vast majority of candidates will be rejected in favor of relatively few confidently-detected CPs. The fraction of candidates that are associated with rain event starts ($F_{RE}$) increases with increasing Depth in both groups (Figure 3b and 3e), eventually exceeding the average for the group. We then perform $\chi^2$ tests for dependence between the $F_{RE}$ from the entire group and the $F_{RE}$ as a function of Depth, with the null hypothesis being that the distribution of $F_{RE}$ for the entire group is not different from the distribution of $F_{RE}$ for candidates with a given Depth value. These subsets are defined using a moving window approach, allowing for a dynamic assessment of variations in $F_{RE}$ over different segments of the group. The p-value from the test indicates whether there is a significant difference between the full-group average and the subset values. The weakest metric value for which $p \leq 0.01$ and $F_{RE}>$mean($F_{RE}$) is selected as the metric's threshold for that month/hour group. Figures 2c and 2f demonstrate that the Depth threshold for daytime summer is larger than for nighttime winter, reflecting the different levels of variability and rain event frequency between those regimes.

[Figure]

**Figure 3: The process of defining Depth thresholds (red dashed lines) for January 0000-0200 UTC (a-c) and July 1200-1400 UTC (d-f). (a and d) Histogram of the Depth in black. (b and e) Rain event frequency as a function of Depth metric in black with the group-average $F_{RE}$ in solid-red. (c and f) $\chi^2$ and associated p-value for a test of dependence between $F_{RE}$ as a function of Depth and the grou-average $F_{RE}$. Thresholds are defined when the p-value diminishes below 0.01.**

Figure 4a shows the full annual/diurnal cycle of candidate frequency at ENA. The distribution peaks during the winter and spring evening and night a secondary peak during the summer daytime. However, we do not expect Figure 4a to reflect the occurrence frequency of actual CPs due to time-varying background thermodynamic variability and rain event frequency. Figure 4b shows that the frequency of distinct rain events is actually at a minimum in the summer daytime and evening, with its peak during the winter daytime. The thresholds for Depth (Figure 4c) and Rate (Figure 4e) are highly correlated, as expected, with the strictest thresholds found during the summer daytime when the influence of solar heating-driven boundary layer variability is greatest. The Gust thresholds peak during the summer nighttime, with a minimum

1000-1400 UTC from winter to late spring. Figure 4 also shows the fraction of candidates that exceed their respective thresholds (Figure 4d, 4f, and 4h). Fractionally, more candidates exceed their thresholds during the fall and winter seasons, especially during the night hours. In summary, Figure 4 demonstrates this technique's ability to allow the Depth, Rate, and Gust thresholds to reflect expected seasonal and diurnal patterns of background thermodynamic variability and rain event frequency.

[Figure]

**Figure 4: a) Total number of candidates belonging to each month/hour group. (b) Expected frequency of rain events. (c, e, and g) Depth, Rate, and Gust thresholds found at ENA. (d, f, and h) The fraction of candidates that exceed the respective Depth, Rate, and Gust thresholds. The time zone at ENA is UTC-1.**

**3.4 Detecting cold pools from the metrics and time-varying thresholds**

Now we turn to the designation of CPs based on a candidate's metrics exceeding the relevant thresholds. First, we group the temperature-related metrics (Depth and Rate) together. If either the Depth or Rate metrics do not exceed their thresholds, the candidate is not considered to be a CP and its weight is set to 0.0. If all three Depth, Rate, and Gust metrics exceed their

thresholds, the candidate is given a weight of 1.0, indicating that we are fully confident that it is a CP and not background thermodynamic variability. We also find cases where the Depth and Rate thresholds are exceeded but the Gust threshold is not. Some of these cases are accompanied by surface precipitation, but some are clearly not. With a lack of further information, we set the weight to 0.5 for these cases, reflecting an uncertainty in their identify while including them with an appropriate lower confidence. These cases weighted 0.5 have less influence on computed statistics than cases for which all three thresholds are exceeded.

Following these rules, we assign a weight to each candidate in the record, constituting our "Best" estimate of the CPs at ENA. Alternative estimates produce qualitative uncertainty ranges; the "Exclusive" estimate requires all three metrics to exceed their thresholds and the "Permissive" estimate requires only the Depth and Rate metrics to exceed their thresholds. Unless otherwise noted, results are shown for this Best estimate. Because the choices in this final step are subjective, we later examine alternative choices to gain an understanding of qualitative uncertainties. First we proceed with a discussion of the general properties of CPs retrieved by our Best estimates.

[Figure]

Figure 5: (a) Number of candidates, (b) $F_{RE}$, (c) sum of candidate weights, and (d) the average candidate weight as a function of the Depth and Gust metrics.

Figure 5 provides an understanding of where retrieved CPs inhabit the bivariate Depth and Gust space, with the understanding that Depth and Rate provide similar information. Most candidates have very weak Depth and Gust values (Figure 5a). However, those weak-value candidates are usually given a weight of zero (Figure 5c and 5d), consistent with the very low $F_{RE}$ in that space (Figure 5b). Note that Depth and Gust are not directly correlated, meaning each

is providing new information to the algorithm. Figure 5b shows that extreme cases in which either Depth≫0 and Gust≈0 or Depth≈0 and Gust≫0 are associated with an increase in the fraction of candidates that contain rain compared to when both Depth and Gust are weak, supporting their inclusion in the algorithm. Most of the summed candidate weights are associated with Depth≈0.4 K and Gust≈1 m/s. Figure 5d communicates that, while these cases are quite frequent at ENA, they are usually assigned a weight of 0.5. This points to the challenge of separating CPs from background thermodynamic variability at ENA, especially using simple 1-minute changes in temperature and wind speed. Regardless, the reduced weights of these candidates reduce their overall influence on the statistics presented in the CP climatology we present in the next section. Overall, Figure 5 shows that most candidates have weak Depth and Gust metrics are generally discarded by the algorithm and that higher candidate weights are associated with increased frequency of observed surface rain.

Figure 6 provides an alternative view, where the fraction of candidates that are designated as CPs is displayed as a function of candidate Depth. The candidate frequency distribution (log-spaced bins) sharply decreases for stronger Depth values greater than about 0.2 K. However, many of those weak-Depth candidates are likely background thermodynamic variability and are therefore not associated with CPs. Note again that a constant threshold in Depth would result in both missed and false detections because the CPs at ENA tend to be weak and their Depth distribution merges with the background variability. Weak candidates (e.g. Depth <0.2 K) are almost never diagnosed as CPs, while strong candidates (Depth>2.0 K) are almost always diagnosed as CPs.

[Figure]

Figure 6: Histogram of the strength of candidates (dotted black) and retrieved cold pools (solid black) with the fraction of candidates that are estimated to be cold pools in blue.

**3.5 Algorithm justification**

Figure 7 provides a proof of concept for CP identification through a series of relatively heavy and frequency rain events occurring during the ACE-ENA winter intensive observation period (Wang et al., 2022). CP candidates are highlighted with red lines, where the line thickness

scales with the confidence weights assigned by the detection algorithm. The candidates with thick red lines received a weight of 1.0, indicating strong confidence that they are CPs. Indeed, the first five of these CPs coincide with surface-reaching precipitation. Later, two CP candidates received a weight of 0.5 (shown with thin red lines) because their Gust values did not exceed their thresholds. The KaSACR scans spanning these final candidates are provided in Supplementary Figure 2 and reveal the horizontal structure of evolving precipitation passing near the ENA site. Although the KaSACR did not observe rain near the instrumentation during the 13:45 UTC candidate determined to be a CP, we suspect that the steep drop in T' represents a CP *remnant* from a terminated rain event that continues to propagate with the boundary layer flow. The KaSACR scans around 14:20 UTC and 15:04 UTC indicate scattered precipitation near the site and KaSACR scans at 15:15 UTC reveal that rain did indeed pass quite near but not directly over the ENA instrumentation (Supplemental Figure 2), suggesting that the CP from that nearby rain expanded laterally after descending to the surface. We suspect that the gust front intensity decreases as the CP expands farther from the downdraft, but we leave that investigation to future work. Figure 7 demonstrates that the "Best" CP detection algorithm can identify both CPs with and without associated surface rain observed by the ORG, PWD, and LD.

[Figure]

**Figure 7: (a) T', (b) 1-minute T' changes, and (c) S' for a selected period during ACE-ENA. Thick and thin red lines denote candidates for which weight W=1.0 W=0.5, respectively. Candidates with W=0.0 are not labeled. At the top of each panel for each candidate, the top number reflects the relevant metric value and the bottom number reflects the threshold for that month/hour group.**

Although there is no truth dataset against which we can judge the algorithm's performance, we executed the $E_{NR}/E_{FFR}$ analysis previously shown for the constant threshold technique (Figure 1f). The "Best" algorithm produces $E_{NR}/E_{FFR}$=2.65, indicating CPs are 2.65 as likely to be found in the NearRain periods as the FarFromRain periods. This value is coincidentally only slightly above the $E_{NR}/E_{FFR}$=2.35 value found for the constant threshold of -0.05 K/min used in Vogel et al. (2021) applied to ENA observations. The coincidence further

extends to the total expected rates of CPs between our algorithm (5.95 CPs per 24 hours) and the Vogel et al. (2021) algorithm (5.96 CPs per 24 hours). However, the similarities end there, as the diurnal and annual cycles will be shown to be quite different between the two techniques (Section 4.2). The Exclusive and Permissive estimates produce a qualitative uncertainty range of 3.57 to 8.34 CPs per 24 hours of valid observations. Note that while this qualitative uncertainty range is large, we will show that *the seasonal and diurnal cycles are not strongly affected by the choice of algorithm, as long as the thresholds vary throughout the year and day.*

To justify the Best estimate's performance, Figure 8 shows the evolution of surface rain for four regimes in the Depth/Gust metric space. The regime boundaries were chosen to separate the candidates by their metric values while maintaining adequate CP sampling and have the following sampling counts listed as the number of estimated CPs divided by the number of candidates: Weak-Gusty (340/1600), Strong-Gusty (1096/1102), Weak-Calm (166/42373), and Strong-Calm (460/948). At $t_{Rate}$, the Weak-Calm candidates have a rain fraction (8.6 %), slightly higher than the overall average (7.9 %), supporting the reduced average CP weight of the Weak-Calm candidates (0.053). Conversely, The Strong-Gusty and Strong-Calm candidates have elevated rain fraction, conditional rain rate, and average rain rate surrounding $t_{Rate}$, supporting their much higher average weight (0.995 and 0.485, respectively). While the rain fraction within ±1 hour surrounding $t_{Rate}$ is near-constant at 4.3 times the overall average for Weak-Gusty candidates, the rain rates are both elevated and variable, suggesting variability in rain rates and wind speeds within longer-lived rain events, possibly frontal systems or mesoscale systems.

[Figure]

**Figure 8: Mean evolution of surface rain for candidates falling within four regions of the Depth/Gust bivariate metric space. "Weak" and "Strong" candidates have 0.1<Depth<0.3 K and 1.0<Depth<5.0 K, respectively. "Calm" and "Gusty" candidates have 0.0<Gust<1.0 m/s and 2.0<Gust<13.0 m/s, respectively. Uncertainty envelopes represent 95% confidence intervals around the mean, obtained from $10^3$ bootstrap resamples with replacement at each time point. Panels depict (a) rain fraction, (b) conditional rain rate excluding non-raining times, and (c) average rain rate including non-raining times.**

**4 Annual and diurnal cycles of cold pools at ENA**

We now examine the 2016-2024 record of cold pools observed at the ENA site. In total, the algorithm identifies 308,701 CP candidates throughout the 9-year analysis period. However, once we account for the time-dependent background variability, there are only 23,509 candidates with weights greater than zero (Figure 5a). Those weights sum to an expected total of 19,198 CPs that are distinguishable from the background variability. Recall that the candidates for which the Depth and Rate thresholds *are* exceeded but the Gust threshold *is not* exceeded are assigned a weight of 0.5 and therefore have a reduced influence on the CP statistics, reflecting the reduced confidence that those cases are CPs. For context, there are a total of 27,596 distinct rain events during this period. So the sum of the CP candidate weights is only about 61.3 % of the number of

observed surface rain events. On the other hand, only 33.7 % of CP candidates with weight equal to 1 are associated with observed surface rain during the candidate. This section describes the characteristics of those 19,198 retrieved CPs.

**4.1 Cold pool properties**

We now examine CP statistics from the entire population of CP candidates. To understand the typical cloud scene associated with CPs, we show the connection between stronger candidates and more cumulus-like scenes in Figure 9. The discrimination between stratocumulus (Sc) and shallow cumulus (Cu) cloud regimes is based on Zheng and Miller (2022), who diagnosed cloud structure in 6-hour segments with three cloud/precipitation variables gleaned from the ARM Ka-band zenith-pointing radar (KaZR) and ceilometer at ENA. Their "Thickness Index" (TI in Figure 9) broadly represents the fraction of the boundary layer inhabited by clouds, their Drizzle Index (DI in Figure 9) increases for segments with more intense drizzle, and their Complexity Index (CI in Figure 9) increases in time segments characterized by variable cloud base heights. See Zheng and Miller (2022) for details. Rather than perform a k-means categorization to define Cu and St segments as done in Zheng and Miller (2022), we simply assign thresholds to the TI, DI, and CI values as detailed in the Figure 9 caption. Not all segments are assigned as Sc or Cu. Figure 9c shows that as Depth increases, the fraction of candidates independently classified as Cu scenes also increases. Contrastingly, the fraction classified as Sc begins slightly under the overall average decreases to zero in candidates with stronger Depth, indicating that stronger CPs are not associated with Sc cloud types at ENA. Figure 9 demonstrates that CPs are more associated with cumulus-topped boundary layers instead of stratocumulus regimes, especially for strong CPs.

[Figure]

**Figure 9: (a) Scatterplot of categorized 6-hour segments. Segments are labeled as Sc if they have TI<0.20, DI<0.10, CI<0.30, and more than 80 % cloud cover. Segments are labeled as Cu if they have TI<1.00, DI>0.05, CI>0.20, and less than 70 % cloud cover. Not all segments are assigned as Sc or Cu. A rotated perspective of (a) is shown in (b). (c) Fraction of cold pool detections that are associated with Sc and Cu cloud types as a function of CP candidate Depth. Overall average values (thin horizontal lines) are computed as the fraction of all valid 1-minute time points that are classified as either Sc or Cu.**

[Figure]

**Figure 10: Average vertical air motions from the zenith-pointing Doppler lidar during candidates for which (a) RR==0 and Depth>1.0 K, (b) 0<maximum RR<10 mm/hr, and (c) RR>10 mm/hr. The color bar limits in (b) and (c) are saturated to show details of the areas of both upward and downward motions. The strongest downward motion in (b) and (c) is -0.23 m/s and -0.59 m/s, respectively.**

CPs at ENA are coincident with downward motion that is preceded by upward motion. Figure 10 shows this evolution by compositing time series of sub-cloud vertical motion retrieved by the zenith-pointing Doppler lidar during CP candidates. In all panels, upward motion precedes downward motion surrounding $t_{Rate}$. These patterns show the updrafts and downdrafts coincide with the CPs observed by the surface station. In all panels, ascending motion precedes descending motion coinciding with $t_{Rate}$, likely representing a combination of the updraft and mechanical lifting of boundary layer air by dense CP air. About 20-30 minutes after $t_{Rate}$, it appears that another updraft tends to pass over the lidar, with similar timing to the increases in rain rate shown in Figure 8. In Figure 10a, which includes strong Depth candidates with no observed surface rain, there still exists some downward motion preceded by upward motion, indicating that the rain system may have passed nearby but not directly over the site, as was demonstrated in Figure 7 and Supplementary Figure 2. Although the vertical motions in Figure 10 appear to be weak, note that CP sizes can be small (shown later in Figure 12), so averaging vertical motions that are slightly offset in time will result in a diminished picture of the individual in-downdraft descent rate.

**4.2 Cold pool climatology and synoptic setting**

The CP annual and diurnal cycles in ENA CPs are shown in Figure 11. The annual cycle (Figure 11a) is more pronounced than the diurnal cycle (Figure 11b), with CPs being observed most frequently in the boreal winter months and at night when the background variability is

relatively calm (Figure 3 and Figure 4). In contrast to the constant-threshold method (Figure 1c), retrieved CPs using the Depth, Rate, and Gust metrics with dynamic thresholds (Figure 11a) closely match the annual cycle of precipitation events at ENA (Figure 1a). Note that the diurnal cycle of CPs (Figure 11b) peaks between 0000-0500 UTC, while the peak in the diurnal cycle of rain events peaks at about 0800 UTC and is slightly elevated during until about 1400 UTC (Figure 1b). This discrepancy is explained by the increased background thermodynamic variability in the boundary layer during daylight hours, requiring increases in the threshold value that is used to identify CPs (as shown in Figures 4c and e). Rain events during the peak times (~0800 UTC) certainly lead to cool, negatively buoyant air that descends to the surface. However, the signals from those events are embedded in stronger background variability, making their signals both more difficult to detect but also less meaningful regarding their effects on further convection and changes to the boundary layer structure and cloud morphology. The seasonal and diurnal cycles are not strongly affected by the choice of algorithm, as long as the thresholds vary throughout the season and day following background thermodynamic variability and the frequency of rain events (Figures 11a and 11b).

 Figure 11d shows the monthly expected number of CPs and rain starts for the 2016-2024 record. Correlations between the number of expected monthly rain events of any minimum rain rate with the number of expected monthly CPs is 0.68. The correlation increases to 0.73 when including only rain events with maximum rain rate greater than 1 mm/hr. A simple regression of the time series of monthly expected CP counts yields no significant linear trend at the 95% confidence level, though more sophisticated techniques are warranted for a clearer picture of the dependence of CPs on atmospheric and oceanic trends and cycles affecting the boundary layer at ENA.

[Figure]

Figure 1d reveals bivariate histograms of ΔT'/Δt and Δq$_v$'/Δt for time periods that are associated with rain (NearRain; surface rain observed within 1 hour before or after) and not associated with rain (FarFromRain; no surface rain observed within 6 hours before or after). We assume that the FarFromRain category does not contain cold pool signatures and is therefore representative of the "background turbulence" experienced at the site, though we note the possibility of an isolated rain shower passing near the met station, thus contaminating the FarFromRain category with larger deviations. In total, there are 703,041 individual FarFromRain minutes, about 17.6% of the total observational record. Temperature and moisture deviations during FarFromRain periods are generally symmetric about the origin and are usually very small at the 1-minute time scale used in this work. In contrast, the temperature deviation extremes for NearRain periods are skewed towards ΔT'/Δt<0, indicating a tail in the distribution resulting from evaporation of rain. Note that the NearRain category is expected to contain most of the true cold pools but there is no distinct distribution of ΔT'/Δt and Δq$_v$'/Δt that can be clearly delineated from the non-cold pool distribution. Therefore, defining a strict threshold in ΔT'/Δt and/or Δq$_v$'/Δt would result in both missed detections of small cold pools and false alarms due to anomalously large background turbulence. ¶
In addition to these metrics that are observable with the surface meteorological instrumentation, we expect that rain events whose first surface-reaching drops fall upon the surface met station have not yet had enough time to develop a cold pool, leaving them unobservable by the surface instrumentation. On the other hand, we expect cold pools to survive longer than the rain falling from an individual cloud. Figure 1 illustrates some of the challenges in detecting cold pools formed by boundary layer precipitation.¶
¶
... [4]

**Figure 11: Expected frequency of retrieved cold pools as a function of (a) month, (b) UTC, and (c) month/UTC. In (a) and (b), thick black lines represent our Best estimate (described in the text), while the dotted line represents the most restrictive estimate (Depth, Rate, and Gust required to exceed their thresholds; Exclusive) and the dashed line represents the most**

**permissive estimate (Depth and Rate required to exceed their thresholds; Permissive). Expected rates of distinct rain events (all events and only those with maximum rain rate stronger than 1 mm/hr) along with the expected cold pools for each month during 2016-2024 (d). The time zone at ENA is UTC-1.**

Figure 12 presents the retrieved CP size distribution, where the sizes have been estimated from the CP candidate time durations with the following arguments. As explained in Section 3, we define the start and finish of a given CP candidate as the preceding local maximum T' and following local minimum T', respectively. The along-wind length of the part of the CP that passes over the ENA site can be estimated by the elapsed time multiplied by the propagation speed, as long as the CP evolution is slow compared to the time the CP requires to pass over the instrumentation. However, because the CPs are expected to intersect the surface instrumentation at a random location along the across-wind direction of the CP, this simple calculation is likely to underestimate the actual CP radius. By assuming that (1) CPs are circular (with the ΔT'<0 portion forming the leading semicircle), (2) intersections with the ENA instrumentation are uniformly random, and (3) the average wind speed during the CP temperature decrease represents the CP propagation speed, we create a distribution of possible radii (*r*) for a given CP from the procedure detailed in Supplementary Figure 3. In short, the procedure involves computing a distribution of possible radii for each CP candidate and adding the possible radii for all CPs, accounting for each CP candidate's weight. The weighted mean CP radius is found to be 8.0 km (Figure 12) though most CPs have a smaller size. The CP radius distribution peaks at about 4.5 km. About 5.1% of retrieved CPs have radii less than 1 km, while about 26.1% have radii greater than 10 km. For comparison with the southeast Pacific Ocean, Terai and Wood (2014) found a median CP size of about 6 km but Wilbanks et al. (2015) found CP lengths ranging from 5-40 km with a mean length of 15 km.

[Figure]

**Figure 12: PDFs of observed chord length (blue) and estimated CP radius (red).**

Figure 13 shows the synoptic setting for days with a high versus low number of CPs. Sea level pressure values are taken from the Modern-Era Retrospective Analysis for Research and Applications, Version 2 (MERRA-2; Gelero et al., 2017). We compute the daily-average of the 3-hourly instantaneous values provided by the "inst3_3d_asm_Nv" product for days that either have a large number of CPs in that day (277 days) or a small number of CPs in that day (206

To characterize the background bivariate deviations separately in each month/hour group, we apply singular value decomposition (SVD) to the ΔT'/Δt and Δqᵥ'/Δt belonging to that group and the eight neighboring groups. This produces the eigenvalues and eigenvectors that represent the relative magnitude of variabili… [5]

[Figure]

days). Days with many CPs tend to be associated with sea level pressure (SLP) depressions (Figure 13a and 13b), especially those centered to the North and East of the Azores such that Graciosa Island is likely experiencing a post-frontal environment with frequent shallow precipitation events due to subsidence in the lee of a passing trough (Lamer et al. 2020). In contrast, days with few CPs are characterized by slight ridging near and to the East of the Azores islands (Figure 13c and 13d).

[Figure]

**Figure 13:** (a) Sea level pressure (SLP) maps for days with frequent cold pools. (b) Same as (a) but as a deviation the average SLP at each location. Panels (c) and (d) are the same as (a) and (b) but for days with few cold pools. In each panel, the black "x" designates the location of the ENA site. Sea level pressure values are taken from MERRA-2

**5 Which Rain Events Produce Observable Cold Pools?**

Our Best estimate for CP frequency at ENA is 5.95 CPs per 24 hours but the expected rate of rain events ($F_{RE}$) is 9.08 rain events per 24 hours. Given that we detect CPs that are not associated with observed rain at the site, why do so many rain events *not produce* an observable CP at the surface as the rain passes over the ENA instrumentation? This section explores which rain events result in CPs that are distinct from the background thermodynamic variability.

Figure 14 shows the frequency distribution of mean CP candidate weights as a function of variables relevant to boundary layer structure and CP formation, while Figure 15 shows the frequency distributions of retrieved CPs compared to the full distributions and the distributions during rain event starts. As expected, rain events with heavier rain in the beginning of the event are more likely to be accompanied by measurable CPs (Figure 14a). Rain events starting with rain rates greater than about 0.5 mm/hr result in a significant elevation of CP weight compared to

4 Characteristics of

the average of all rain events. Indeed, CPs tend to form in association with higher rain rates than the typical rain event (Figure 15a).

While only 38.4 % of rain events occur when the surface relative humidity is less than 80 %, 56 % of CPs occur in these relatively dry conditions (Figure 15b). A given rain event is more likely to create a measurable CP if the relative humidity immediately preceding the rain event start is between 60-80 % (Figure 14b). There are multiple plausible explanations for this, including that (1) less evaporation occurs when the sub-cloud layer is humid, leading to diminished downdrafts and therefore diminished CP signatures and (2) a previous CP conditioned the near-surface layer with moist air, effectively creating a low-level inversion that traps moisture near the surface. The boundary layer integrated saturation deficit exhibits clear associations with mean CP weights, as well (Figure 14h). We define the integrated saturation deficit as $H_{CB} \sum (100 - RH(z))$ in all evenly-spaced sub-cloud layers, where $H_{CB}$ is the cloud base height from ceilometer and RH(z) is the relative humidity [%] in a given layer. This metric is intended to summarize both the relative humidity of the sub-cloud layer and the distance the drops need to fall through sub-saturated air. Rain events with very low integrated saturation deficit (either high humidity or low cloud base heights or both) have a strongly reduced chance of being associated with CPs. CPs tend to occur when the saturation deficit is higher than typical for rain events (Figure 15h).

We also examined the rate of surface pressure change while rain events when rain is first observed at the station (Figure 14c and Figure 15c). The 1-hour pressure change was computed as P(t+30) – P(t-30), where P is the air pressure measured by either MET or MAWS and t is the time of the rain event start. CP weights tend to be slightly greater than the average when surface pressure increases by more than about 0.50 hPa over the course of an hour, which we speculate is related to post-frontal scattered showers and potentially cold air outbreaks. These cases represent the upper tail of the 1-hour pressure change distribution (Figure 15c).

Near-surface horizontal winds during rain events are also shown to be connected to CP passages at ENA. Rain events with surface horizontal wind speed greater than 7 m/s are associated with elevated chances of CPs, while calm conditions are not (Figure 14d). Because the strongest wind speeds are less frequent, most CPs are associated with wind speeds between 5 – 11 m/s (Figure 15d). Increased wind speed variability is also associated with a higher chance that a rain event will lead to a measurable CP (Figure 14f). While this CP detection algorithm utilizes the change in wind speed within ±10 minutes of $t_{Rate}$, here we assess the wind speed variability over a longer period of two hours, better representing the boundary layer characteristics outside of the immediate vicinity of the rain event. Figure 15f shows that, although rain events are associated with a higher wind speed variability than the distribution for all times, rain events that lead to CPs exist within environments with further increased dynamical variability.

CPs are associated with 2-hour low cloud cover (defined as the fraction of the time that the ceilometer retrieves a cloud base lower than 4 km from 60 minutes prior to 60 minutes after a given time) between about 0.35 and 0.90 (Figure 14e). While rain events are associated with low cloud fraction near 1.0 (Figure 15e), CPs are not as closely associated with complete cloud coverage. This is consistent with the findings of Figure 9, where the stratocumulus cloud type is less frequently associated with CPs in candidates with larger Depth metric.

The mean CP weight is positively correlated with estimated inversion height, especially inversion heights greater than 2 km (Figure 14g). In the shallowest boundary layers, precipitation experiences less time in the sub-saturated sub-cloud layer, resulting in less total evaporation. We also speculate that the rain rates tend to be weaker in very shallow boundary layers than in

deeper layers where convection has developed further. Inversion heights are taken from the Heffter method (Heffter, 1980), except in cases when the Heffter inversion height is less than 300 m, in which case we use layer with the greatest increase in potential temperature below 4 km. Figure 15g shows that the boundary layer depth distribution is increased towards higher values when CPs occur than for other rain events.

The boundary layer decoupling index and shear exhibit reduced signals compared to some other variables. Severely decoupled cases (DEC>10 K) tend to have smaller cold pool weights and most rain-associated CPs occur when 2<DEC K<7 (Fig. 13g). Very low values of PBL shear are associated with slightly increased CP weights (Fig. 13h). PBL shear at ENA is usually less than 10 m/s and CPs rarely occur above this value.¶

¶                                      ... [15]

[Figure]

**Figure 14: Blue lines show the uncertainty ranges for the average CP weight (including zeros for rejected candidates) for candidates that occur during the start of a rain event, as a function of variables obtained from the *in-situ* surface instrumentation (a, b, c, d, f), ceilometer (e), and balloon sondes (g, h). Error bars denote the 95% confidence intervals computed via bootstrap resampling with 10³ resamples with replacement. Red lines provide the corresponding confidence intervals for the overall average candidate weight during all rain events.**

[Figure]

[Figure]

**Figure 15:** Like Figure 14, but the fraction of retrieved cold pools (blue), the during rain starts regardless of cold pool detection (black), and the overall distributions regardless of cold pools or rain events (grey).

**6 Conclusions**

This work presents cold pool (CP) statistics from 9 years of surface measurements at the ARM ENA site on Graciosa Island in the Azores. Seeking to avoid a combination of false alarms and missed detections that necessarily result from constant temperature change methods, we developed a new methodology that: (1) uses three metrics (Depth, Rate, and Gust) to characterize CP candidates, (2) assigns thresholds for each metric based on statistical association between the frequency of distinct rain events and increasing magnitudes of the three metrics, and (3) allows those thresholds to vary throughout the annual and diurnal cycles, thus accounting for time-varying background thermodynamic variability that is not associated with CPs.

Despite having no "truth" benchmark with which to validate the CP detections, we demonstrated expected CP behavior related to a time series of surface-reaching rain coupled with scanning radar of passing precipitation systems, surface-reaching frequency and intensity of rain for candidates belonging to four regions of the Depth/Gust bivariate space, and the frequencies of CPs in cases we are confident did or did not contain CPs using a combination of surface-observed precipitation and satellite-based IMERG precipitation retrievals. We presented statistics on CPs over 9 years, including their total number and temporal cycles, their relationships to synoptic weather regimes, and their physical sizes. We then analyzed the characteristics of the boundary layer associated with the observed CPs during surface-reaching rain events. Our findings can be summarized as follows:

1. Identifying ENA CPs using a simple 1-minute temperature change threshold from meteorological station observations results in both missed detections and false alarms because a fixed threshold does not account for changes in the background variability throughout the year and day, regardless of the exact threshold value used. This simple constant threshold method tends to overcount CPs (i.e., detects false alarms) during the afternoon hours due to increased boundary layer thermodynamic variability while simultaneously missing weaker CPs that occur during winter and nighttime precipitation, leading to an overly-strong and reversed representation of the CP diurnal cycle. We also demonstrated the strong and undesirable sensitivity of the number of CPs retrieved to the choice of constant temperature threshold value.

2. The main advance presented in this work is the development of a method to assign time-varying thresholds determined by statistical associations between the frequency of distinct rain events and increasing magnitudes of three CP candidate metrics. The three metrics, each receiving one value for each CP candidate, permitted capture of diverse CP types, which may be targeted for investigation in future work. These types were presented in Figure 8 and include strong CPs with strong gusts (near or directly under the heavier precipitation source), strong CPs with weak gusts (likely CPs farther from the rain source), and weak CPs with strong gusts (associated with longer-lived rain events, potentially frontal systems).

3. Summing the candidate weights for our Best estimate, we identify 19,198 individual CPs that are distinct from the background variability from 2016 to 2024. The seasonal cycle of ENA CPs peaks in the colder months, with ~7-8 CPs per 24 hours compared to ~4 expected CPs per 24 hours in the summer months. The CP diurnal cycle is less pronounced than the annual cycle, consistent the muted diurnal cycle in the frequency of distinct rain events. Although the frequency of rain events peaks from 0800-1400 UTC, the temperature and wind gust signals

of these afternoon CPs are often obscured by elevated background thermodynamic variability, leaving them less likely to affect the boundary layer and leaving them statistically indistinguishable from non-CP processes. Days with more than 8 retrieved CPs are associated with the post-frontal sector of synoptic troughs.

4. The number of CPs detected here represents only about 66 % of the number of distinct rain events, as observed by contiguous 1-minute rain rates from surface instrumentation. When rain is first measured by the in-situ surface instrumentation, detected CPs tend to have the following properties: leading-edge rain rates greater than 0.5 mm/hr, near-surface relative humidity less than 80% and large values of integrated saturation deficit, deeper boundary layers, stronger horizontal near-surface wind speed and variability, and broken fractional low cloud cover between 0.35 and 0.90.

Though we are confident that the methodologies detailed here represent improvements in retrieving CP properties from simple near-surface time series observations from surface instrumentation, we also note the following unknowns, limitations, and potential future improvements to this methodology:

1. CP candidates that are similar in magnitude to the background thermodynamic variability are included in the statistics, though many of them are likely not actually CPs.

2. While we endeavored to demonstrate expected behavior of the algorithm in Section 3, we do not have a "truth" dataset that can provide a full validation of CP retrievals.

Although this algorithm was developed to examine weak CP signatures resulting from precipitation that is most-often confined to the boundary layer, we note that the metric thresholds could also be recomputed at other locations without the need for subjectively-tuned or constant thresholds, as long as the location has a long enough record to adequately characterize the background thermodynamic variability that is not associated with CPs and varies throughout the time of year and day.

**Code Availability**

Relevant algorithm and analysis code will be made publicly available on Zenodo prior to publication.

**Data Availability**

ARM ENA observations may be downloaded from the ARM data archive at https://www.arm.gov/data/. Half-hourly IMERG Final gridded precipitation retrievals can be accessed from NASA's GES DISC at https://disc.gsfc.nasa.gov/datasets/GPM_3IMERGHH_07/summary?keywords="IMERG final". MERRA-2 reanalysis sea level pressure fields may be accessed from NASA's GED Disc at

Cold pools

https://disc.gsfc.nasa.gov/datasets/M2I3NVASM_5.12.4/summary?keywords=inst3_3d_asm_Nv
.

**Author Contribution**

MAS led this work, from the acquisition of the observations from the ARM data archive, the development and refinement of the cold pool detection algorithm, the figure generation, and the writing of the manuscript. MKW led the proposal that funded this work and the general direction of this work. MJC provided context regarding ongoing large eddy simulation work building off these results. JHJ organized the ACE-ENA scanning radar figures and provided advice for the Doppler lidar analyses. All co-authors contributed to the discussion of the techniques and results during biweekly meetings and to editing the manuscript.

**Competing Interests**

The authors declare that they have no conflict of interest.

**Acknowledgements**

This work was supported by the U.S. Department of Energy's Atmospheric System Research, an Office of Science Biological and Environmental Research program, under awards DE-SC0022992 and 89243022SSC000094. The analyses presented here relied upon accessible observations from the ARM Data Archive, for which we are grateful. This work was performed at the Jet Propulsion Laboratory, California Institute of Technology, under a contract with the National Aeronautics and Space Administration. The authors offer their sincere thanks to ARM director Jim Mather and instrument mentor Jenni Kyrouac for their helpful insights about the surface sensor noise characteristics at ENA.

**References**

Blossey, P. N., Bretherton, C. S., & Mohrmann, J. (2021). Simulating Observed Cloud Transitions
in the Northeast Pacific during CSET, Mon. Wea. Rev., 149(8), 2633-2658,
https://journals.ametsoc.org/view/journals/mwre/149/8/MWR-D-20-0328.1.xml.

Bretherton, C. S., & M. C. Wyant (1997), Moisture transport, lower-tropospheric stability, and decoupling of cloud-topped boundary layers, J. Atmos. Sci., 54(1), 148–167.

Ceppi, P., & Nowack, P. (2021). Observational evidence that cloud feedback amplifies global warming. Proceedings of the National Academy of Sciences, 118(30).

de Szoeke, S. P., Skyllingstad, E. D., Zuidema, P., & Chandra, A. S. (2017). Cold Pools and Their Influence on the Tropical Marine Boundary Layer. Journal of the Atmospheric Sciences, 74(4), 1149–1168. https://doi.org/10.1175/JAS-D-16-0264.1

Eastman, R., & Wood, R. (2018). The Competing Effects of Stability and Humidity on Subtropical Stratocumulus Entrainment and Cloud Evolution from a Lagrangian Perspective, *J. Atmos. Sci.*, **75**(8), 2563-2578, https://journals.ametsoc.org/view/journals/atsc/75/8/jas-d-18-0030.1.xml

Engerer, N.A., Stensrud, D.J. and Coniglio, M.C. (2008) Surface characteristics of observed cold pools. *Monthly Weather Review*, **136**(12), 4839–4849.

Feingold, G., I. Koren, H. Wang, H. Xue, and W. Brewer, 2010: Precipitation-generated oscillations in open cellular cloud fields. Nature, 466, 849–852.

Feng, Z., Hagos, S., Rowe, A. K., Burleyson, C. D., Martini, M. N., & de Szoeke, S. P. (2015). Mechanisms of convective cloud organization by cold pools over tropical warm ocean during the AMIE/DYNAMO field campaign. Journal of Advances in Modeling Earth Systems, 7(2), 357–381. https://doi.org/https://doi.org/10.1002/2014MS000384

Gelaro, R., and Coauthors, 2017: The Modern-Era Retrospective Analysis for Research and Applications, Version 2 (MERRA-2). J. Climate, 30, 5419–5454, https://doi.org/10.1175/JCLI-D-16-0758.1

Ghate, V. P., Cadeddu, M. P., & Wood, R. (2020). Drizzle, turbulence, and density currents below post cold frontal open cellular marine stratocumulus clouds, J. Geophys. Res. Atmos., 125, e2019JD031586, https://doi.org/10.1029/2019JD031586.

Ghate, V. P., M. P. Cadeddu, X. Zheng, and E. O'Connor, 2021: Turbulence in the Marine Boundary Layer and Air Motions below Stratocumulus Clouds at the ARM Eastern North Atlantic Site. J. Appl. Meteor. Climatol., 60, 1495–1510, https://doi.org/10.1175/JAMC-D-21-0087.1.

Giangrande, S. E., Wang, D., Bartholomew, M. J., Jensen, M. P., Mechem, D. B., Hardin, J. C., & Wood,R. (2019). Midlatitude oceanic cloud and precipitation properties as sampled by the ARM Eastern North Atlantic Observatory. Journal of Geophysical Research: Atmospheres,124, 4741–4760.https://doi.org/10.1029/2018JD029667

Goren, T., Kazil, J., Hoffmann, F., Yamaguchi, T., & Feingold, G. (2019). Anthropogenic air pollution delays marine stratocumulus breakup to open cells. Geophys. Res. Lett., 46(23), 14135–14144. https://doi.org/10.1029/2019GL085412.

Hartman, D. L., and D. Short, 1980: On the use of earth radiation budget statistics for studies of clouds and climate. J. Atmos. Sci., 37, 1233–1250.

Heffter, J. L.: Transport Layer Depth Calculations, Second Joint Conference on Applications of Air Pollution Meteorology, 24–27 March 1980, New Orleans, Louisiana, https://doi.org/10.1175/1520-0477-61.1.65, 1980.

Huffman, G.J. et al. (2020). Integrated Multi-satellite Retrievals for the Global Precipitation Measurement (GPM) Mission (IMERG). In: Levizzani, V., Kidd, C., Kirschbaum, D.B., Kummerow, C.D., Nakamura, K., Turk, F.J. (eds) Satellite Precipitation Measurement. Advances in Global Change Research, vol 67. Springer, Cham. https://doi.org/10.1007/978-3-030-24568-9_19

Huffman, D. T. Bolvin, R. Joyce, E. J. Nelkin, J. Tan, D. Braithwaite, K. Hsu, O. A. Kelley, P. Nguyen, S. Sorooshian, D. C. Watters, B. J. West, and P. Xie, 2023: NASA Global Precipitation Measurement (GPM) Integrated Multi-satellite Retrievals for GPM (IMERG) Version 07. Algorithm Theoretical Basis Doc., version 7, 47 pp.

Hunzinger, A., J. C. Hardin, N. Bharadwaj, A. Varble, and A. Matthews, 2020: An extended radar relative calibration ad- justment (eRCA) technique for higher-frequency radars and range–height indicator (RHI) scans. Atmos. Meas. Tech., 13, 3147–3166, https://doi.org/10.5194/amt-13-3147-2020.

Jeong, J.-H., Witte, M. K., Glenn, I. B., Smalley, M., Lebsock, M. D., Lamer, K., & Zhu, Z. (2022). Distinct dynamical and structural properties of marine stratocumulus and shallow cumulus clouds in the Eastern North Atlantic. Journal of Geophysical Research: Atmospheres, 127, e2022JD037021. https://doi.org/10.1029/2022JD037021

Jiang, J. H., Su, H., Wu, L., Zhai, C., & Schiro, K. A. (2021). Improvements in cloud and water vapor simulations over the tropical oceans in CMIP6 compared to CMIP5. Earth and Space Science, 8, e2020EA001520, https://doi.org/10.1029/2020EA001520.

Keeler, E., Kyrouac, J., & Ermold, B. Automatic Weather Station (MAWS), 2017. Atmospheric Radiation Measurement (ARM) User Facility. https://doi.org/10.5439/1162061

Keeler, E., Burk, K., & Kyrouac, J. Balloon-Borne Sounding System (SONDEWNPN), 2013-09-28 to 2025-04-08, Eastern North Atlantic (ENA), Graciosa Island, Azores, Portugal (C1). Atmospheric Radiation Measurement (ARM) User Facility. https://doi.org/10.5439/1595321

Kirsch, B., F. Ament, and C. Hohenegger, 2021: Convective Cold Pools in Long-Term Boundary Layer Mast Observations. *Mon. Wea. Rev.*, **149**, 811–820, https://doi.org/10.1175/MWR-D-20-0197.1.

Klein, S. A., Hall, A., Norris, J. R., & Pincus, R. (2017). Low-cloud feedbacks from cloud-controlling factors: A review. Shallow clouds, water vapor, circulation, and climate sensitivity, 135-157.

Kollias, P., and Coauthors, 2016: Development and applications of ARM millimeter-wavelength cloud radars. The Atmospheric Radiation Measurement (ARM) Program: The First 20 Years, Meteor. Monogr., No. 57, Amer. Meteor. Soc., https://doi.org/10.1175/AMSMONOGRAPHS-D-15-0037.1

Kruse, I.L., Haerter, J.O. & Meyer, B. (2022) Cold pools over the Netherlands: A statistical study from tower and radar observations. *Q J R Meteorol Soc*, 711–726, https://doi.org/10.1002/qj.4223

Kyrouac, Jenni, and Yan Shi. "Surface Meteorological Instrumentation (MET)." Atmospheric Radiation Measurement (ARM) User Facility, doi:10.5439/1786358. Accessed 3 May 2023.

Lamer, K., Naud, C. M., & Booth, J. F. (2020). Relationships between precipitation properties and large-scale conditions during subsidence at the Eastern North Atlantic observatory. *Journal of Geophysical Research: Atmospheres*, 125. https://doi.org/10.1029/2019JD031848

Mechem, D. B., C. S. Wittman, M. A. Miller, S. E. Yuter, and S. P. de Szoeke, 2018: Joint Synoptic and Cloud Variability over the Northeast Atlantic near the Azores. J. Appl. Meteor. Climatol., 57, 1273–1290, https://doi.org/10.1175/JAMC-D-17-0211.1.

Myers, T.A., et al. (2021). Observational constraints on low cloud feedback reduce uncertainty of climate sensitivity. Nat. Clim. Chang. 11, 501–507, https://doi.org/10.1038/s41558-021-01039-0.

Newsom, R., Shi, Y., & Krishnamurthy, R. Doppler Lidar (DLFPT). Atmospheric Radiation Measurement (ARM) User Facility. https://doi.org/10.5439/1025185

Provod, M., J. H. Marsham, D. J. Parker, and C. E. Birch, 2016: A Characterization of Cold Pools in the West African Sahel. *Mon. Wea. Rev.*, **144**, 1923–1934, https://doi.org/10.1175/MWR-D-15-0023.1.

Redl, R., A. H. Fink, and P. Knippertz, 2015: An Objective Detection Method for Convective Cold Pool Events and Its Application to Northern Africa. *Mon. Wea. Rev.*, **143**, 5055–5072, https://doi.org/10.1175/MWR-D-15-0223.1.

Rémillard, J., and G. Tselioudis, 2015: Cloud Regime Variability over the Azores and Its Application to Climate Model Evaluation. J. Climate, 28, 9707–9720, https://doi.org/10.1175/JCLI-D-15-0066.1.

Ritsche MT. 2011. ARM Surface Meteorology Systems Handbook. U.S. Department of Energy. DOE/SC-ARM/TR-086. 10.2172/1007926.

Scott, R. C., Myers, T. A., Norris, J. R., Zelinka, M. D., Klein, S. A., Sun, M., & Doelling, D. R. (2020). Observed sensitivity of low-cloud radiative effects to meteorological perturbations over the global oceans. J. Cli., 33(18), 7717-7734.

Smalley, K. M., Lebsock, M. D., Eastman, R., Smalley, M., and Witte, M. K.: A Lagrangian analysis of pockets of open cells over the southeastern Pacific, Atmos. Chem. Phys., 22, 8197–8219, https://doi.org/10.5194/acp-22-8197-2022, 2022.

Terai, C. R., & Wood, R. (2013). Aircraft observations of cold pools under marine stratocumulus. Atmos. Chem. Phys., 13, 9899–9914, doi:10.5194/acp-13-9899-2013.

Vogel, R., Konow, H., Schulz, H., and Zuidema, P.: A climatology of trade-wind cumulus cold pools and their link to mesoscale cloud organization, Atmos. Chem. Phys., 21, 16609–16630, https://doi.org/10.5194/acp-21-16609-2021, 2021.

Wang, J., and Coauthors, 2022: Aerosol and Cloud Experiments in the Eastern North Atlantic (ACE-ENA). Bull. Amer. Meteor. Soc., 103, E619–E641, https://doi.org/10.1175/BAMS-D-19-0220.1.

Wang, D., Bartholomew, M. J., Zhu, Z., & Shi, Y. Laser Disdrometer (LD). Atmospheric Radiation Measurement (ARM) User Facility. https://doi.org/10.5439/1973058

Wilbanks, M. C., Yuter, S. E., de Szoeke, S. P., Brewer, W. A., Miller, M. A., Hall, A. M., & Burleyson, C. D. (2015). Near-Surface Density Currents Observed in the Southeast Pacific Stratocumulus-Topped Marine Boundary Layer. Mon. Wea. Rev., 143(9), 3532-3555, https://journals.ametsoc.org/view/journals/mwre/143/9/mwr-d-14-00359.1.xml.

Wood, R., 2012: Stratocumulus Clouds. Mon. Wea. Rev., 140, 2373–2423, https://doi.org/10.1175/MWR-D-11-00121.1.

Wu, P., X. Dong, and B. Xi, 2020: A Climatology of Marine Boundary Layer Cloud and Drizzle Properties Derived from Ground-Based Observations over the Azores. J. Climate, 33, 10133–10148, https://doi.org/10.1175/JCLI-D-20-0272.1.

Wyant, M. C., Bretherton, C. S., Rand, H. A., & Stevens, D. E. (1997). Numerical simulations and a conceptual model of the stratocumulus to trade cumulus transition. J. Atmos. Sci., 54, 168–192, https://doi.org/10.1175/1520-0469(1997)054<0168:NSAACM>2.0.CO;2.

Yamaguchi, T., Feingold, G., & Kazil, J. (2017). Stratocumulus to cumulus transition by drizzle. J. Adv. Model. Earth Syst., 9(6), 2333–2349, https://doi.org/10.1002/2017MS001104.

Zheng, Q., and M. A. Miller, (2022). Summertime Marine Boundary Layer Cloud, Thermodynamic, and Drizzle Morphology over the Eastern North Atlantic: A Four-Year Study. J. Climate, 35, 4805–4825, https://doi.org/10.1175/JCLI-D-21-0568.1.

| Page 21: [1] Deleted | Smalley, Mark A (US 329J-Affiliate) | 4/18/25 12:56:00 PM |
| --- | --- | --- |
| Page 21: [2] Deleted | Smalley, Mark A (US 329J-Affiliate) | 4/18/25 12:56:00 PM |
| Page 21: [3] Deleted | Smalley, Mark A (US 329J-Affiliate) | 4/18/25 12:56:00 PM |
| Page 37: [4] Deleted | Smalley, Mark A (US 329J-Affiliate) | 4/18/25 12:56:00 PM |
| Page 38: [5] Deleted | Smalley, Mark A (US 329J-Affiliate) | 4/18/25 12:56:00 PM |
| Page 38: [5] Deleted | Smalley, Mark A (US 329J-Affiliate) | 4/18/25 12:56:00 PM |
| Page 38: [6] Deleted | Smalley, Mark A (US 329J-Affiliate) | 4/18/25 12:56:00 PM |
| Page 38: [6] Deleted | Smalley, Mark A (US 329J-Affiliate) | 4/18/25 12:56:00 PM |
| Page 38: [6] Deleted | Smalley, Mark A (US 329J-Affiliate) | 4/18/25 12:56:00 PM |
| Page 38: [6] Deleted | Smalley, Mark A (US 329J-Affiliate) | 4/18/25 12:56:00 PM |
| Page 38: [6] Deleted | Smalley, Mark A (US 329J-Affiliate) | 4/18/25 12:56:00 PM |
| Page 38: [6] Deleted | Smalley, Mark A (US 329J-Affiliate) | 4/18/25 12:56:00 PM |
| Page 38: [6] Deleted | Smalley, Mark A (US 329J-Affiliate) | 4/18/25 12:56:00 PM |
| Page 38: [6] Deleted | Smalley, Mark A (US 329J-Affiliate) | 4/18/25 12:56:00 PM |
| Page 38: [6] Deleted | Smalley, Mark A (US 329J-Affiliate) | 4/18/25 12:56:00 PM |
| Page 38: [6] Deleted | Smalley, Mark A (US 329J-Affiliate) | 4/18/25 12:56:00 PM |
| Page 38: [6] Deleted | Smalley, Mark A (US 329J-Affiliate) | 4/18/25 12:56:00 PM |
| Page 38: [6] Deleted | Smalley, Mark A (US 329J-Affiliate) | 4/18/25 12:56:00 PM |

| Page 38: [6] Deleted | Smalley, Mark A (US 329J-Affiliate) | 4/18/25 12:56:00 PM |
|---|---|---|

| Page 38: [6] Deleted | Smalley, Mark A (US 329J-Affiliate) | 4/18/25 12:56:00 PM |
|---|---|---|

| Page 38: [6] Deleted | Smalley, Mark A (US 329J-Affiliate) | 4/18/25 12:56:00 PM |
|---|---|---|

| Page 38: [6] Deleted | Smalley, Mark A (US 329J-Affiliate) | 4/18/25 12:56:00 PM |
|---|---|---|

| Page 38: [6] Deleted | Smalley, Mark A (US 329J-Affiliate) | 4/18/25 12:56:00 PM |
|---|---|---|

| Page 38: [6] Deleted | Smalley, Mark A (US 329J-Affiliate) | 4/18/25 12:56:00 PM |
|---|---|---|

| Page 38: [6] Deleted | Smalley, Mark A (US 329J-Affiliate) | 4/18/25 12:56:00 PM |
|---|---|---|

| Page 38: [6] Deleted | Smalley, Mark A (US 329J-Affiliate) | 4/18/25 12:56:00 PM |
|---|---|---|

| Page 38: [6] Deleted | Smalley, Mark A (US 329J-Affiliate) | 4/18/25 12:56:00 PM |
|---|---|---|

| Page 38: [6] Deleted | Smalley, Mark A (US 329J-Affiliate) | 4/18/25 12:56:00 PM |
|---|---|---|

| Page 38: [6] Deleted | Smalley, Mark A (US 329J-Affiliate) | 4/18/25 12:56:00 PM |
|---|---|---|

| Page 38: [6] Deleted | Smalley, Mark A (US 329J-Affiliate) | 4/18/25 12:56:00 PM |
|---|---|---|

| Page 38: [6] Deleted | Smalley, Mark A (US 329J-Affiliate) | 4/18/25 12:56:00 PM |
|---|---|---|

| Page 38: [6] Deleted | Smalley, Mark A (US 329J-Affiliate) | 4/18/25 12:56:00 PM |
|---|---|---|

| Page 38: [6] Deleted | Smalley, Mark A (US 329J-Affiliate) | 4/18/25 12:56:00 PM |
|---|---|---|

| Page 38: [6] Deleted | Smalley, Mark A (US 329J-Affiliate) | 4/18/25 12:56:00 PM |
|---|---|---|

| Page 38: [7] Deleted | Smalley, Mark A (US 329J-Affiliate) | 4/18/25 12:56:00 PM |
|---|---|---|

| Page 38: [7] Deleted | Smalley, Mark A (US 329J-Affiliate) | 4/18/25 12:56:00 PM |
|---|---|---|

| Page 38: [7] Deleted | Smalley, Mark A (US 329J-Affiliate) | 4/18/25 12:56:00 PM |
|---|---|---|

| Page 38: [8] Deleted | Smalley, Mark A (US 329J-Affiliate) | 4/18/25 12:56:00 PM |
| Page 38: [8] Deleted | Smalley, Mark A (US 329J-Affiliate) | 4/18/25 12:56:00 PM |
| Page 38: [9] Deleted | Smalley, Mark A (US 329J-Affiliate) | 4/18/25 12:56:00 PM |
| Page 38: [9] Deleted | Smalley, Mark A (US 329J-Affiliate) | 4/18/25 12:56:00 PM |
| Page 39: [10] Deleted | Smalley, Mark A (US 329J-Affiliate) | 4/18/25 12:56:00 PM |
| Page 39: [10] Deleted | Smalley, Mark A (US 329J-Affiliate) | 4/18/25 12:56:00 PM |
| Page 39: [10] Deleted | Smalley, Mark A (US 329J-Affiliate) | 4/18/25 12:56:00 PM |
| Page 39: [10] Deleted | Smalley, Mark A (US 329J-Affiliate) | 4/18/25 12:56:00 PM |
| Page 39: [11] Deleted | Smalley, Mark A (US 329J-Affiliate) | 4/18/25 12:56:00 PM |
| Page 39: [11] Deleted | Smalley, Mark A (US 329J-Affiliate) | 4/18/25 12:56:00 PM |
| Page 39: [11] Deleted | Smalley, Mark A (US 329J-Affiliate) | 4/18/25 12:56:00 PM |
| Page 39: [11] Deleted | Smalley, Mark A (US 329J-Affiliate) | 4/18/25 12:56:00 PM |
| Page 39: [11] Deleted | Smalley, Mark A (US 329J-Affiliate) | 4/18/25 12:56:00 PM |
| Page 39: [11] Deleted | Smalley, Mark A (US 329J-Affiliate) | 4/18/25 12:56:00 PM |
| Page 39: [11] Deleted | Smalley, Mark A (US 329J-Affiliate) | 4/18/25 12:56:00 PM |
| Page 39: [11] Deleted | Smalley, Mark A (US 329J-Affiliate) | 4/18/25 12:56:00 PM |
| Page 39: [11] Deleted | Smalley, Mark A (US 329J-Affiliate) | 4/18/25 12:56:00 PM |

| Page 39: [12] Deleted | Smalley, Mark A (US 329J-Affiliate) | 4/18/25 12:56:00 PM |
|---|---|---|
| Page 39: [12] Deleted | Smalley, Mark A (US 329J-Affiliate) | 4/18/25 12:56:00 PM |
| Page 39: [12] Deleted | Smalley, Mark A (US 329J-Affiliate) | 4/18/25 12:56:00 PM |
| Page 39: [12] Deleted | Smalley, Mark A (US 329J-Affiliate) | 4/18/25 12:56:00 PM |
| Page 39: [12] Deleted | Smalley, Mark A (US 329J-Affiliate) | 4/18/25 12:56:00 PM |
| Page 39: [12] Deleted | Smalley, Mark A (US 329J-Affiliate) | 4/18/25 12:56:00 PM |
| Page 39: [12] Deleted | Smalley, Mark A (US 329J-Affiliate) | 4/18/25 12:56:00 PM |
| Page 39: [13] Deleted | Smalley, Mark A (US 329J-Affiliate) | 4/18/25 12:56:00 PM |
| Page 39: [13] Deleted | Smalley, Mark A (US 329J-Affiliate) | 4/18/25 12:56:00 PM |
| Page 39: [13] Deleted | Smalley, Mark A (US 329J-Affiliate) | 4/18/25 12:56:00 PM |
| Page 39: [13] Deleted | Smalley, Mark A (US 329J-Affiliate) | 4/18/25 12:56:00 PM |
| Page 39: [13] Deleted | Smalley, Mark A (US 329J-Affiliate) | 4/18/25 12:56:00 PM |
| Page 39: [14] Deleted | Smalley, Mark A (US 329J-Affiliate) | 4/18/25 12:56:00 PM |
| Page 39: [14] Deleted | Smalley, Mark A (US 329J-Affiliate) | 4/18/25 12:56:00 PM |
| Page 39: [14] Deleted | Smalley, Mark A (US 329J-Affiliate) | 4/18/25 12:56:00 PM |
| Page 39: [14] Deleted | Smalley, Mark A (US 329J-Affiliate) | 4/18/25 12:56:00 PM |
| Page 39: [14] Deleted | Smalley, Mark A (US 329J-Affiliate) | 4/18/25 12:56:00 PM |
| Page 41: [15] Deleted | Smalley, Mark A (US 329J-Affiliate) | 4/18/25 12:56:00 PM |
| Page 42: [16] Deleted | Smalley, Mark A (US 329J-Affiliate) | 4/18/25 12:56:00 PM |

| Page 43: [17] Deleted | Smalley, Mark A (US 329J-Affiliate) | 4/18/25 12:56:00 PM |

| Page 43: [18] Deleted | Smalley, Mark A (US 329J-Affiliate) | 4/18/25 12:56:00 PM |

1.